# Designing Algorithms Empowered by Language Models: An Analytical Framework, Case Studies, and Insights

**Yanxi Chen**                                        *chenyanxi.cyx@alibaba-inc.com*
*Alibaba Group*

**Yaliang Li**                                        *yaliang.li@alibaba-inc.com*
*Alibaba Group*

**Bolin Ding**                                        *bolin.ding@alibaba-inc.com*
*Alibaba Group*

**Jingren Zhou**                                        *jingren.zhou@alibaba-inc.com*
*Alibaba Group*

**Reviewed on OpenReview:** *https://openreview.net/forum?id=nJ7RECkxQr*

## Abstract

This work presents an analytical framework for the design and analysis of LLM-based algorithms, i.e., algorithms that contain one or multiple calls of large language models (LLMs) as sub-routines and critically rely on the capabilities of LLMs. While such algorithms, ranging from basic LLM calls with prompt engineering to complicated LLM-powered agentic workflows and compound AI systems, have achieved remarkable empirical success, their design and optimization oftentimes require extensive trial-and-errors and case-by-case analysis. Our proposed framework serves as an attempt to mitigate such headaches, offering a formal and systematic approach for analyzing how the accuracy and efficiency of an LLM-based algorithm will be impacted by critical design choices, such as the pattern and granularity of task decomposition, or the prompt for each LLM call. Through a wide range of case studies covering diverse algorithm patterns (including parallel/hierarchical/recursive task decomposition and generic directed acyclic graphs), we demonstrate the proposed framework in action and derive interesting insights that generalize across scenarios, accompanied by systematic empirical validation in synthetic settings.

## 1 Introduction

The rapid advancements of pre-trained large language models (LLMs) (Bubeck et al., 2023; Wei et al., 2022a; Schaeffer et al., 2023) have given rise to the new paradigm of algorithmic problem solving, which utilizes LLMs as general-purpose solvers and prompts them with task descriptions and additional inputs that enhance their performance (Wei et al., 2022b; Kojima et al., 2022; Zhang et al., 2024b). However, even the most advanced LLMs today exhibit limitations, such as finite context window sizes and difficulty with complex reasoning. Some of these limitations are fundamental (Merrill & Sabharwal, 2023; Peng et al., 2024; Thomm et al., 2024; Hahn & Rofin, 2024; Wen et al., 2024; Merrill et al., 2024), and resolving them requires new breakthroughs. Moreover, in resource-constrained scenarios where using the state-of-the-art LLMs is not feasible, one might have to resort to smaller and weaker LLMs for solving complex tasks.

All these have motivated the developments of what we call *LLM-based algorithms*, i.e., algorithms that contain one or multiple LLM calls as sub-routines and fundamentally rely on the capabilities of LLMs. In its most basic form, an LLM-based algorithm can be a combination of one or multiple LLM calls and some prompt engineering (Yao et al., 2023a; Besta et al., 2023; Zhou et al., 2023; Wang et al., 2023).

More advanced examples include LLM-powered agentic workflows (Mialon et al., 2023; Pezeshkpour et al., 2024; Xi et al., 2023) and compound AI systems (Zaharia et al., 2024) that augment LLMs with additional abilities like tool use and long-term memory, as well as the emerging paradigm of LLM programming (Schlag et al., 2023; Khattab et al., 2024; Zheng et al., 2024). LLM-based algorithms, in a similar spirit to neuro-symbolic programming (Chaudhuri et al., 2021; Gupta & Kembhavi, 2023) and learning-augmented algorithms (Mitzenmacher & Vassilvitskii, 2022; Lindermayr & Megow, 2022), combine the advantages of LLMs and traditional algorithms. An LLM-based algorithm, designed with human's intelligence and knowledge of algorithmic problem solving, can exhibit much better controllability and interpretability, stronger performance that is less reliant on delicate prompting, and capabilities that far exceed what a single LLM call can achieve.

**Goal and motivations.** A key principle underlying LLM-based algorithms is *task decomposition*. This is exemplified by the ReAct algorithm (Yao et al., 2023b) that solve a problem via a sequence of reasoning and acting steps, or long-text summarization methods that process the original text by chunks (Kryscinski et al., 2022; Chang et al., 2024). The art of LLM-based algorithm design often lies in figuring out how to decompose the original task into sub-tasks appropriately, so that each can be handled by one LLM call or non-LLM program accurately and efficiently, and the performance of the overall algorithm is guaranteed. Numerous questions can arise in this process, such as:

- While finer-grained decomposition makes each sub-task easier, it also increases the number of sub-tasks, which might potentially amplify the risk of error accumulation or cascading that could lead to catastrophic failure of the overall algorithm. What would be the appropriate granularity of task decomposition that achieves a balance of these two aspects?

- Different ways of task decomposition incur different accuracy and efficiency. Does higher accuracy necessarily come with worsened efficiency? When is it possible to achieve the best of both?

- Long chain-of-thought (CoT) reasoning (Wei et al., 2022b; OpenAI, 2024a; DeepSeek-AI et al., 2025) can enhance the output quality of an LLM call, but also at a substantially higher cost. Is the cost overhead worthy of the gain in accuracy? Should we apply CoT reasoning in solving every sub-task, apply it to a small number of critical sub-tasks only (and tolerate errors in other sub-tasks), or not using CoT at all?

Answering such questions often relies on extensive trial-and-errors in practice. Worse still, the obtained answers are often specific to a particular scenario and do not generalize to others. These challenges call for a more formal and systematic approach of algorithm design and analysis, which might help to predict the empirical performance of LLM-based algorithms, explain curious empirical phenomena, guide the choices of hyperparameters, and potentially inspire better algorithm design.

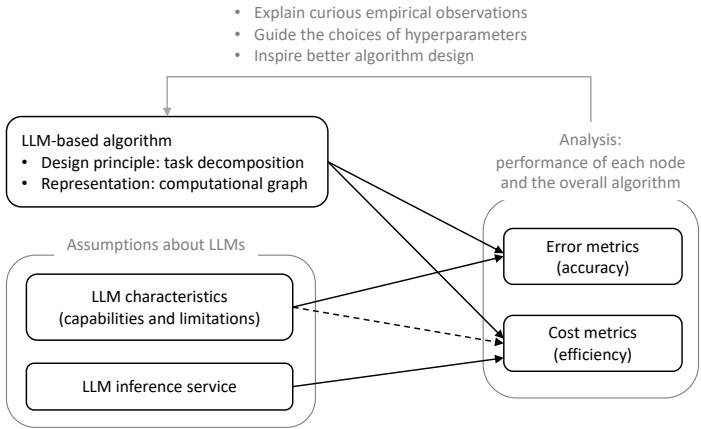

Figure 1: An overview of the proposed analytical framework.

**Main contributions.** As an attempt towards achieving this goal, we propose an analytical framework for the design and analysis of LLM-based algorithms, which is summarized in Figure 1 and elaborated in Section 2. The framework represents an LLM-based algorithm as a computational graph (whose nodes correspond to the sub-tasks), and identifies key abstractions such as LLM characteristics (i.e., capabilities and limitations) and LLM inference service. Built

upon these, error and cost metrics of the overall algorithm are determined by those of all graph nodes, in line with the task decomposition principle.

We further demonstrate the proposed framework in action for a diverse variety of case studies, deriving interesting and practical insights that generalize across scenarios. In particular, Section 3 investigates parallel decomposition, an elementary yet fundamental pattern of LLM-based algorithms, with a focus on understanding how the granularity of task decomposition impacts the efficiency (Insight 1) and accuracy (Insight 2) of the overall algorithm. Section 4 further studies more general patterns of LLM-based algorithms, offering insights for unified error analysis of generic algorithms under certain technical assumptions (Insight 3), critical design choices for iterative retrieval and reasoning (Insight 4), and making the proposed framework applicable to recursive task decomposition via unrolling (Insight 5).

## 2 An analytical framework for LLM-based algorithms

This section presents our proposed framework for the design and analysis of LLM-based algorithms.

### 2.1 Definition and representation

An LLM-based algorithm is simply an algorithm that contains one or multiple LLM calls as its key components. The simplest example would be a single LLM call. More generally, an LLM-based algorithm can be an agentic workflow or compound AI system that utilizes multiple LLM calls and non-LLM programs to handle sub-tasks derived from the original problem, whose outputs together give rise to the final solution of the overall algorithm.

An LLM-based algorithm can be naturally represented as a *computational graph.* Each graph node takes some inputs from its predecessor nodes, executes certain operations, and returns some outputs. The nodes can be categorized into two types — LLM nodes and non-LLM nodes — as demonstrated in Figure 2. Within an *LLM node*, the operations consist of a prompter that formats a prompt based on the inputs, an LLM call that processes the prompt[1], and a parser that extracts the targeted information from the LLM's response. The prompter and parser, designed for the sub-task of the current node, serve as translators between natural language and traditional data structures. Within a *non-LLM node*, the operations can be anything that does not involve LLMs. Examples include a symbolic algorithm, an API call for a search engine, or a classical machine learning model.

Given such nodes, the computational graph is built by connecting them with directed edges that specify the data flows within the LLM-based algorithm. For example, Figure 3a shows the graph for the pattern of parallel decomposition, which divides the input problem into parallel sub-tasks, solves each of them with one LLM node, and aggregates the results for the final solution; Figure 3b illustrates an algorithm for book-length summarization (Chang et al., 2024; OpenAI, 2024b), which divides the input text into multiple chunks and maintains a global summary via incremental updating; and Figure 3c represents the ReAct algorithm (Yao et al., 2023b), which consists of multiple iterations of reasoning, tool use and aggregation. The computational-graph formulation is expressive enough to cover these diverse LLM-based algorithms that have been proposed separately in prior works, and facilitates formal analysis for all of them in a unified manner. The computational graph of an LLM-based algorithm can be pre-specified and static, or dynamically constructed at runtime, as will be demonstrated later in our case studies in Sections 3 and 4. Configurations of an LLM-based algorithm include how task decomposition is done (reflected by the graph topology and the sub-tasks of graph nodes), the methods of prompting and parsing for each LLM node, configurations of the LLM(s) being used, among others. The LLM nodes within one graph might use the same or different backbone LLMs.

### 2.2 Error and cost analysis

Given an input task and the corresponding LLM-based algorithm, we aim to analyze its performance — how accurately and efficiently the algorithm solves the task — akin to analysis for any generic algorithm.

---

[1]We assume without loss of generality that each LLM node contains one single LLM call, unless specified otherwise.

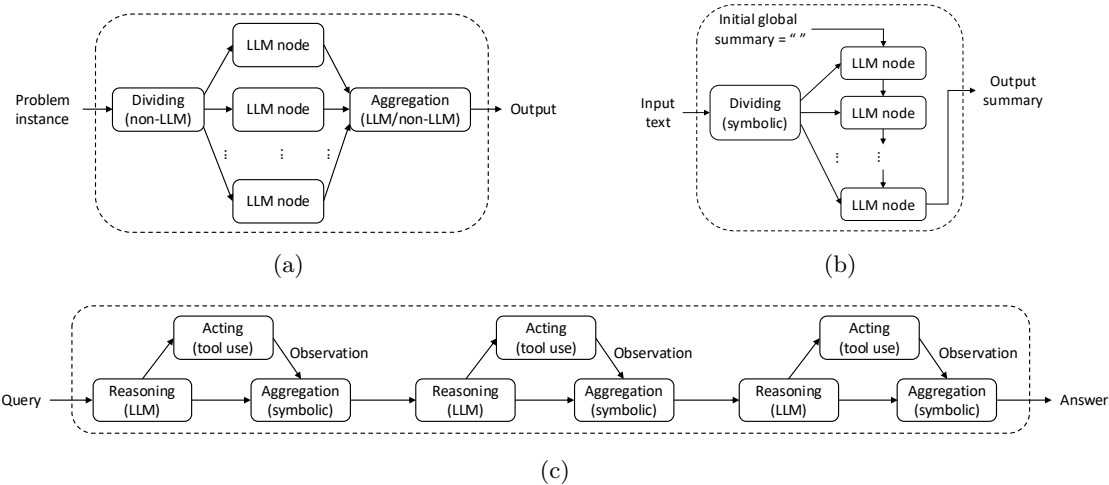

Figure 2: An LLM node (left) or non-LLM node (right) in the computational graphs of LLM-based algorithms. Each node can have one or multiple inputs/outputs. We use the abbreviation "NL" for "natural language", and "DS" for "data structure".

Figure 3: Examples of computational graph representations for LLM-based algorithms. **(a)** Parallel decomposition. Detailed analysis and concrete examples for this graph pattern will be elaborated in Section 3. **(b)** Book-length summarization, cf. Figure 1 in (Chang et al., 2024). The "dividing" node contains a symbolic program that divides the input text into multiple smaller chunks. **(c)** The ReAct algorithm (Yao et al., 2023b). Each "acting" node represents one API call for a certain tool, and each "aggregation" node aggregates the outputs of its predecessor nodes, e.g., by simple concatenation.

Following the task decomposition principle, error and cost analysis is done by analyzing for each LLM or non-LLM node first, and then for the overall algorithm. Unless otherwise specified, our analysis is deterministic, for a given task instance and fixed random seed(s).

**Error and cost metrics.** The accuracy and efficiency of each graph node and the overall LLM-based algorithm can be quantified by certain error metrics and cost metrics respectively. The performance of individual graph nodes, together with the graph topology, implies the performance of the overall algorithm. For each specific task or algorithm, one might define multiple error metrics and cost metrics, which can be analyzed in a unified manner within our proposed framework.

*Error metrics* are task-specific in general. Moreover, the overall algorithm and the LLM nodes within it might have different error metrics. For instance, consider an LLM-based algorithm that answers a given question by first retrieving relevant sentences from the input text with multiple LLM nodes, and then reasoning over the extracted sentences with another LLM node. In this case, the overall algorithm is evaluated by whether it answers the question correctly, while each LLM node responsible for retrieval is evaluated by whether it successfully extracts all necessary sentences within the input text that it handles.[2]

*Cost metrics*, on the other hand, are largely task-agnostic. Examples include the total length of prompts and total length of generated texts, measured by the numbers of characters or tokens, within one run of the overall algorithm; these metrics are correlated with the monetary costs for using proprietary LLMs via commercial

---

[2]Throughout this work, we use "accuracy" to refer to the broader concept of "quality", and an "error metric" can be any metric that measures how much the output of an algorithm deviates from certain criteria.

API calls, as well as the time and memory complexities of LLM inference. Another important cost metric is the end-to-end latency of running the overall algorithm; one particular property of this metric is that it can be impacted by parallelism of LLM calls, an important aspect of LLM inference service in practice. Other possible cost metrics include the peak memory usage, the total number of LLM calls, FLOPs, energy consumption, carbon emission, among others. Unless specified otherwise, we focus on the costs of the LLM nodes and neglect those of the non-LLM nodes, since the latter is much smaller than the former in all concrete scenarios that we will consider later in this work.

**LLM characteristics and inference service.** For error and cost analysis of an LLM-based algorithm, it is necessary to make certain assumptions about the characteristics and inference service of the LLM(s) used within it.

*Characteristics of LLMs*, namely their *capabilities and limitations*, determine what the generated text will be for a specific prompt, and thus directly affect the error metrics of individual graph nodes and the overall algorithm. They also affect the cost metrics indirectly, via the lengths of prompts and generated texts. Assumptions on LLM characteristics can be task-specific or task-agnostic.

Assumptions on *LLM inference service* (Yuan et al., 2024; Pope et al., 2023; Zhou et al., 2024), on the other hand, are task-agnostic and only affect the cost metrics, not error metrics. In particular, they determine the relation between the cost metrics and the lengths of prompt and generated text for each LLM call, the parallelism of LLM calls, among others. While LLM inference service in practice can be very diverse, we will see very soon that unified and formal analysis is possible with appropriate abstractions of it.

**Error analysis.** The error metrics for the output of each LLM node, which are calculated with respect to what the output should have been if all nodes accomplish their tasks with exact accuracy, depend on the characteristics, i.e., capabilities and limitations, of the LLM, as well as the specific problem instance and random seed(s). For a certain node $v$, the error of its output $y$ can be bounded by some function $f_v$ (which depends on the aforementioned factors if $v$ is an LLM node) of the errors of its inputs $x_1, x_2, \ldots, x_k$, i.e., the outputs of its predecessor nodes:

$$\mathcal{E}(y) \leq f_v\big(\mathcal{E}(x_1), \mathcal{E}(x_2), \ldots, \mathcal{E}(x_k)\big). \tag{1}$$

The function $f_v$ can be general, for example:

- $\mathcal{E}(y) \leq \sum_{i \in [k]} \mathcal{E}(x_i)$ or $\frac{1}{k} \sum_{i \in [k]} \mathcal{E}(x_i)$. Such linear relationship will appear in a counting task that will be elaborated in Section 3.

- $\mathcal{E}(y) = \min_{i \in [k]} \mathcal{E}(x_i)$. In a coding task, the algorithm might generate multiple code samples, and succeed as long as one of the samples is correct (by passing all test cases).

- $\mathcal{E}(y) \leq \max_{i \in [k]} \mathcal{E}(x_i)$. This is the case for a sorting task to be introduced in Section 3, where node $v$ is responsible for merging multiple sorted lists and $\mathcal{E}$ stands for the $\ell_\infty$ error.

Finally, the error metrics of the overall algorithm are exactly those of the particular graph node that generates the final solution returned by the algorithm.

**Cost analysis.** Let us first consider the cost of one LLM call for a specific input prompt, which consists of a *prefilling phase* with a prompt of length $\mathcal{L}_{\mathsf{pre}}$, and a *decoding phase* that generates text of length $\mathcal{L}_{\mathsf{dec}}$. In our framework, we assume that the cost $\mathcal{C}$ of one LLM call can be bounded by

$$\mathcal{C} \leq \mathcal{C}(\text{prefilling}) + \mathcal{C}(\text{decoding}) = \mathcal{C}_{\mathsf{pre}}(\mathcal{L}_{\mathsf{pre}}) + \mathcal{C}_{\mathsf{dec}}(\mathcal{L}_{\mathsf{pre}}, \mathcal{L}_{\mathsf{dec}}) =: \mathcal{C}_{\mathsf{LLM}}(\mathcal{L}_{\mathsf{pre}}, \mathcal{L}_{\mathsf{dec}}). \tag{2}$$

The functions $\mathcal{C}_{\mathsf{pre}}$ and $\mathcal{C}_{\mathsf{dec}}$ are specific to each LLM call, and depend on the choices of cost metrics and LLM inference service. For example:

- Charges of API calls: $\mathcal{C}_{\mathsf{pre}}$ and $\mathcal{C}_{\mathsf{dec}}$ are linear functions, namely $\mathcal{C}_{\mathsf{pre}}(\mathcal{L}_{\mathsf{pre}}) = c_{\mathsf{pre}} \times \mathcal{L}_{\mathsf{pre}}$, $\mathcal{C}_{\mathsf{dec}}(\mathcal{L}_{\mathsf{pre}}, \mathcal{L}_{\mathsf{dec}}) = c_{\mathsf{dec}} \times \mathcal{L}_{\mathsf{dec}}$, where $c_{\mathsf{pre}}$ and $c_{\mathsf{dec}}$ are the cost per token for the prompt and generated text respectively.

- Latency: in a memory-bound scenario (Agarwal et al., 2023) where $\mathcal{L}_{\mathsf{pre}}$ is limited and the latency of memory IO (for loading model weights and KV caches) is dominant, we have approximately $\mathcal{C}_{\mathsf{pre}}(\mathcal{L}_{\mathsf{pre}}) = O(1)$ and $\mathcal{C}_{\mathsf{dec}}(\mathcal{L}_{\mathsf{pre}}, \mathcal{L}_{\mathsf{dec}}) = O(\mathcal{L}_{\mathsf{dec}})$; more generally, $\mathcal{C}_{\mathsf{pre}}$ and $\mathcal{C}_{\mathsf{dec}}$ might grow linearly or quadratically (e.g., for Transformers with full attention (Vaswani et al., 2017)) with $\mathcal{L}_{\mathsf{pre}}$ and $\mathcal{L}_{\mathsf{dec}}$.

The cost of the overall algorithm can be written as a function of the cost metrics of all LLM nodes within its computational graph. It might be a simple sum, e.g., for LLM API costs, or a more complex function, e.g., for the end-to-end latency in the presence of parallelism. As an example, consider $k$ independent LLM calls with latencies $\mathcal{C}_1, \mathcal{C}_2, \ldots, \mathcal{C}_k$ and *ideal* parallelism of degree $p$, i.e., no extra cost is induced by parallelism; then, the $k$ LLM calls can be divided into $g = \lceil k/p \rceil$ sequential groups, where each group of $p$ parallel LLM calls is bottlenecked by the one with the largest latency, and thus the end-to-end latency becomes

$$\mathcal{C} = \sum_{j \in [g]} \max \left\{ \mathcal{C}_{(j-1)p+1}, \mathcal{C}_{(j-1)p+2}, \ldots, \mathcal{C}_{\min\{jp, k\}} \right\}. \tag{3}$$

This result will be especially useful for our analysis of parallel decomposition later in Section 3.

*Remark* 1. When applying the proposed framework to a new task, we need to first characterize the capabilities and limitations of LLMs for this task (or for the sub-tasks derived from it), e.g., via profiling with a handful of "training samples". This characterization can be quantitative or qualitative, precise or coarse, depending on the purpose of analysis. For example, in many tasks of interest, a larger problem instance is harder than a smaller one, and thus incurs larger error metrics of an LLM call. For practical purposes like hyperparameter optimization, such coarse characterization might be sufficient already.

*Remark* 2. Understanding LLM inference service, especially from a system perspective, is crucial for in-depth analysis of cost metrics. LLM inference service can be diverse in practice: LLMs might run on CPUs in a personal laptop or on a distributed GPU cluster, inference might be compute-bound or memory-bound, the complexity of long-sequence processing and generation might be linear or quadratic, parallelism at various levels (e.g., multiple LLMs deployed on multiple machines, or batch inference with one LLM) might be supported, and so on. These are covered by our proposed framework in a unified manner.

## 3 Case studies and insights: parallel decomposition

This section is dedicated to parallel decomposition, a basic MapReduce-like pattern (Dean & Ghemawat, 2008) visualized in Figure 3a. For concreteness, we consider three canonical tasks, which are abstract and synthetic versions of practical LLM applications. (1) *Counting*: given a string of length $n$ consisting of letters and digits, count the number of digits in it. (2) *Sorting*: given a list of $n$ numbers, sort it in ascending order. (3) *Retrieval*: answer a given question that requires retrieving some key information from a long piece of text. Table 1 summarizes the LLM-based algorithms for solving these tasks by parallel decomposition, and Table 2 presents a worked example for the counting task. Supplementary details for these tasks, as well as additional examples of parallel decomposition, can be found in Appendix A.

A problem of particular interest in practice is: *what is the proper granularity of task decomposition, or the sub-task size $m$, for solving a task of size $n$ by parallel decomposition?* More specifically, we strive to answer the following questions:

> **Q1.** How does the overall cost of the LLM-based algorithm change with the sub-task size $m$?
>
> **Q2.** Does a smaller $m$, i.e., finer-grained task decomposition, always imply a smaller final error?

Throughout this section, we answer these questions by leveraging the analytical framework proposed in Section 2. The results are summarized in Insights 1 and 2, which will also be validated empirically.

**Why synthetic tasks?** For the purpose of building a solid foundation for rigorously understanding the accuracy and efficiency of LLM-based algorithms, we have chosen to stick with synthetic settings for our empirical studies in this work. This brings various benefits, such as reducing the risk of test data contamination, facilitating unambiguous and precise measurement of accuracy, allowing full transparency

Table 1: Three concrete examples of parallel decomposition.

| Task | Counting | Sorting | Retrieval |
|---|---|---|---|
| Step 1 | Divide the input string into $k$ disjoint sub-strings of lengths $m_1, m_2, \ldots, m_k$. | Divide the input list into $k$ disjoint sub-lists of lengths $m_1, m_2, \ldots, m_k$. | Divide the input text into $k$ overlapping chunks of lengths $m_1, m_2, \ldots, m_k$. |
| Step 2 | For each $i \in [k]$, use one LLM call to count the number of digits in the $i$-th sub-string, which returns an answer $y_i$. | For each $i \in [k]$, use one LLM call to sort the $i$-th sub-list, which returns a solution $\boldsymbol{y}_i$. | For each $i \in [k]$, use one LLM call to try to answer the question based on that chunk. |
| Step 3 | The final solution of the algorithm is $y = \sum_{i \in [k]} y_i$. | Merge the sub-lists $\boldsymbol{y}_1, \ldots, \boldsymbol{y}_k$ into a single list $\boldsymbol{y}$ using a symbolic algorithm (Listing 1). | Generate the final answer by majority voting* of the valid answers†. |

\* We adopt majority voting because, as the input text is chunked with overlapping, the correct answer might appear multiple times in Step 2. Moreover, there might exist false positives in Step 2, in which case majority voting brings extra robustness.

† In Step 2, an LLM call might return "I don't know" if it decides that the corresponding chunk does not contain sufficient information for answering the question. Such an answer is considered invalid, and excluded from majority voting in Step 3.

Table 2: A worked example for the counting task, showing the prompt and response of a specific LLM node.

```
1  # Prompt
2  User: Count the number of digits in the following string:
3  dhd73695051jcbcg9jbeh903bi64ic736icfag96
4  Answer in the following format: 'There are {answer} digits in the given string'.
5  Do NOT output anything else.
6
7  # Response
8  Assistant: There are 19 digits in the given string
```

and control over task configurations, and making the current work as self-contained as possible. We start with the simplest settings and algorithms (for didactic purpose) in this section, and will gradually expand our proposed framework to more complex and practical scenarios resembling those in prior empirical works, e.g., retrieval-augmented generation in Appendix A.5, iterative retrieval and reasoning in Section 4.2, or active information seeking and aggregation with recursive task decomposition in Section 4.3.

**Formal notations.** We let $n$ denote the size of the input problem instance; it can be the length (e.g., the number of characters or tokens) of the text that represents the input problem instance, or defined in terms of traditional data structures (e.g., the length of a list). Let $k$ denote the number of parallel sub-tasks after decomposition, and $m_i$ denote the size of the $i$-th sub-task, where $i \in [k]$. It is assumed that $m_i \leq \overline{m}$ for some maximum value $\overline{m}$ that can fit into the context window of the LLM used by the algorithm. For most of our analysis, we will assume for simplicity that $m_i = m$ for all $i \in [k]$, and that $k = \Theta(n/m)$. We denote $p \geq 1$ as the maximum degree of parallelism supported by LLM inference service. Finally, let $\mathcal{L}_{\mathsf{sys}}$ be an upper bound for the length of the system prompt of each LLM call, which includes everything in the prompt except for the input problem instance itself. The presence of $\mathcal{L}_{\mathsf{sys}}$, which can be large in practice, is essentially due to the fact that the LLM is used as a general-purpose sub-routine, hence specifying the concrete task for each LLM call constitutes part of the complexity.

For notational convenience, we will often write $f(n) \lesssim g(n)$ in place of $f(n) = O(g(n))$, which means there exists a universal constant $C > 0$ such that $f(n) \leq C \cdot g(n)$ for any positive integer $n$. In addition, $f(n) \asymp g(n)$ means $f(n) \lesssim g(n)$ and $g(n) \lesssim f(n)$ both hold. To further simplify notation, we will often omit the big-O notation in the input arguments of cost functions $\mathcal{C}_{\mathsf{pre}}/\mathcal{C}_{\mathsf{dec}}/\mathcal{C}_{\mathsf{LLM}}$; for example, given a prompt of length $\mathcal{L}_{\mathsf{sys}} + O(n)$, we will write the cost of the prefilling phase as $\mathcal{C}_{\mathsf{pre}}(\mathcal{L}_{\mathsf{sys}} + n)$ rather than $\mathcal{C}_{\mathsf{pre}}(\mathcal{L}_{\mathsf{sys}} + O(n))$.

### 3.1 Answering Q1: cost analysis

We assume for simplicity that all parallel LLM nodes share the same LLM backend and inference service, and thus the same cost functions $\mathcal{C}_{\mathsf{pre}}$, $\mathcal{C}_{\mathsf{dec}}$ and $\mathcal{C}_{\mathsf{LLM}}$.

**The direct sum of costs.** In many cases, the cost of the overall algorithm is the direct sum of the costs of all LLM calls involved, e.g., the monetary cost of API calls for proprietary LLMs, or the end-to-end latency when all LLM calls are executed sequentially. In such cases, we can write the total cost $\mathcal{C}$ as

$$\mathcal{C} = \mathcal{C}(\text{sub-tasks}) + \mathcal{C}(\text{aggregation}),$$

where $\mathcal{C}(\text{aggregation})$ is the cost of the final aggregation step, and

$$\mathcal{C}(\text{sub-tasks}) = k \times \mathcal{C}(\text{one sub-task}) \le k \times \mathcal{C}_{\mathsf{LLM}}(\mathcal{L}_{\mathsf{sys}} + m, \mathcal{L}_{\mathsf{dec}})$$
$$\lesssim n \times \frac{\mathcal{C}_{\mathsf{LLM}}(\mathcal{L}_{\mathsf{sys}} + m, \mathcal{L}_{\mathsf{dec}})}{m}. \tag{4}$$

Here, the first inequality follows Eq. (2), the second inequality follows $k \lesssim n/m$, and $\mathcal{L}_{\mathsf{dec}} = \mathcal{L}_{\mathsf{dec}}(m)$ is task-specific, e.g., $O(1)$ for the counting and retrieval tasks, or $O(m)$ for the sorting task.

This basic analysis already provides some hints for tuning the hyperparameter $m$:

- If $\mathcal{C}_{\mathsf{LLM}}(\mathcal{L}_{\mathsf{sys}} + m, \mathcal{L}_{\mathsf{dec}}) = \mathcal{C}_{\mathsf{pre}}(\mathcal{L}_{\mathsf{sys}} + m) + \mathcal{C}_{\mathsf{dec}}(\mathcal{L}_{\mathsf{sys}} + m, \mathcal{L}_{\mathsf{dec}})$ grows with $m$ at a linear or sub-linear rate, then the right-hand side of Eq. (4) is monotonely decreasing in $m$, which means $\mathcal{C}(\text{sub-tasks})$ is minimized at $m = \min\{n, \overline{m}\}$.

- A Transformer model with full attention suffers from quadratic complexity in long-sequence processing (Yuan et al., 2024; Zhou et al., 2024). With the simplified assumption that $\mathcal{C}_{\mathsf{LLM}}(\mathcal{L}_{\mathsf{sys}} + m, \mathcal{L}_{\mathsf{dec}}) \lesssim (\mathcal{L}_{\mathsf{sys}} + m)^2$, we have

$$\mathcal{C}(\text{sub-tasks}) \lesssim n \times \frac{\mathcal{C}_{\mathsf{LLM}}(\mathcal{L}_{\mathsf{sys}} + m, \mathcal{L}_{\mathsf{dec}})}{m} \lesssim n \times \frac{(\mathcal{L}_{\mathsf{sys}} + m)^2}{m} = n \times \left( \frac{\mathcal{L}_{\mathsf{sys}}^2}{m} + m + 2\mathcal{L}_{\mathsf{sys}} \right).$$

  Assuming that $\overline{m}$ is sufficiently large, the right-hand side achieves the minimum $4 \times n \times \mathcal{L}_{\mathsf{sys}}$ at an intermediate value $m = \mathcal{L}_{\mathsf{sys}}$.

- More generally, one may assume that

$$\mathcal{C}_{\mathsf{LLM}}(\mathcal{L}_{\mathsf{sys}} + m, \mathcal{L}_{\mathsf{dec}}) \le \alpha \times (\mathcal{L}_{\mathsf{sys}} + m)^2 + \beta \times (\mathcal{L}_{\mathsf{sys}} + m) + \gamma,$$

  which takes into account a more precise characterization of LLM inference service, including the FLOPs as well as memory IO for loading model weights and KV caches. In this case, we have

$$\mathcal{C}(\text{sub-tasks}) \lesssim n \times \frac{\mathcal{C}_{\mathsf{LLM}}(\mathcal{L}_{\mathsf{sys}} + m, \mathcal{L}_{\mathsf{dec}})}{m}$$
$$\le n \times \frac{\alpha \times (\mathcal{L}_{\mathsf{sys}} + m)^2 + \beta \times (\mathcal{L}_{\mathsf{sys}} + m) + \gamma}{m}$$
$$= n \times \left( \alpha \times m + \frac{\alpha \times \mathcal{L}_{\mathsf{sys}}^2 + \beta \times \mathcal{L}_{\mathsf{sys}} + \gamma}{m} + 2 \times \alpha \times \mathcal{L}_{\mathsf{sys}} + \beta \right),$$

  and the right-hand side is minimized at $m = \sqrt{\mathcal{L}_{\mathsf{sys}}^2 + \mathcal{L}_{\mathsf{sys}} \times \beta/\alpha + \gamma/\alpha}$.

**The latency with parallelism.** The above analysis can be extended to the case with parallelism, which is especially relevant to the end-to-end latency of the overall algorithm. Considering parallel LLM calls for homogeneous sub-tasks in an ideal setting where parallelism incurs no additional cost, we have

$$\mathcal{C}(\text{sub-tasks}) \le \left\lceil \frac{k}{p} \right\rceil \times \mathcal{C}_{\mathsf{LLM}}(\mathcal{L}_{\mathsf{sys}} + m, \mathcal{L}_{\mathsf{dec}}) \tag{5}$$

according to Eq. (3). For large $m$ such that $k \leq p$, we have $\lceil k/p \rceil = 1$, and thus $\mathcal{C}(\text{sub-tasks}) \leq \mathcal{C}_{\text{LLM}}(\mathcal{L}_{\text{sys}} + m, \mathcal{L}_{\text{dec}})$ is monotonely *increasing* in $m$. On the other hand, for sufficiently small $m$ and hence large $k$, we have $\lceil k/p \rceil \approx k/p$, and thus the upper bound for $\mathcal{C}(\text{sub-tasks})$ can be approximated by

$$\mathcal{C}(\text{sub-tasks}) \leq \frac{k}{p} \times \mathcal{C}_{\text{LLM}}(\mathcal{L}_{\text{sys}} + m, \mathcal{L}_{\text{dec}}) \lesssim \frac{n}{p} \times \frac{\mathcal{C}_{\text{LLM}}(\mathcal{L}_{\text{sys}} + m, \mathcal{L}_{\text{dec}})}{m},$$

which might be monotonely *decreasing* in $m$ if $\mathcal{C}_{\text{LLM}}(\mathcal{L}_{\text{sys}} + m, \mathcal{L}_{\text{dec}})$ grows with $m$ at a linear or sub-linear rate. In this case, the overall cost $\mathcal{C}(\text{sub-tasks})$ is minimized by $k \asymp p$ and hence $m \asymp n/p$.

**Summary.** With the above analysis, our answer to Q1 is summarized as follows:

---

**Insight 1.** The relationship between the overall cost $\mathcal{C}(\text{sub-tasks})$ and the sub-task size $m$ depends on various factors, such as the choices of cost metrics and assumptions about LLM inference service. For example: (1) the direct sum of costs is minimized at a smaller value of $m$ if each sub-task cost $\mathcal{C}_{\text{LLM}}(\mathcal{L}_{\text{sys}} + m, \mathcal{L}_{\text{dec}}(m))$ grows with $m$ at a faster rate; (2) if $\mathcal{C}_{\text{LLM}}(\mathcal{L}_{\text{sys}} + m, \mathcal{L}_{\text{dec}}(m))$ grows with $m$ (sub-)linearly, then the end-to-end latency with parallelism degree $p$ is minimized at $m \asymp n/p$.

---

## 3.2 Answering Q2: error analysis

While error analysis is task-specific and has to be conducted case by case, we will see that general principles can be derived for the pattern of parallel decomposition, to be summarized in Insight 2.

### 3.2.1 Counting

Assume for notational convenience that $m_i = m$ for all $i \in [k]$, and $k = n/m$ is an integer. We denote the ground-truth count for the complete string as $y^\star$, and the ground-truth count for the $i$-th sub-string as $y_i^\star$.

If $\mathcal{E}$ represent the absolute counting error, then the overall error is bounded by the sum of sub-task errors:

$$\mathcal{E}(y) := |y - y^\star| = \left| \sum_{i \in [k]} (y_i - y_i^\star) \right| \leq \sum_{i \in [k]} |y_i - y_i^\star| = \sum_{i \in [k]} \mathcal{E}(y_i).$$

If $\mathcal{E}$ represent the normalized counting error instead, then

$$\mathcal{E}(y) := \frac{|y - y^\star|}{n} \leq \frac{1}{n} \sum_{i \in [k]} |y_i - y_i^\star| = \frac{1}{k} \sum_{i \in [k]} \frac{|y_i - y_i^\star|}{m} = \frac{1}{k} \sum_{i \in [k]} \mathcal{E}(y_i),$$

i.e., the overall error is bounded by the average of sub-task errors; since a smaller value of $m$ makes each sub-task easier, it is reasonable to expect that the overall error $\mathcal{E}(y)$ will also become smaller as $m$ decreases.

### 3.2.2 Sorting

Compared to counting, there are more diverse phenomena in the sorting task in terms of error metrics. In particular, there exist various failure modes in sorting with an LLM-based algorithm: (1) the output list might not be monotone; (2) the length of the output list might be larger or smaller than that of the input list; (3) the output list might contain numbers that do not match exactly those of the input list.

To accommodate these failure modes, we may define the following error metrics for sorting a list, where $\boldsymbol{y}$ denotes the solution returned by the algorithm and $\boldsymbol{y}^\star$ denotes the ground-truth solution:

- *Exact-match error*: $\mathcal{E} = 0$ if $\boldsymbol{y}$ matches $\boldsymbol{y}^\star$ exactly, and $\mathcal{E} = 1$ otherwise;

- *Non-monotonicity error*: $\mathcal{E} = \sum_{i \in [n-1]} \max\{y_i - y_{i+1}, 0\}$, which is zero if and only if $\boldsymbol{y}$ is sorted;

- *Length-mismatch error*: $\mathcal{E} = \frac{1}{n} |\text{len}(\boldsymbol{y}) - \text{len}(\boldsymbol{y}^\star)| = \frac{1}{n} |\text{len}(\boldsymbol{y}) - n|$;

- *Fuzzy $\ell_\infty$ and fuzzy normalized $\ell_1$ errors*: we first convert, via simple extending or truncating, the output solution $\boldsymbol{y}$ to a version $\widehat{\boldsymbol{y}}$ that matches the length $n$ of the input list, and then calculate the fuzzy $\ell_\infty$ error as $\mathcal{E} = \|\widehat{\boldsymbol{y}} - \boldsymbol{y}^\star\|_\infty = \max_{i \in [n]} |\widehat{y}_i - y_i^\star|$, or the fuzzy normalized $\ell_1$ error as $\mathcal{E} = \frac{1}{n}\|\widehat{\boldsymbol{y}} - \boldsymbol{y}^\star\|_1 = \frac{1}{n}\sum_{i \in [n]} |\widehat{y}_i - y_i^\star|$.

Note that the same error metrics can be similarly defined for each parallel sub-task in Step 2 of the LLM-based algorithm, and it is reasonable to expect that they become smaller as the sub-task size $m$ decreases. On the other hand, analyzing the errors of the overall algorithm after the final merging step can be more complicated. As an example, focusing on the third failure mode and the $\ell_\infty$ error metric, we have the following theoretical guarantee (whose proof is deferred to Appendix A.3.4):

**Proposition 1.** *Assume that for each $i \in [k]$, the solution $\boldsymbol{y}_i$ returned by one LLM call for the $i$-th sub-task is monotone, matches the length of the corresponding input $\boldsymbol{x}_i$, and has an $\ell_\infty$ error $\mathcal{E}_i$. Then the $\ell_\infty$ error of the final solution $\boldsymbol{y}$ is upper bounded by $\mathcal{E} \leq \max\{\mathcal{E}_1, \ldots, \mathcal{E}_k\}$.*

Consequently, if each sub-task error is monotone in $m$, then this error bound is also monotone in $m$.

### 3.2.3 Retrieval

For the retrieval (or needle-in-a-haystack (Kamradt, 2023)) task, we start by identifying two failure modes of each LLM call for retrieving the targeted information from a chunk of size $m$ in Step 2 of the algorithm:

1. The LLM might mistakenly return "I don't know" or an incorrect passcode when the chunk contains the specific needle that is necessary and sufficient for answering the question. In general, this failure mode will occur more frequently for larger values of $m$. Our early experiments with various LLMs confirmed that this failure mode begin to manifest when $m$ exceeds a certain LLM-specific threshold.

2. The LLM might mistakenly return a wrong answer when the needle is actually absent from the chunk (i.e., false positives). We observed empirically that some LLMs are more susceptible to this failure mode even when the value of $m$ is small, while others are less so.

Based on the above observations, we can hypothetically categorize LLMs into two types: *Type-A LLMs* are only prone to the first failure mode, while *Type-B LLMs* are prone to both.

It turns out that error analysis for the overall LLM-based algorithm depends critically on which type of LLM is used. If a Type-A LLM is used, a smaller value of $m$ means the first failure mode is less likely to occur in Step 2 of the algorithm, which implies higher accuracy for the final solution of the overall algorithm. On the other hand, analysis is more complicated if a Type-B LLM is used, since both failure modes can possibly occur in Step 2 of the algorithm: a larger value of $m$ means the first failure mode is more likely to occur, while a smaller value of $m$ implies a larger number of chunks $k \asymp n/m$, which can potentially increase the chance of error in the final step of the algorithm (i.e., majority voting), due to the frequent occurrence of the second failure mode in Step 2 of the algorithm. Consequently, the minimum error of the overall algorithm might be achieved by some intermediate value of $m$ that achieves a balance between the two failure modes. If $n$ is too large, then there might not exist a good value of $m$ that can achieve a low error, as either failure mode must occur with high probability.

*Remark* 3. We present an informal probabilistic analysis for the case of using a Type-B LLM. Given the sub-task size $m$, denote the probability of the first and second failure modes as $p_1(m)$ and $p_2(m)$ respectively. Then, the success rate of retrieval for the chunk containing the needle is $1 - p_1(m)$, while the expected number of false positives from the remaining chunks is approximately $k \times p_2(m) \asymp n \times p_2(m)/m$. One might opt for a relatively small value of $m$, which hopefully increases $1 - p_1(m)$ and hence mitigates the first failure mode. However, even if $p_2(m)/m$ is very small, say $10^{-3}$, the number of false positives can be large if the size $n$ of the original problem is large, causing errors in the solution returned by majority voting.

### 3.2.4 Summary

With the above analysis, we summarize our answer to Q2 as follows:

**Insight 2.** If (1) each sub-task error is monotonely increasing in the sub-task size $m$, and (2) the overall error is bounded by a smooth aggregation (e.g., mean or max) of sub-task errors, then the overall error can be reduced by finer-grained parallel task decomposition, i.e., smaller $m$. Otherwise, caution should be taken: there exist cases (e.g., retrieval with a Type-B LLM) where the final output of the overall algorithm can be sensitive to individual sub-task errors, and finer-grained parallel decomposition (accompanied by a larger number of sub-tasks) does not necessarily imply a smaller final error.

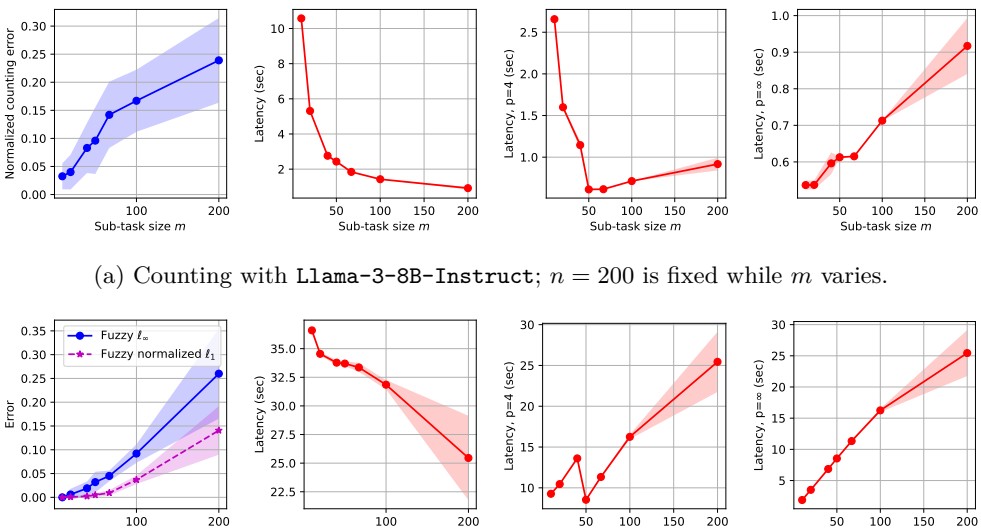

(a) Counting with `Llama-3-8B-Instruct`; $n = 200$ is fixed while $m$ varies.

(b) Sorting with `Llama-3-70B-Instruct`; $n = 200$ is fixed while $m$ varies. See Appendix A.3.3 for analysis and explanation of the zigzag part in the latency with $p = 4$.

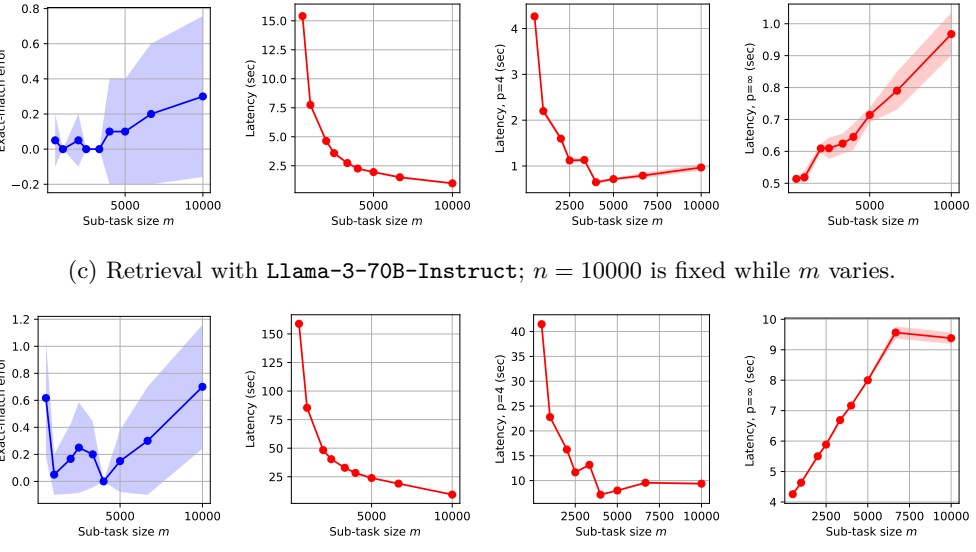

(c) Retrieval with `Llama-3-70B-Instruct`; $n = 10000$ is fixed while $m$ varies.

(d) Retrieval with `Llama-3-8B-Instruct`; $n = 10000$ is fixed while $m$ varies.

Figure 4: Empirical results for concrete examples of parallel decomposition. Each curve represents the mean and standard deviation of 10 independent trials. See Appendix A for further details and complete results for the case studies on parallel decomposition.

### 3.3 Empirical validation

The previous analysis has been validated empirically via experiments with Llama 3 models (Meta, 2024). A subset of empirical results are presented in Figure 4, with full implementation details and experimental results deferred to Appendix A. For each figure, we fix the problem size $n$ and use the sub-task size $m$ as the X-axis.

Figure 4 confirms that for all tasks, the end-to-end latency of the overall algorithm is monotonely decreasing in $m$ when the parallelism degree is $p = 1$ (i.e., no parallelism), non-monotone in $m$ with a finite parallelism degree $p = 4$, and monotonely increasing in $m$ when $p = \infty$, as predicted by Insight 1.

Figure 4 also shows that error metrics increase with $m$ for the counting and sorting tasks, as well as for retrieval with `Llama-3-70B-Instruct`; in contrast, errors are non-monotone in $m$ for the retrieval task with `Llama-3-8B-Instruct`, which is susceptible to false positives in retrieval (see Figure 13b in Appendix A.4.3) and thus regarded as a Type-B LLM (cf. Section 3.2.3).

## 4 Case studies and insights: more general patterns

In this section, we demonstrate the proposed analytical framework in action for more diverse patterns of LLM-based algorithms. Our exploration covers unified error analysis under the assumption of additive errors and bounded sensitivity (Section 4.1), two algorithms for iterative retrieval and reasoning that exhibit the pattern of hierarchical decomposition (Section 4.2), and an algorithm that conducts recursive task decomposition and constructs its computational graph dynamically at runtime (Section 4.3). The corresponding analysis and results are summarized in Insights 3, 4 and 5 respectively.

### 4.1 Unified error analysis for directed acyclic graphs

For generic LLM-based algorithms that are represented by a direct acyclic graphs (DAG) and satisfy the technical assumption below, our framework allows deriving error bounds for them in a unified manner:

**Assumption 1.** *We say that an LLM-based algorithm satisfies the assumption of* additive errors and bounded sensitivity, *if for each node $v$ with $k$ inputs $x_1, \ldots, x_k$ and a single output $y$, it holds that*

$$\mathcal{E}(y) \leq f_v\big(\mathcal{E}(x_1), \mathcal{E}(x_2), \ldots, \mathcal{E}(x_k)\big) := \mathcal{E}_v + S \times \sum_{i \in [k]} \mathcal{E}(x_i)$$

*for some node-specific additive error $\mathcal{E}_v$ and finite sensitivity parameter $S \geq 0$.*

**Proposition 2.** *Consider an LLM-based algorithm, represented by a DAG, that satisfies Assumption 1. Then the error of the output $y(v)$ of any node $v$, including the one that generates the final output of the overall algorithm, is bounded by*

$$\mathcal{E}\big(y(v)\big) \leq \sum_{w \in \mathcal{V}} \sum_{path \in \mathcal{P}(w \to v)} S^{|path|} \times \mathcal{E}_w. \tag{6}$$

*Here, $\mathcal{V}$ denotes the node set of the DAG, $|path|$ denotes the length of a path on the DAG, and $\mathcal{P}(w \to v)$ represents the set of paths from node $w$ to node $v$ if $w \neq v$, or a singleton set containing one hypothetical path of length 0 if $w = v$. For the special case of $S = 1$, this upper bound can be simplified as $\mathcal{E}(y(v)) \leq \sum_{w \in \mathcal{V}} |\mathcal{P}(w \to v)| \times \mathcal{E}_w$, where $|\mathcal{P}(w \to v)|$ denotes the number of paths from $w$ to $v$.*

This result, whose proof can be found in Appendix B, characterizes precisely how individual sub-task errors propagate through the graph to the final output of the overall algorithm, depending on the number and lengths of paths between graph nodes and the value of the sensitivity parameter $S$:

---

**Insight 3.** Under the assumption of additive errors and bounded sensitivity, Proposition 2 indicates that: (1) the final error of the overall algorithm is a weighted average of all node-specific additive errors $\{\mathcal{E}_w\}$; (2) the impact of each node-specific error $\mathcal{E}_w$ grows (resp. shrinks) exponentially along each path to the final output if $S > 1$ (resp. $S < 1$), and also grows proportionally with the total number of paths.

---

**Justifications for Assumption 1.** In some cases, Assumption 1 can be rigorously proved for a non-LLM node. For example, considering the aggregation node in the counting algorithm, our analysis in Section 3.2.1 implies that Assumption 1 holds true with node-specific additive error $\mathcal{E}_v = 0$, and sensitivity parameter $S = 1$ for the absolute counting error, or $S = 1/k$ for the normalized counting error. As another example, considering the aggregation node in the sorting algorithm, Proposition 1 indicates that its output $\ell_\infty$ error is bounded by $\mathcal{E} \leq \max\{\mathcal{E}_1, \ldots, \mathcal{E}_k\} \leq \mathcal{E}_1 + \cdots + \mathcal{E}_k$, which means Assumption 1 holds with $\mathcal{E}_v = 0$ and $S = 1$.

For an LLM node, "additive error and bounded sensitivity" is an assumption that we impose on the *stability* of an LLM's behavior when solving a *sufficiently simple* (with respect to its capability) sub-task. One motivating example comes from our case study to be presented in Section 4.3, which involves many LLM calls that are asked to calculate a 1-Lipschitz function (such as sum or max) of multiple input variables. We have observed empirically that the LLM, with CoT prompting, behaves with sufficient stability and makes small errors only occasionally. It is this property that ensures the final error of the overall algorithm stays low and grows slowly (instead of blowing up) with the task complexity and size of computational graph.

Needless to say, there exist practical situations where Assumption 1 does not hold (e.g., when the LLM's behavior is sensitive to input errors), in which case Proposition 2 is not applicable.

## 4.2 Iterative retrieval and reasoning

We consider a task, as an extension of the retrieval task in Section 3.2.3, that requires multi-hop reasoning over multiple clues ("needles") embedded separately in a long piece of text ("haystack"). For example, the targeted question might be "what is the numeric value of A", while the needles are "A = B", "B = C", and "C = 100". A practical approach for such a task is iterative retrieval and reasoning (Creswell et al., 2023; Xiong et al., 2024; Qwen-Team, 2024; Yue et al., 2024): at each iteration, the algorithm performs retrieval based on the targeted question and the collected information from previous iterations, followed by reasoning over the retrieved information and deciding whether the targeted question can be answered already or requires further retrieval and reasoning. Within each iteration, retrieval can be done along with chunking, which leads to a pattern of hierarchical decomposition. Two variants of this approach are visualized in Figure 5: the "cyclic" version conducts retrieval over multiple chunks sequentially within each iteration, while the "parallel" version conducts retrieval in parallel. When adopting such an algorithm, two critical design choices need to be made:

1. For the "reasoning" nodes, one might prompt the LLM to "answer directly" for efficiency, or "think step by step" to potentially improve accuracy. Which option is better? How much cost overhead will be caused by chain-of-thought prompting in exchange for accuracy improvement?

2. For the task of retrieval from multiple chunks at each iteration, which option would be more efficient, the "cyclic" or "parallel" one?

Our answers, derived with the proposed analytical framework, can be summarized as follows:

> **Insight 4.** (1) Using chain-of-thought prompting for the reasoning nodes (which play a critical role in deciding the final output) will significantly boost accuracy, while incuring only a minor, *additive* cost overhead to the overall algorithm. See Eq. (9) and Eq. (10) in Appendix C.3.2 for the quantitative version of this statement. (2) The "cyclic" version requires fewer iterations, while the "parallel" version can benefit from parallelism. As a result, in terms of the end-to-end latency, the "cyclic" version is likely to have advantage when the parallelism degree is small, while the reverse is true otherwise.

These insights have been validated empirically via experiments with `Qwen-2-72B-Instruct` (Yang et al., 2024). For example, Figure 6a shows that, across a broad range of degree $g$ (a measure of task complexity), prompting the LLM to calculate step by step for the reasoning nodes significantly reduces the error of the final output, while incuring negligible or no overhead in the total number of prefilling tokens, and minor overhead in the total number of decoding tokens. Moreover, Figure 6b compares the end-to-end latencies of two options for retrieval: "cyclic" achieves lower latencies across a broad range of width $w$ (another measure of task complexity) when there is no parallelism among LLM calls (i.e., $p = 1$), wheras "parallel" can benefit from an increased value of parallelism degree $p$ and gain advantages over "cyclic".

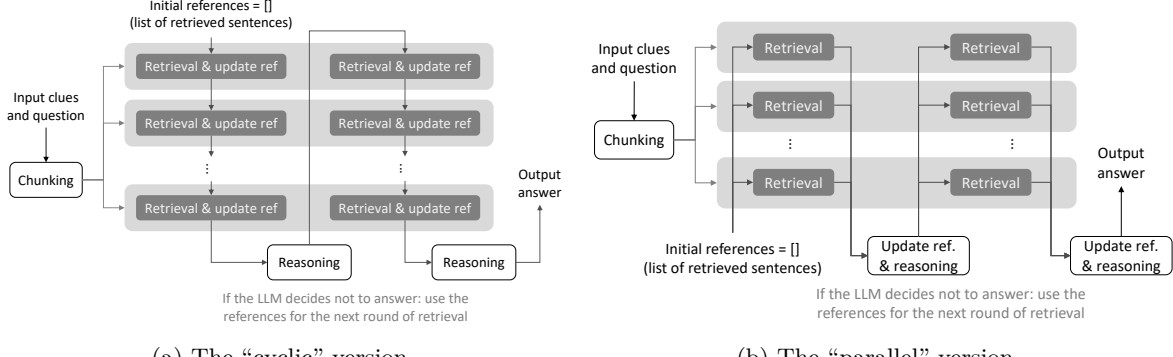

(a) The "cyclic" version.        (b) The "parallel" version.

Figure 5: Two algorithms for iterative retrieval and reasoning. The number of iterations is assumed to be 2 here; in reality, this value is determined adaptively by the algorithm itself at runtime. For clarity, some LLM or non-LLM nodes (cf. Figure 2) are merged into one, and an arrow from the "chunking" node to a shaded block means that each "retrieval" node within the block takes the corresponding chunk as input.

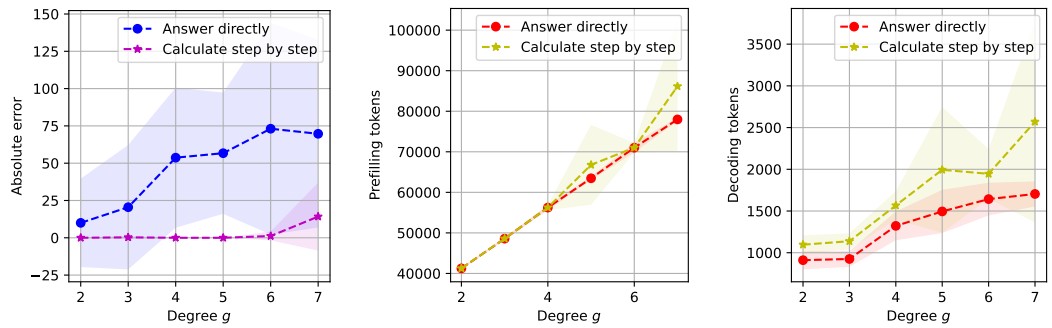

(a) Comparing two prompting methods ("answer directly" versus "step by step") for the reasoning nodes.

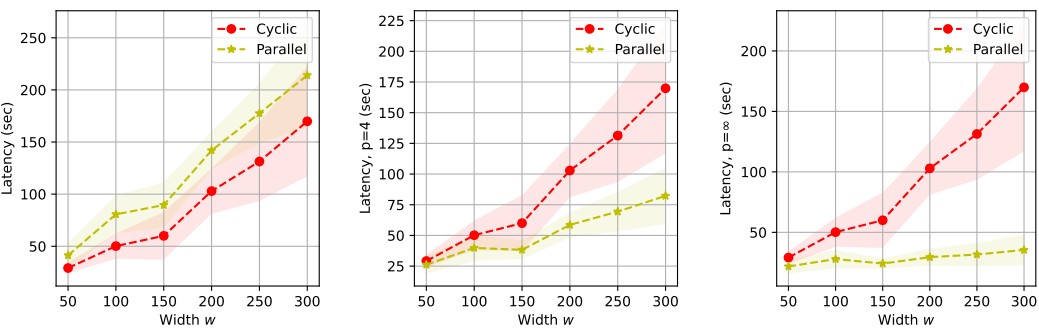

(b) Comparing two versions ("cyclic" versus "parallel") of iterative retrieval and reasoning.

Figure 6: Empirical results for iterative retrieval and reasoning. The X-axis of each figure denotes a measure of task complexity. See Appendix C for further details and complete results of this case study.

We refer interested readers to Appendix C for full details and results of this case study.

### 4.3 Recursive task decomposition

For the last case study, we study LLM-based algorithms with *recursive task decomposition*. A recursive LLM-based algorithm starts from the original task of concern and recursively generates more intermediate sub-tasks, so that each sub-task can be solved by aggregating the solutions to its children tasks, while being

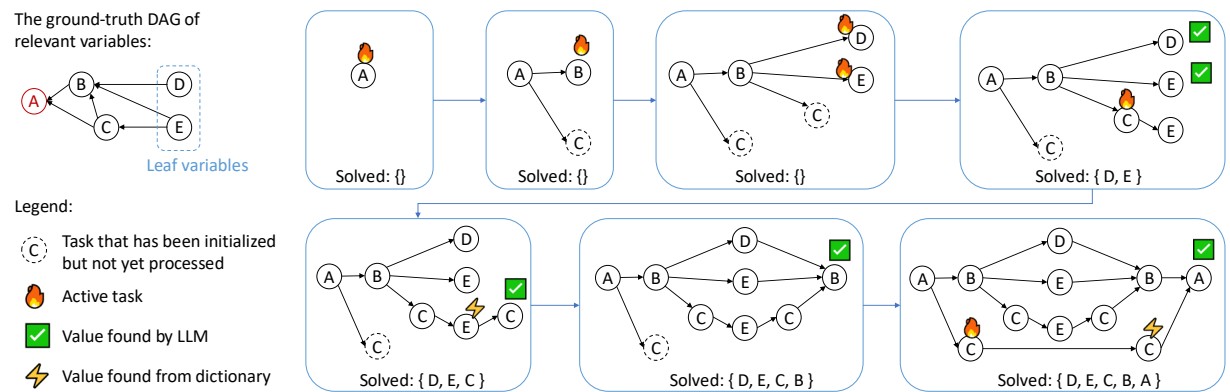

Figure 7: The computational graph of a recursive LLM-based algorithm is constructed dynamically, while a dictionary of solved tasks is maintained to avoid solving the same intermediate sub-task repetitively.

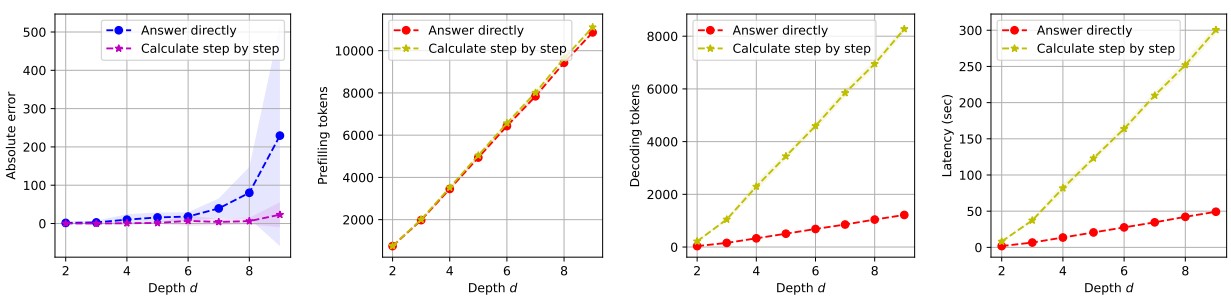

Figure 8: Empirical results for recursive task decomposition. The X-axis of each figure denotes a measure of task complexity. See Appendix D for further details and complete results of this case study.

unaware of other sub-tasks involved in the overall algorithm. In particular, solving and/or decomposing each sub-task can be achieved by LLM calls, while the outline of recursive task decomposition remains symbolic. Another feature of a recursive LLM-based algorithm is that its computational graph is not pre-specified or static, but rather constructed dynamically at runtime. Such algorithms have been widely adopted in prior works (Kazemi et al., 2023; Schlag et al., 2023; Prasad et al., 2024; Schroeder et al., 2024; Lee & Kim, 2023; Khot et al., 2023; Zhu et al., 2024).

For concreteness, consider a task that requires finding the value of a specific variable connected to a large number of other variables. Figure 7 shows an example, where the targeted variable A depends on B and C, and variable B further depends on C, D and E, and so on. The LLM-based algorithm need to decide adaptively which variables to query information about, and to execute the necessary calculation along the way. A complete description of the recursive algorithm for this task can be found in Appendix D.2. Despite how different such an algorithm might seem from the previously investigated ones, it can still be covered by our proposed analytical framework:

> **Insight 5.** (1) The framework proposed in Section 2 is applicable to a recursive LLM-based algorithm, by *unrolling* it into a directed acyclic graph, as illustrated in Figure 7. (2) For the concrete example, prompting the LLM to conduct arithmetic calculation step by step will significantly boost the accuracy of the overall algorithm, but also increase the cost by a *multiplicative* factor. See Eq. (11) in Appendix D for the quantitative version of this statement.

The second insight has been validated empirically via experiments with `Qwen-2-72B-Instruct` (Yang et al., 2024). For example, Figure 8 confirms the accuracy gain by prompting the LLM to calculate step by step, and shows that cost metrics grow linearly with the depth $d$ (a measure of task complexity); furthermore,

step-by-step prompting has negligible impacts on the total number of prefilling tokens, but increases the total number of decoding tokens and end-to-end latency multiplicatively.

We refer interested readers to Appendix D for full details and results of this case study.

## 5    Related works

**LLM-based algorithms.**    The concept of LLM-based algorithms is fairly broad and general, ranging from a combination of one or multiple LLM calls with some prompt engineering (Yao et al., 2023a; Besta et al., 2023; 2024; Lei et al., 2023; Saha et al., 2023; Zhou et al., 2023; Wang et al., 2023; Prasad et al., 2024; Chang et al., 2024), to LLM-powered agent systems (Mialon et al., 2023; Wu et al., 2023; Hong et al., 2024; Gao et al., 2024a; Shen et al., 2023; Zhuge et al., 2024; Pezeshkpour et al., 2024; Xi et al., 2023; Qian et al., 2024; Kapoor et al., 2024; Sumers et al., 2024; Anthropic, 2024) and compound AI systems (Zaharia et al., 2024; Chen et al., 2024) that augment LLMs with additional abilities like tool use and long-term memory, and to the emerging paradigm of LLM programming (Schlag et al., 2023; Khot et al., 2023; Khattab et al., 2024; Zheng et al., 2024; Kambhampati et al., 2024). Some elements of the framework proposed in Section 2, such as the task decomposition principle, the computational graph representation, evaluation of and comparison between LLM-based algorithms with accuracy and efficiency taken into account simultaneously (Wang et al., 2024; Kapoor et al., 2024), etc., have already appeared in one way or another in these prior works. It is primarily our *unified, systematic and formal* investigation into the design of generic LLM-based algorithms, and the analysis of their accuracy and efficiency (akin to analysis of traditional algorithms), that distinguish the current work from this vast literature. It is worth noting that (Wu et al., 2025) presents formal analysis for chain-of-thought reasoning, and (Snell et al., 2024) presents formal analysis for sequential and parallel sampling. Both works are similar in spirit to the current work, although the problem formulations and methodologies are mostly orthogonal and complementary.

**LLM test-time scaling.**    Recent works have started to investigate, analytically or empirically, the scaling laws of LLM test-time computation. One line of research is about the scaling properties of repeated sampling, e.g., randomly sampling multiple generations for the same prompt and then aggregating them for the final output (Chen et al., 2024; Brown et al., 2024; Snell et al., 2024; Wu et al., 2024; Chen et al., 2025), which might be regarded as a stochastic variant of parallel decomposition investigated in Section 3. Our work is orthogonal and complementary to this line, as our analysis is deterministic and targeted at generic patterns of LLM-based algorithms. One potential direction for expanding the analytical framework proposed in this work would be to augment it with stochastic decoding and repeated sampling, along with relevant theoretical results. Another line of works is about improving the output quality of an LLM call by generating more tokens autoregressively, e.g., via chain-of-thought (Wei et al., 2022b; Kojima et al., 2022) reasoning. This idea has been investigated theoretically (Feng et al., 2023; Merrill & Sabharwal, 2024; Li et al., 2024b), and further popularized recently by OpenAI-o1 (OpenAI, 2024a), DeepSeek-R1 (DeepSeek-AI et al., 2025), among many others. As this approach of scaling up the test-time computation of *individual* LLM calls is becoming more widely adopted, it is important to understand, analytically and quantitatively, how it will impact the *overall* accuracy and efficiency when such LLM calls are embedded within an LLM-based algorithm. Our work can be useful in achieving this, as illustrated in the case studies in Sections 4.2 and 4.3, where the impacts of prompting LLM calls to "think step by step" versus "answer directly" can be characterized analytically with our proposed framework.

**Connections to non-LLM literature.**    Our work has drawn inspiration from the area of *learning-augmented algorithms* (Mitzenmacher & Vassilvitskii, 2022; Lindermayr & Megow, 2022). One standard paradigm of this area is to consider a specific task (e.g., sorting (Bai & Coester, 2023) or clustering (Ergun et al., 2022)), assume access to a black-box machine learning model that satisfies certain properties (e.g., what additional computation or information it can offer), propose a novel algorithm that leverages this ML model, provide theoretical guarantees for its accuracy and efficiency, and show the improvements over traditional, purely symbolic algorithms. Our work is similar in spirit to this line of research, but also substantially different, in that our framework is targeted at generic tasks and LLM-based algorithms, with assumptions on the capabilities, limitations and inference service of LLMs (regarded as general-purpose problem solvers)

that are quite different from typical assumptions for task-specific ML models in the literature of learning-augmented algorithms.

Our work is also connected to the literature of distributed computing (Lynch, 1996) (in particular, the MapReduce model (Dean & Ghemawat, 2008; Karloff et al., 2010) that the parallel decomposition pattern in Section 3 closely resembles), as well as classical / non-LLM algorithm design and analysis (Cormen et al., 2022; Roughgarden, 2021). When positioned in this context, one major contribution of our work is to (1) identify the relevance of foundational elements in computer science — task decomposition, computational graphs, MapReduce, etc. — to the LLM context, and (2) integrate LLM-based algorithms with these elements, other theoretical tools, and symbolic programs, via appropriate abstractions and characterization for black-box LLMs. This is the key to facilitating formal analysis for the accuracy and efficiency of LLM-based algorithms. The unique and diverse features of LLMs — and their implications — distinguish our work from the classical / non-LLM literature.

## 6 Conclusion

This work has introduced an analytical framework for studying the design and analysis of LLM-based algorithms. With the task decomposition principle, the computational graph representation, and other key abstractions tailored to LLMs, we are able to provide formal analysis for the accuracy and efficiency of generic LLM-based algorithms, as demonstrated through a diverse range of case studies.

One limitation of this work is the lack of empirical studies in more realistic settings. Although we have presented case studies and empirical validation for various algorithms and synthetic settings (ranging from the most basic setting — parallel decomposition — to more complex ones that resemble the settings in prior empirical works), there is obviously a gap between them and the most complex agentic workflows in real-world scenarios. Nonetheless, we believe that the results in this work can have broader implications in more general scenarios than the synthetic ones. For example, Insights 1 and 2 can, at the minimum, warn practitioners against the common misconception that "finer-grained task decomposition implies higher accuracy / higher total cost for the overall algorithm", beyond the parallel decomposition setting where these insights were originally derived. Moreover, we anticipate that our analysis for simple patterns could be used as building blocks for analysis of more complex algorithms since, as highlighted in Anthropic (2024), the most successful LLM agents in industry are often built with "simple, composable patterns".

We hope that our work will inspire future work on enhancing the analytical framework (e.g., by incorporating certain properties and challenges of realistic LLM usage that we might have abstracted away in this work), or leveraging it to derive more novel and generalizable insights that can assist the best practice of real-world LLM applications.

### Acknowledgments

The authors would like to thank the reviewers and Action Editor for their constructive feedback that helps improve this work. We also thank Yuchang Sun for helpful discussion, and the AgentScope team (especially Xuchen Pan, Dawei Gao and Weirui Kuang) for infrastructure support.

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

# A    Supplementary materials for Section 3

This appendix complements Section 3, containing details of experiment settings, additional results for the concrete examples presented in Section 3, and additional examples of parallel decomposition.

## A.1    Experiment settings

We use the following LLMs, which cover a wide range of LLM characteristics and inference service:

- `Llama-3-8B-Instruct` (Meta, 2024), supported by ollama (ollama, 2023) and running on a Macbook Pro with a M2 Pro chip and 16GB memory;

- `Llama-3-70B-Instruct` (Meta, 2024) and `Qwen-2-72B-Instruct` (Yang et al., 2024), supported by vLLM (Kwon et al., 2023) and running on a server with 4 Nvidia A100-80G GPUs;

- `GPT-4-Turbo` (OpenAI, 2024), accessed via API queries.

All of these LLMs are chat models. Each LLM call involved in our algorithms is prompted in a chat format, according to the sub-task that it is responsible for. We use greedy decoding in all experiments.

Below are a few more details about our experiments. (1) For ideal parallelism of LLM calls, we consider parallelism degree $p = 4$ and $p = \infty$. Latencies in the presence of parallelism are *simulated* according to Eq. (3). (2) In all experiments, the number of tokens for a piece of text is *estimated* using the same tokenizer, namely the `cl100k_base` encoding of the `tiktoken` package[3]. This simplification has no effect on the major conclusions from our experiment results. (3) Our experiment results include curves of some error metric (in blue) or cost metric (in red) versus the problem size $n$ or sub-task size $m$. For each curve, we plot the mean and standard deviation of measured metrics from multiple independent trials, i.e., multiple randomly generated task instances.

## A.2    Example: counting

### A.2.1    Algorithm details

For each LLM call within the counting algorithm presented in Table 1, we prompt the LLM to generate its answer directly without intermediate reasoning steps[4]. Thus it is reasonable to assume, given the LLM's capability of instruction following, that the text generated by each LLM call has length $\mathcal{L}_{\mathsf{dec}} = O(1)$ and can be parsed into a count number.

### A.2.2    Cost analysis

Our analysis of cost metrics follows Section 3.1. More specifically, we have $\mathcal{L}_{\mathsf{dec}} = O(1)$ by assumption, and $\mathcal{C}(\text{aggregation}) = 0$ since the final step of the algorithm is done by a non-LLM node.

- Considering the total prefilling length and decoding length as cost metrics, one has

$$\mathcal{C}(\text{prefilling}) \lesssim k \times (\mathcal{L}_{\mathsf{sys}} + m) = \frac{n}{m} \times (\mathcal{L}_{\mathsf{sys}} + m) = n \times (\frac{\mathcal{L}_{\mathsf{sys}}}{m} + 1),$$

$$\mathcal{C}(\text{decoding}) \lesssim k \times 1 = k = \frac{n}{m},$$

both of which are monotonely decreasing in $m$.

- More generally, the sum of costs of all LLM calls is

$$\mathcal{C} = \mathcal{C}(\text{sub-tasks}) \leq k \times \Big(\mathcal{C}_{\mathsf{pre}}(\mathcal{L}_{\mathsf{sys}} + m) + \mathcal{C}_{\mathsf{dec}}(\mathcal{L}_{\mathsf{sys}} + m, 1)\Big)$$

---

[3]`https://github.com/openai/openai-cookbook/blob/main/examples/How_to_count_tokens_with_tiktoken.ipynb`

[4]On the other hand, step-by-step counting can yield a more accurate answer, but also a significantly higher cost, with $\mathcal{L}_{\mathsf{dec}} = O(m)$ rather than $O(1)$.

$$= n \times \frac{\mathcal{C}_{\mathsf{pre}}(\mathcal{L}_{\mathsf{sys}} + m) + \mathcal{C}_{\mathsf{dec}}(\mathcal{L}_{\mathsf{sys}} + m, 1)}{m}.$$

The optimal choice of $m$ under various conditions has been discussed in Section 3.1.

- For the end-to-end latency with parallelism degree $p$, we have

$$\mathcal{C} = \mathcal{C}(\text{sub-tasks}) \leq \left\lceil \frac{k}{p} \right\rceil \times \left( \mathcal{C}_{\mathsf{pre}}(\mathcal{L}_{\mathsf{sys}} + m) + \mathcal{C}_{\mathsf{dec}}(\mathcal{L}_{\mathsf{sys}} + m, 1) \right)$$

$$= \left\lceil \frac{n}{p \cdot m} \right\rceil \times \left( \mathcal{C}_{\mathsf{pre}}(\mathcal{L}_{\mathsf{sys}} + m) + \mathcal{C}_{\mathsf{dec}}(\mathcal{L}_{\mathsf{sys}} + m, 1) \right).$$

Under the assumptions explained in Section 3.1, the minimum is achieved around $k = p$ and $m = n/p$.

### A.2.3 Experiments

We validate our analysis with numerical experiments. For each problem instance, the input string is generated by randomly sampling $n$ characters from the union of digits, lower-case letters and upper-case letters.

**Results with `Llama-3-8B-Instruct`.** The results are illustrated in Figure 9.

In Figure 9a, we vary the problem size $n$ and set $m = n$ in our algorithm, in which case the algorithm becomes equivalent to a single LLM call. Unsurprising, error metrics in this counting task monotonely increase with $n$. The number of prefilling tokens increases linearly with $n$, while the number of decoding tokens is insensitive to $n$, since we prompt the LLM to output its answer directly without intermediate reasoning steps. The latency of one LLM call also increases linearly with $n$, which is likely due to the relatively small sequence lengths and the compute-bound nature of LLM inference by a `Llama-3-8B-Instruct` model running on a CPU.

In Figure 9b, we fix $n = 200$ and vary the hyperparameter $m$, i.e., the size of each sub-task. It is confirmed that decomposing the original task into smaller parallel sub-tasks improves the accuracy of the overall algorithm, while incuring higher cost metrics, except for the latency with infinite parallelism (which is monotonely increasing in $m$) and the latency with parallelism degree $p = 4$ (which achieves the minimum at $m = n/p = 50$, as was predicted by our previous analysis).

**Results with `GPT-4-Turbo`.** Figure 10 demonstrates the empirical results for the same experiments but with `GPT-4-Turbo`. We observe from Figure 10a that the latency of one LLM call is insensitive to the input problem size $n$, which is likely because LLM inference of `GPT-4-Turbo` is memory-bound for the range of sequence lengths considered in our experiments. One potential implication is that the end-to-end latency with infinite parallelism $p = \infty$ might slightly increase for smaller $m$ and hence larger $k$, due to the random variation of latencies in reality; this is indeed what we observe from Figure 10b. Other than that, the results in Figure 10 are similar to those in Figure 9.

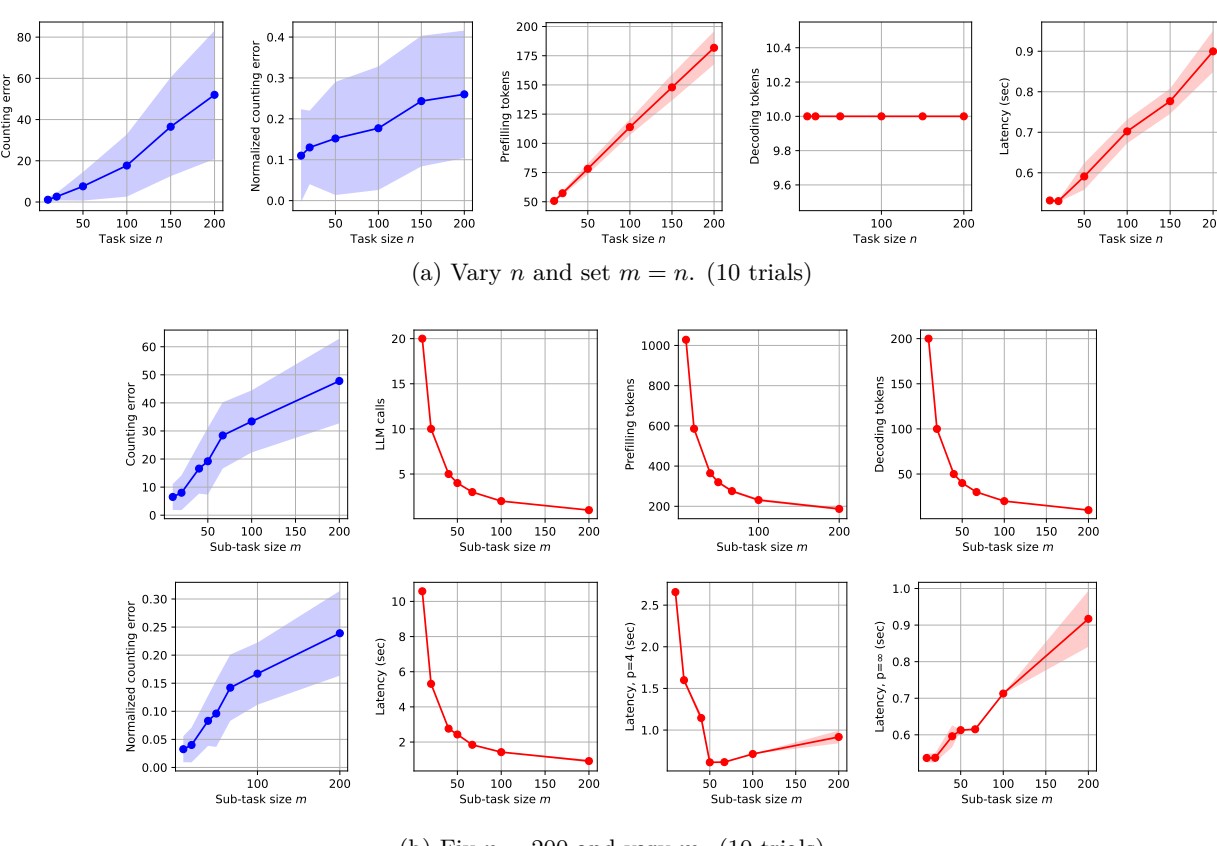

(a) Vary $n$ and set $m = n$. (10 trials)

(b) Fix $n = 200$ and vary $m$. (10 trials)

Figure 9: Empirical results for counting with `Llama-3-8B-Instruct`.

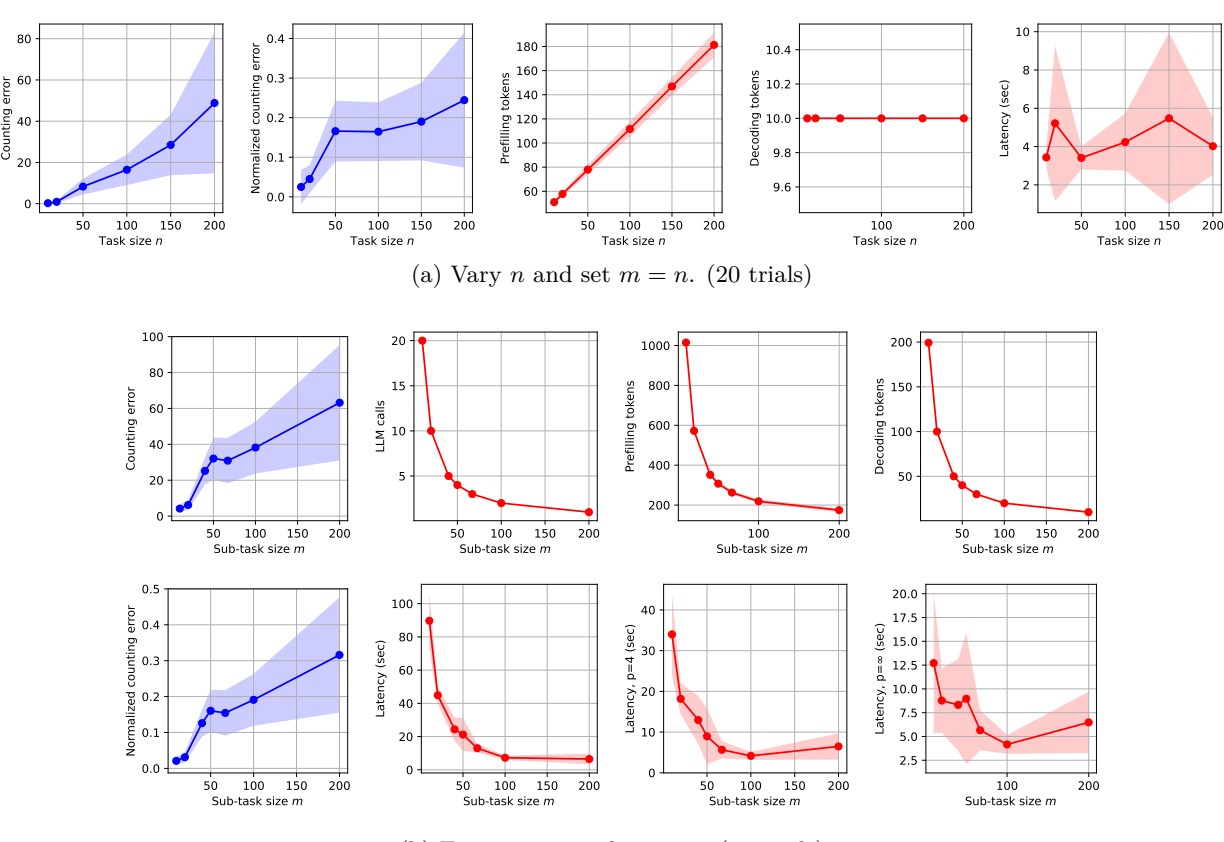

(a) Vary $n$ and set $m = n$. (20 trials)

(b) Fix $n = 200$ and vary $m$. (10 trials)

Figure 10: Empirical results for counting with `GPT-4-Turbo`.

### A.3 Example: sorting

#### A.3.1 Algorithm details

We note two details about the sorting algorithm presented in Table 1.

- To ensure efficiency and stability, for each LLM call, we prompt the LLM to generate the sorted list directly, without intermediate reasoning steps. It has been verified empirically that the LLMs considered in our experiments can follow such instructions, and generate text of length $\mathcal{L}_{\mathsf{dec}}(m) = O(m)$ that can be easily parsed into a list.

- Step 3 of the algorithm relies on a classical symbolic algorithm for merging two sorted lists, which maintains two moving pointers, one for each list, and chooses each entry of the merged list by comparing the values corresponding to the two pointers. Merging multiple sorted lists can be done by merging one pair of lists at a time, in an incremental or hierarchical manner. Python code for these procedures can be found in Listing 1 at the end of this subsection. Although they are designed under the assumption that the input lists are sorted, they can also be applied to input lists that are not fully sorted, which can possibly happen within the LLM-based algorithm since the input lists in Step 3 are generated by LLM calls in Step 2.

#### A.3.2 Cost analysis

Cost analysis of for the LLM-based sorting algorithm follows Section 3.1, and is similar to that for counting. One major difference is that $\mathcal{L}_{\mathsf{dec}} = O(m)$ rather than $O(1)$ for each sub-task.

- Considering the total prefilling length and decoding length as cost metrics, one has

$$\mathcal{C}(\text{prefilling}) \lesssim k \times (\mathcal{L}_{\mathsf{sys}} + m) = \frac{n}{m} \times (\mathcal{L}_{\mathsf{sys}} + m) = n \times (\frac{\mathcal{L}_{\mathsf{sys}}}{m} + 1),$$
$$\mathcal{C}(\text{decoding}) \lesssim k \times m = n.$$

The former is decreasing in $m$, while the latter is insensitive to $m$.

- More generally, the sum of costs of all LLM calls is

$$\mathcal{C} = \mathcal{C}(\text{sub-tasks}) \leq k \times \Big(\mathcal{C}_{\mathsf{pre}}(\mathcal{L}_{\mathsf{sys}} + m) + \mathcal{C}_{\mathsf{dec}}(\mathcal{L}_{\mathsf{sys}} + m, m)\Big)$$
$$= n \times \frac{\mathcal{C}_{\mathsf{pre}}(\mathcal{L}_{\mathsf{sys}} + m) + \mathcal{C}_{\mathsf{dec}}(\mathcal{L}_{\mathsf{sys}} + m, m)}{m}.$$

- For the end-to-end latency with parallelism degree $p$, we have

$$\mathcal{C} = \mathcal{C}(\text{sub-tasks}) \leq \Big\lceil \frac{k}{p} \Big\rceil \times \Big(\mathcal{C}_{\mathsf{pre}}(\mathcal{L}_{\mathsf{sys}} + m) + \mathcal{C}_{\mathsf{dec}}(\mathcal{L}_{\mathsf{sys}} + m, m)\Big)$$
$$= \Big\lceil \frac{n}{p \cdot m} \Big\rceil \times \Big(\mathcal{C}_{\mathsf{pre}}(\mathcal{L}_{\mathsf{sys}} + m) + \mathcal{C}_{\mathsf{dec}}(\mathcal{L}_{\mathsf{sys}} + m, m)\Big).$$

#### A.3.3 Experiments

In our experiments, the input list of each problem instance is generated by randomly sampling entries of the list from the uniform distribution over the interval $[0, 1]$ and rounding each of them to two decimals.

**Results with `Llama-3-70B-Instruct`.** The results are illustrated in Figure 11.

In Figure 11a, we vary the problem size $n$ and set $m = n$ in the LLM-based algorithm, in which case the algorithm becomes equivalent to a single LLM call. We make the following observations:

- Unsurprisingly, all error metrics in this task monotonely increase with $n$.

- While the LLM might output a list that deviates from the ground-truth solution, it is at least good at ensuring that the output list itself is sorted or has a very small non-monotonicity error.

- The prefilling length grows linearly with $n$, while the growth of the decoding length and end-to-end latency slows down slightly for large values of $n$, which is mainly because the LLM is prone to returning a list that is shorter than the input list when $n$ is large, as reflected in the length-mismatch error curve.

In Figure 11b, we fix $n = 200$ and vary the sub-task size $m$. It is confirmed that decomposing the original task into smaller parallel sub-tasks implies lower error metrics achieved by the overall algorithm, while increasing certain cost metrics. Two specific observations:

- The total number of decoding tokens decreases with $m$ at a rate that matches the length-mismatch error curve. This does not contradict our previous analysis, which predicts an *upper bound* that is insensitive to the value of $m$.

- Regarding the end-to-end latency with parallelism degree $p = 4$, the zigzag part of the curve might seems curious. In fact, a fine-grained analysis can well explain this phenomenon. If we approximate the latency for one LLM call solving a sub-task of size $m$ by $O(m)$, then the end-to-end latency of the overall algorithm is approximately $O(m \times \lceil k/p \rceil)$. The numbers calculated in Table 3 for the concrete setting of this experiment match the empirical results and explain the zigzag part.

**Results with `GPT-4-Turbo`.** Figure 12 demonstrates the empirical results for the same experiments but with a `GPT-4-Turbo` model. Similar observations can be made from these results, except that latencies exhibit higher variance due to the inference service of `GPT-4-Turbo`.

Table 3: Fine-grained analysis for the latency with parallelism degree $p = 4$ in Figure 11b, where $n = 200$.

| Sub-task size $m$ | 10 | 20 | 40 | 50 | 67 | 100 | 200 |
|---|---|---|---|---|---|---|---|
| Number of sub-tasks $k = \lceil n/m \rceil$ | 20 | 10 | 5 | 4 | 3 | 2 | 1 |
| Sequential depth $d = \lceil k/p \rceil$ | 5 | 3 | 2 | 1 | 1 | 1 | 1 |
| Predicted latency $\asymp d \times m$ | 50 | 60 | 80 | 50 | 67 | 100 | 200 |

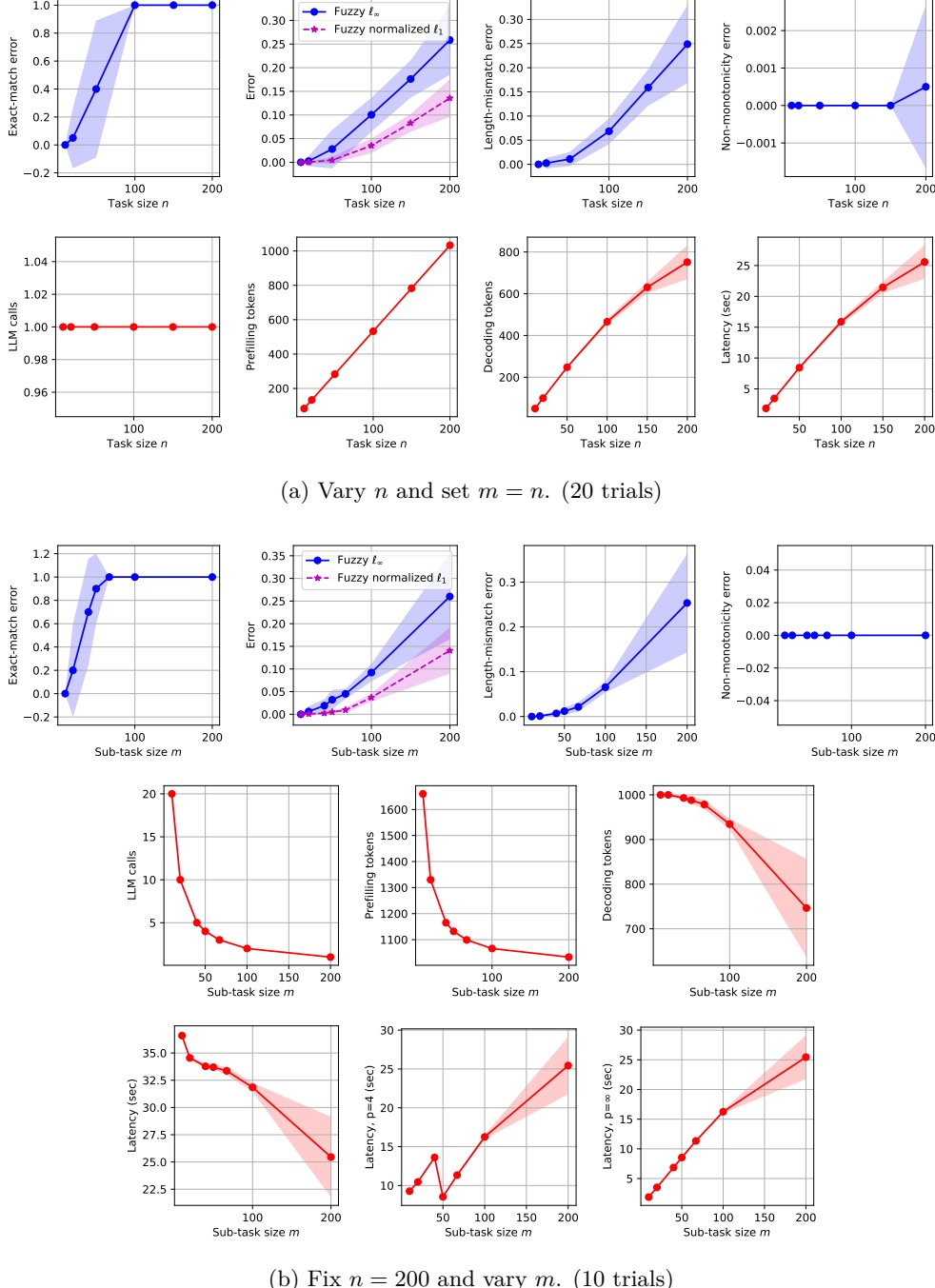

(a) Vary $n$ and set $m = n$. (20 trials)

(b) Fix $n = 200$ and vary $m$. (10 trials)

Figure 11: Empirical results for sorting with `Llama-3-70B-Instruct`.

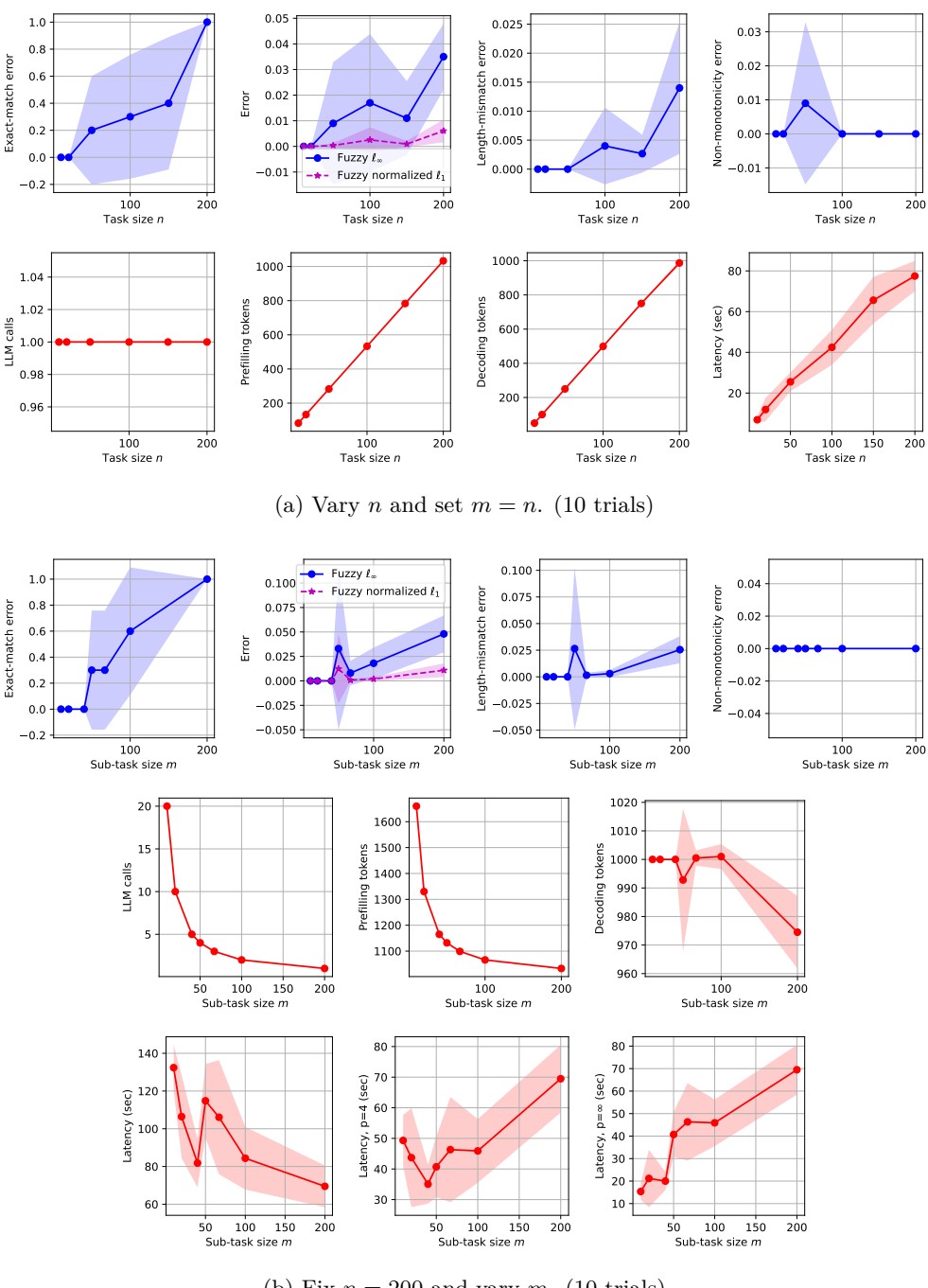

(a) Vary $n$ and set $m = n$. (10 trials)

(b) Fix $n = 200$ and vary $m$. (10 trials)

Figure 12: Empirical results for sorting with `GPT-4-Turbo`.

### A.3.4 Proof of Proposition 1

Recall that $\boldsymbol{y}_i$ denotes the solution returned by an LLM call, which is assumed to be monotone, for sorting the $i$-th input sub-list. Let $\boldsymbol{y}_i^\star$ denote the ground-truth solution for sorting the $i$-th input sub-list. Assuming that $\|\boldsymbol{y}_i - \boldsymbol{y}_i^\star\|_\infty \leq \epsilon$ for all $i \in [k]$, our goal is to prove that $\|\boldsymbol{y} - \boldsymbol{y}^\star\|_\infty \leq \epsilon$.

To start with, it is easy to check that merging multiple sorted lists is equivalent to sorting the concatenation of the lists. Therefore, we have

$$\boldsymbol{y} = \mathsf{sort}\big([\boldsymbol{y}_1, \ldots, \boldsymbol{y}_k]\big) = \mathsf{sort}\big(\mathsf{permute}([\boldsymbol{y}_1, \ldots, \boldsymbol{y}_k])\big),$$
$$\boldsymbol{y}^\star = \mathsf{sort}\big([\boldsymbol{y}_1^\star, \ldots, \boldsymbol{y}_k^\star]\big) = \mathsf{sort}\big(\mathsf{permute}([\boldsymbol{y}_1^\star, \ldots, \boldsymbol{y}_k^\star])\big),$$

where $\mathsf{permute}$ can be any permutation of $n$ elements in a list. In particular, we let $\mathsf{permute}$ be the permutation that sorts $[\boldsymbol{y}_1^\star, \ldots, \boldsymbol{y}_k^\star]$, which implies

$$\boldsymbol{y}^\star = \mathsf{sort}(\mathsf{permute}([\boldsymbol{y}_1^\star, \ldots, \boldsymbol{y}_k^\star])) = \mathsf{permute}([\boldsymbol{y}_1^\star, \ldots, \boldsymbol{y}_k^\star]).$$

Also notice that

$$\big\|\mathsf{permute}([\boldsymbol{y}_1, \ldots, \boldsymbol{y}_k]) - \mathsf{permute}([\boldsymbol{y}_1^\star, \ldots, \boldsymbol{y}_k^\star])\big\|_\infty = \big\|[\boldsymbol{y}_1, \ldots, \boldsymbol{y}_k] - [\boldsymbol{y}_1^\star, \ldots, \boldsymbol{y}_k^\star]\big\|_\infty = \max_{i \in [k]} \|\boldsymbol{y}_i - \boldsymbol{y}_i^\star\|_\infty \leq \epsilon.$$

Based on the above analysis, our initial goal boils down to the following problem: given two lists $\boldsymbol{z}, \boldsymbol{z}^\star \in \mathbb{R}^n$ such that $\|\boldsymbol{z} - \boldsymbol{z}^\star\|_\infty \leq \epsilon$ and $\boldsymbol{z}^\star$ is sorted, we need to show that $\|\mathsf{sort}(\boldsymbol{z}) - \boldsymbol{z}^\star\|_\infty \leq \epsilon$. Here, $\boldsymbol{z}$ corresponds to $\mathsf{permute}([\boldsymbol{y}_1, \ldots, \boldsymbol{y}_k])$, and $\boldsymbol{z}^\star$ corresponds to $\boldsymbol{y}^\star$.

To prove this, let us consider the classical in-place insertion-sort algorithm illustrated in Algorithm 1. We choose to prove by induction that, throughout the execution of this algorithm, it always holds that $\|\boldsymbol{z} - \boldsymbol{z}^\star\|_\infty \leq \epsilon$, which immediately implies $\|\mathsf{sort}(\boldsymbol{z}) - \boldsymbol{z}^\star\|_\infty \leq \epsilon$. To prove this, notice that the only place in Algorithm 1 where $\boldsymbol{z}$ is changed is the step of swapping $z_j$ and $z_{j-1}$ when the condition $z_j < z_{j-1}$ is satisfied. Under this condition, we have the following:

$$
\begin{aligned}
z_{j-1} - z_j^\star &> z_j - z_j^\star &\geq -\epsilon, \\
z_{j-1} - z_j^\star &\leq z_{j-1} - z_{j-1}^\star \leq &\epsilon, \\
z_j - z_{j-1}^\star &< z_{j-1} - z_{j-1}^\star \leq &\epsilon, \\
z_j - z_{j-1}^\star &\geq z_j - z_j^\star &\geq -\epsilon,
\end{aligned}
$$

which implies $|z_{j-1} - z_j^\star| \leq \epsilon$ and $|z_j - z_{j-1}^\star| \leq \epsilon$. This means that the $\ell_\infty$ error bound is preserved after this swapping step, which concludes our proof.

---

**Algorithm 1:** The classical insertion-sort algorithm

**1 Input:** a list $\boldsymbol{z} \in \mathbb{R}^n$ to be sorted.
**2 for** $i = 2, 3, \ldots, n$ **do**
**3**      **for** $j = i, i-1, \ldots, 2$ **do**
**4**          **if** $z_j \geq z_{j-1}$ **then**
**5**              Break.
**6**          Swap $z_j$ and $z_{j-1}$.
**7 Output:** the sorted list $\boldsymbol{z}$.

---

```python
1  import numpy as np
2
3
4  def merge_two_sorted_lists(list1, list2):
5      """Merge two non-empty lists or np.arrays that are
6      assumed to be (at least approximately) sorted"""
7
8      len1, len2 = len(list1), len(list2)
9      idx1, idx2 = 0, 0
10     idx = 0
11     solution = np.zeros(len1 + len2)
12     while idx < len1 + len2:
13         if idx1 == len1:
14             val = list2[idx2]
15             idx2 += 1
16         elif idx2 == len2:
17             val = list1[idx1]
18             idx1 += 1
19         else:
20             val1, val2 = list1[idx1], list2[idx2]
21             if val1 <= val2:
22                 val = val1
23                 idx1 += 1
24             else:
25                 val = val2
26                 idx2 += 1
27         solution[idx] = val
28         idx += 1
29
30     return solution
31
32
33 def merge_sorted_lists_incremental(lists):
34     """Merge lists = [list1, list2, ...] in an incremental manner"""
35
36     for _ in range(len(lists) - 1):
37         list1 = lists.pop()
38         list2 = lists.pop()
39         solution = merge_two_sorted_lists(list1, list2)
40         lists.append(solution)
41
42     return lists[0]
43
44
45 def merge_sorted_lists_hierarchical(lists):
46     """Merge lists = [list1, list2, ...] in a hierarchical manner"""
47
48     while len(lists) > 1:
49         niters = len(lists) // 2
50         for _ in range(niters):
51             list1 = lists.pop(0)
52             list2 = lists.pop(0)
53             solution = merge_two_sorted_lists(list1, list2)
54             lists.append(solution)
55
56     return lists[0]
```

Listing 1: Python code for merging sorted lists. We choose the hierarchical option in our experiments.

### A.4    Example: retrieval

Our investigation of the retrieval task draws inspiration from the needle-in-a-haystack benchmark (Kamradt, 2023) and other similar benchmarks that have been widely adopted for evaluating the long-context capability of LLMs as well as techniques of retrieval-augmented generation (Weston et al., 2015; Roucher, 2023; Kuratov et al., 2024; Mohtashami & Jaggi, 2023; Zhang et al., 2024a). For concreteness, consider the following setting. A key message (the needle) of the form "The passcode to the {targeted object, e.g., red door} is {6-digit passcode}" is randomly inserted into a piece of long text (the haystack). The algorithm is asked to answer the question "What is the passcode to the {targeted object}?". To make the problem more challenging and fun, we let the haystack consist of alike sentences of the form "The passcode to the {colored object, e.g., red lock or green door} is {6-digit passcode}", with colored objects different from the targeted object. This allows us to investigate both sides of retrieval capabilities of LLMs and LLM-based algorithms: retrieving the targeted information correctly, while avoiding being confused or misled by background information that might seem relevant to the question (Shi et al., 2023). Note that while we use this concrete setting for our empirical study, the proposed algorithm and analysis in the following are actually applicable to generic cases of this retrieval task.

#### A.4.1    Algorithm details

We note a few details about the LLM-based retrieval algorithm presented in Table 1.

- All lengths involved in the algorithm are measured by the number of characters.

- In the first step of chunking, we let each pair of adjacent chunks share an overlap that is larger than the length of the needle, to ensure that the needle will appears as a whole in at least one chunk.

- For each LLM call in the second step, the LLM is prompted to answer "I don't know" if it believes that there is not sufficient information in the corresponding chunk, e.g., when the chunk simply does not contain the needle. Such answers will be excluded from the final step of the algorithm.

- In the final step, i.e., majority voting, it is possible that there exist multiple (say $h$) candidate solutions with the same frequency, in which case we let the algorithm return the list of such candidates. If this list contains the ground-truth solution, we calculate the exact-match error as $1 - 1/h$ in our experiments.

#### A.4.2    Cost analysis

Let us assume for concreteness that each pair of adjacent chunks share an overlap of length $m/2$, and $m_i = m$ for all $i \in [k-1]$, while $m/2 \leq m_k \leq m$. In this case, we have $k = \lceil 2n/m - 1 \rceil$.

Our cost analysis for this task is largely the same as that for the counting task, despite how different these two tasks might seem. Under some mild conditions explained in Section 3.1, most cost metrics of interest are monotonely decreasing in $m$, except for the end-to-end latency with parallel LLM calls, which is increasing in $m$ for parallelism degree $p = \infty$ and possibly non-monotone for finite $p$. One thing to note is that, due to the overlaps between consecutive chunks, the number of parallel sub-tasks in Step 2 of the algorithm is $k = \lceil 2n/m - 1 \rceil$, rather than $\lceil n/m \rceil$ as in the counting task. This implies that the value of $m$ minimizing the latency can be predicted by letting $2n/m - 1 \approx p$, namely $m \approx 2n/(p+1)$.

#### A.4.3    Experiments

We validate our analysis with numerical experiments. For each task instance, the passcodes of all objects, the targeted object, the position of the haystack where the needle is inserted, etc., are all randomly chosen. The error metric of interest, namely the exact-match error, takes value 0 if the final solution of the algorithm is exactly the same as the ground-truth passcode to the targeted object, $1 - 1/h$ if the algorithm returns a list of $h$ candidate solutions that includes the ground-truth passcode, and 1 otherwise.

**Results with `Llama-3-8B-Instruct`.**   The results are illustrated in Figure 13.

In Figure 13a, we vary the length $n$ of the input text containing one needle in a haystack, and set $m = n$ for the LLM-based algorithm, which becomes equivalent to a single LLM call. Notice that the exact-match error, which corresponds to the first failure mode explained earlier, approaches zero as the problem size $n$ decreases. In addition, the decoding length is $O(1)$, and the wall-clock latency is $O(\mathcal{L}_{\mathsf{sys}} + n)$; one detailed observation is that as $n$ increases, it becomes more likely that the LLM call returns "I don't know", which slightly decreases the number of decoding tokens and thus the latency.

Figure 13b includes the results for the same experiment setting, except that no needle is inserted into the haystack. One crucial observation is that the exact-match error in this case, which corresponds to the second failure mode of retrieval, remains non-zero even for very small values of $n$. This suggests that `Llama-3-8B-Instruct` should be regarded as a Type-B LLM that is prone to both failure modes explained in Section 3.2.3.

In Figures 13c and 13d, we vary the sub-task size $m$ while $n$ is fixed at 10000 and 20000 respectively. As was predicted by our analysis, the error of the overall algorithm is not monotone in the value of $m$, due to the presence of two failure modes. Another difference from the previous counting or sorting task is that the latency with parallelism degree $p = 4$ achieves the minimum around $m = 2n/(p + 1) = 0.4n$ rather than $m = n/p = 0.25n$, which again matches our previous analysis.

**Results with `Llama-3-70B-Instruct`.**   Figure 14 includes the results for the same experiments but with `Llama-3-70B-Instruct` used within the LLM-based algorithm. In particular, Figure 14a shows that this LLM achieves lower errors in the first failure mode of retrieval compared to `Llama-3-8B-Instruct`, while Figure 14b suggests that it is much less prone to the second failure mode and hence might be regarded as a Type-A LLM. Consequently, in Figures 14c and 14d, the exact-match error exhibits a more monotone relation with the sub-task size $m$. Regarding the cost metrics, results with `Llama-3-70B-Instruct` are similar to those with `Llama-3-8B-Instruct`.

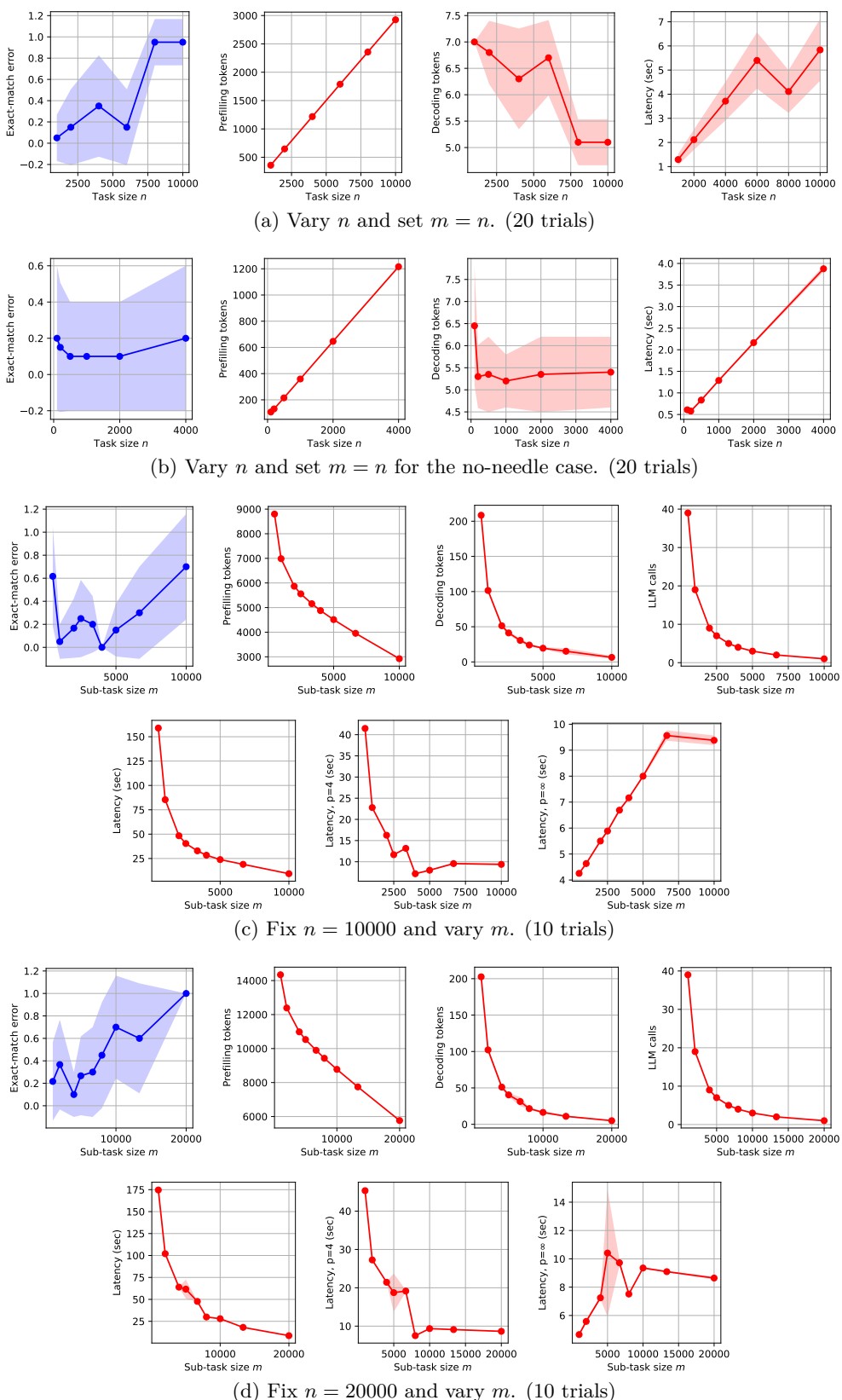

(a) Vary $n$ and set $m = n$. (20 trials)

(b) Vary $n$ and set $m = n$ for the no-needle case. (20 trials)

(c) Fix $n = 10000$ and vary $m$. (10 trials)

(d) Fix $n = 20000$ and vary $m$. (10 trials)

Figure 13: Empirical results for retrieval with `Llama-3-8B-Instruct`.

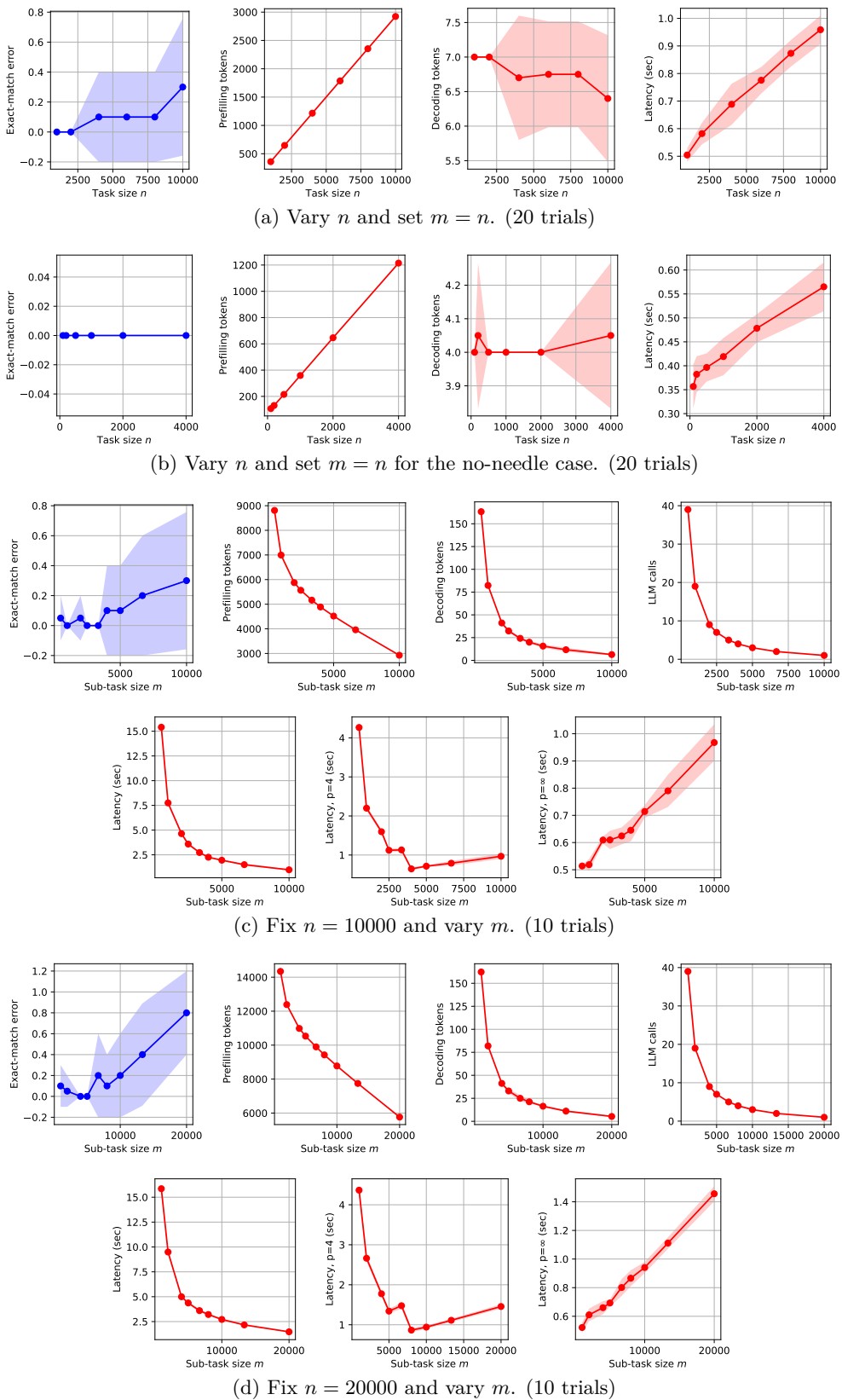

(a) Vary $n$ and set $m = n$. (20 trials)

(b) Vary $n$ and set $m = n$ for the no-needle case. (20 trials)

(c) Fix $n = 10000$ and vary $m$. (10 trials)

(d) Fix $n = 20000$ and vary $m$. (10 trials)

Figure 14: Empirical results for retrieval with `Llama-3-70B-Instruct`.

### A.5 Example: retrieval-augmented generation

Our next example is a multiple-needle generalization of the previous retrieval task, which can be regarded as a synthetic and simplified version of retrieval-augmented generation (RAG) (Lewis et al., 2020; Gao et al., 2024b). In particular, we consider fine-grained sentence-level retrieval by LLMs, rather than document-level or chunk-level retrieval by certain similarity measure of dense embedding vectors.

More concretely, suppose that the input text is composed of sentences of the form "The {i}-th digit of the passcode to the {colored object}" is {digit}", where $i \in [6]$. The algorithm is asked to answer the question "What is the 6-digit passcode to the {targeted object}?". Compared with the previous single-needle retrieval task, here the algorithm need to retrieve multiple needles, each for one digit of the targeted object, in order to answer the question correctly; moreover, the final aggregation step requires certain capability of reasoning or summarization over the retrieved needles.

Note again that while we focus on this specific setting in our experiments, the algorithm and analysis in the following can be extended to generic settings of this RAG task.

#### A.5.1 Algorithm

We consider the following LLM-based algorithm for solving this task:

1. Divide the input text of length $n$ into $k$ overlapping chunks of lengths $m_1, \ldots, m_k$;

2. For each chunk, use one LLM call to retrieve sentences that can be useful for answering the question;

3. Put the retrieved sentences together, based on which one LLM call is invoked for answering the question.

We note a few details about this algorithm. (1) For each chunk in Step 2, we prompt the LLM to retrieve relevant sentences, or return "None" if no relevant sentence is found in that chunk. Such "None" results will be excluded from Step 3 of the algorithm. (2) Unlike previous examples, the final aggregation step of this algorithm involves an LLM node, which adds to the cost metrics of the overall algorithm. (3) For simplicity, we assume that the number of needles (i.e., length of the passcode) and the length of each needle are both $O(1)$. For more general cases, the final aggregation step might benefit from further task decomposition. (4) We allow the algorithm to return a partial answer, by placing a special character in the digit(s) of the passcode that it is uncertain about.

#### A.5.2 Analysis

Let us assume for concreteness that each pair of adjacent chunks share an overlap of length $m/2$, and $m_i = m$ for all $i \in [k-1]$, while $m/2 \le m_k \le m$. In this case, we have $k = \lceil 2n/m - 1 \rceil$.

**Error analysis.** Our analysis of error metrics for this task is similar to that for the previous retrieval example. In particular, there are two possible failure modes in the retrieval step, and conclusions for the errors of the final solution returned by the overall algorithm are dependent on whether a Type-A or Type-B LLM is being used. For example, if a Type-A LLM, which will not mistakenly retrieve irrelevant sentences from the input text, is used within the algorithm, then a smaller value of $m$ implies higher accuracy of retrieval in Step 2 of the algorithm, which further leads to lower error metrics for the solution returned by the final aggregation step.

**Cost analysis.** For simplicity, let us assume that a Type-A LLM is used within the overall algorithm. Consequently, in Step 2 of the algorithm, each LLM call has $\mathcal{L}_{\mathsf{pre}} \le \mathcal{L}_{\mathsf{sys}} + O(m)$ and $\mathcal{L}_{\mathsf{dec}} = O(1)$, since only the relevant text within the chunk is retrieved. Moreover, among the $k$ LLM calls, only $O(1)$ of them return answers that are not "None". By excluding the "None" results from the final aggregation step, the last LLM call has $\mathcal{L}_{\mathsf{pre}} \le \mathcal{L}_{\mathsf{sys}} + O(1)$ and $\mathcal{L}_{\mathsf{dec}} = O(1)$. Putting things together, with a degree of parallelism $p$ and the number of chunks $k = \lceil 2n/m - 1 \rceil$, the end-to-end latency $\mathcal{C}$ of the overall algorithm is bounded by

$$\mathcal{C} = \mathcal{C}(\text{sub-tasks}) + \mathcal{C}(\text{aggregation}), \quad \text{where}$$

$$\mathcal{C}(\text{sub-tasks}) \leq \left\lceil \frac{k}{p} \right\rceil \times \Big( \mathcal{C}_{\mathsf{pre}}(\mathcal{L}_{\mathsf{sys}} + m) + \mathcal{C}_{\mathsf{dec}}(\mathcal{L}_{\mathsf{sys}} + m, 1) \Big),$$

$$\mathcal{C}(\text{aggregation}) \leq \mathcal{C}_{\mathsf{pre}}(\mathcal{L}_{\mathsf{sys}} + 1) + \mathcal{C}_{\mathsf{dec}}(\mathcal{L}_{\mathsf{sys}} + 1, 1).$$

### A.5.3  Experiments

We empirically validate the performance of the proposed LLM-based algorithm with a `Llama-3-70B-Instruct` model. Two error metrics are considered: the exact-match error taking value in $\{0, 1\}$, and the fraction of incorrect digits, which takes value in $[0, 1]$ and is always no larger than the exact-match error.

In Figure 15a, we vary the problem size $n$, and set $m = n$ in the LLM-based algorithm. It is observed that the error metrics are monotonely increasing in $n$, and approach zero as $n$ decreases. Most cost metrics are also increasing in $n$, except for the number of decoding tokens, which is supposed to be determined by the number and lengths of the needles only, not the haystack, and thus should be insensitive to $n$.

In Figure 15b, we fix $n = 20000$ and vary the chunk size $m$. As was predicted by our analysis for a Type-A LLM, a smaller value of $m$ implies lower error metrics, indicating the efficacy of task decomposition.

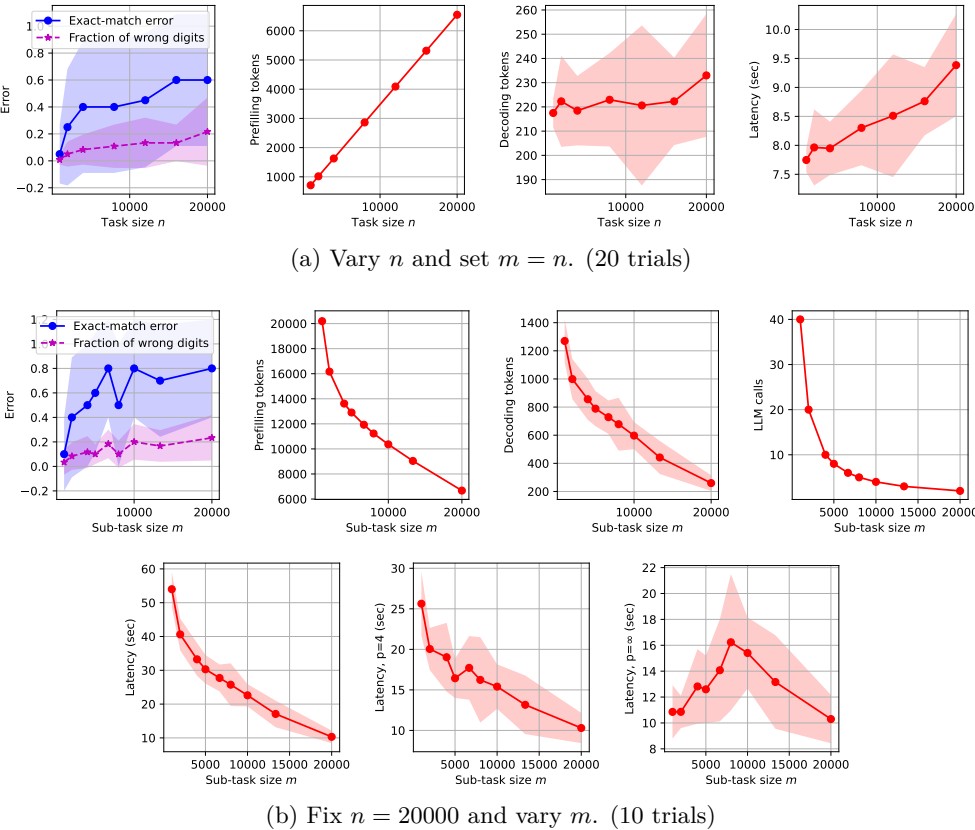

(a) Vary $n$ and set $m = n$. (20 trials)

(b) Fix $n = 20000$ and vary $m$. (10 trials)

Figure 15: Empirical results for RAG with `Llama-3-70B-Instruct`.

## A.6 Example: long-text summarization with chunking

For the final example of parallel decomposition, we apply our analytical framework to the task of long-text summarization (Kryscinski et al., 2022; Chang et al., 2024), where generating a summary for the long input text with a single LLM call is infeasible. Algorithms that process the long text to be summarized by chunks have been proposed, such as hierarchical merging (Wu et al., 2021) and incremental updating (OpenAI, 2024b). The former first generates one summary for each chunk independently and then merge these intermediate summaries into the final one, while the latter maintains and updates a global summary as the chunks get processed one by one in order. Since objective and quantitative evaluation for the summarization task is known to be challenging (Chang et al., 2024) and remains an active research area, we focus our study on the cost metrics of both algorithms. Our analysis leads to interesting observations that are different from those in previous examples, primarily due to the various options of setting the number of generated tokens $\mathcal{L}_{\mathsf{dec}}$ for each LLM call within the algorithms.

### A.6.1 Processing chunks in parallel

**Algorithm.** Let us consider the following algorithm that processes chunks in parallel, which is a simplification of hierarchical merging (Chang et al., 2024) and visualized in Figure 3a:

1. Divide the input text of length $n$ into $k$ chunks of length $m_1, m_2, \ldots, m_k$.

2. Summarize each chunk with one LLM call, independently and in parallel. For the $i$-th chunk, the LLM is prompted to generate a summary of length no larger than $s_i$.

3. Invoke one LLM call to merge the summaries into a single summary of length no larger than $s$.

Note that the targeted lengths of the intermediate and final summaries, denoted by $\{s_i\}_{i \in [k]}$ and $s$, are hyperparameters of the algorithm that need to be pre-specified, in addition to the number of chunks $k$ and the chunk sizes $\{m_i\}_{i \in [k]}$.

**Analysis.** For notational convenience, we assume that the chunk sizes satisfy $m_i \asymp m := n/k$ for all $i \in [k]$.

First, consider each individual LLM call: the cost of summarizing the $i$-th chunk can be bounded by $\mathcal{C}_{\mathsf{LLM}}(\mathcal{L}_{\mathsf{sys}} + m, s_i)$, while the cost of the final merging step can be bounded by $\mathcal{C}_{\mathsf{LLM}}(\mathcal{L}_{\mathsf{sys}} + \sum_{i \in [k]} s_i, s)$. To further simplify notations, we assume that $s_i = s_1$ for all $i \in [k]$, which will soon be justified. Then the total cost $\mathcal{C}$ of the overall algorithm can be bounded by

$$\mathcal{C} \le \mathcal{C}(\text{summarize chunks}) + \mathcal{C}(\text{merge summaries})$$
$$= k \times \mathcal{C}_{\mathsf{LLM}}(\mathcal{L}_{\mathsf{sys}} + m, s_1) + \mathcal{C}_{\mathsf{LLM}}(\mathcal{L}_{\mathsf{sys}} + k \times s_1, s),$$

while the end-to-end latency with parallelism degree $p$ can be bounded by

$$\mathcal{C} \le \lceil \frac{k}{p} \rceil \times \mathcal{C}_{\mathsf{LLM}}(\mathcal{L}_{\mathsf{sys}} + m, s_1) + \mathcal{C}_{\mathsf{LLM}}(\mathcal{L}_{\mathsf{sys}} + k \times s_1, s).$$

Let us specify the hyperparameters $s_1$ and more generally $\{s_i\}$, assuming that the targeted length $s$ of the final summary has been determined. If the targeted information that we wish to be included in the final summary is distributed evenly across the input text, e.g., in a story with a linear narrative, then it is reasonable to set $s_i = s_1 = s/k$ for all $i \in [k]$. In other scenarios where the targeted information is distributed unevenly (e.g., in query-focused summarization (Laban et al., 2024; Vig et al., 2022)), it is generally safer to set $s_i = s_1 = s$ for all $i \in [k]$. The latency with parallelism degree $p$ in both cases is summarized as follows:

$$\mathcal{C} \le \begin{cases} \lceil k/p \rceil \times \mathcal{C}_{\mathsf{LLM}}(\mathcal{L}_{\mathsf{sys}} + m, s/k) + \mathcal{C}_{\mathsf{LLM}}(\mathcal{L}_{\mathsf{sys}} + s, s) & \text{if} \quad s_1 = s/k, \\ \lceil k/p \rceil \times \mathcal{C}_{\mathsf{LLM}}(\mathcal{L}_{\mathsf{sys}} + m, s) + \mathcal{C}_{\mathsf{LLM}}(\mathcal{L}_{\mathsf{sys}} + k \times s, s) & \text{if} \quad s_1 = s. \end{cases}$$

Further analysis can be derived from here. To give an example, let us focus on the case of $s_1 = s$ and try to find the chunk size $m$ that minimizes the latency with parallelism degree $p$. For simplicity, we ignore the length $\mathcal{L}_{\mathsf{sys}}$ of the system prompt and the limit $\overline{m}$ on the context window size. We also assume that the time complexity of each LLM call is linear, namely $\mathcal{C}_{\mathsf{LLM}}(\mathcal{L}_{\mathsf{pre}}, \mathcal{L}_{\mathsf{dec}}) \lesssim \mathcal{L}_{\mathsf{pre}} + \mathcal{L}_{\mathsf{dec}}$. Then we have

$$\mathcal{C} \leq \lceil \frac{k}{p} \rceil \times \mathcal{C}_{\mathsf{LLM}}(\mathcal{L}_{\mathsf{sys}} + m, s) + \mathcal{C}_{\mathsf{LLM}}(\mathcal{L}_{\mathsf{sys}} + k \times s, s) \lesssim \lceil \frac{k}{p} \rceil \times (m + s) + k \times s.$$

Even in such an oversimplified case, finding the optimal chunk size $m$ is non-trivial:

- For large $m \geq \widetilde{m} := n/p$ such that $k = n/m < p$, we have $\lceil k/p \rceil = 1$, and hence

$$\mathcal{C} \lesssim m + s + k \times s = m + \frac{n \times s}{m} + s.$$

  The right-hand side attains the minimum at $\widehat{m} := \sqrt{n \times s}$.

- For small $m < \widetilde{m} = n/p$, we have the approximation $\lceil k/p \rceil \approx k/p$, and hence

$$\mathcal{C} \lesssim \frac{k}{p} \times (m + s) + k \times s = k \times \left( \frac{m}{p} + \left( \frac{1}{p} + 1 \right) \times s \right)$$
$$\lesssim \frac{n}{m} \times \left( \frac{m}{p} + s \right) = n \times \left( \frac{1}{p} + \frac{s}{m} \right).$$

  The right-hand side is monotonely decreasing in $m$.

Given the above, it can be verified that the optimal chunk size $m^\star$ that minimizes the latency $\mathcal{C}$ can be estimated by $m^\star \asymp \max\{\widehat{m}, \widetilde{m}\} = \max\{\sqrt{n \times s}, n/p\}$.

### A.6.2 Processing chunks sequentially

**Algorithm.** Let us consider the following algorithm that processes chunks sequentially, which is a simplification of incremental updating (Chang et al., 2024) and visualized in Figure 3b:

1. Divide the input text of length $n$ into $k$ chunks of length $m_1, m_2, \ldots, m_k$.

2. Initialize the global summary as an empty string, and update it incrementally via processing the chunks in order: at the $i$-th iteration, invoke one LLM call to utilize the global summary and the $i$-th chunk for generating a new global summary of length no larger than $s_i$.

3. The output of the overall algorithm is that of the final LLM call for summarizing the last chunk.

**Analysis.** While this algorithm clearly has a sequential nature, our previous analysis for parallel decomposition can be easily adapted for this case. For the $i$-th step of updating the global summary, we have $\mathcal{L}_{\mathsf{pre}} \leq \mathcal{L}_{\mathsf{sys}} + m + s_{i-1}$ and $\mathcal{L}_{\mathsf{dec}} \leq s_i$, and thus its cost is bounded by

$$\mathcal{C}_i \leq \mathcal{C}_{\mathsf{LLM}}(\mathcal{L}_{\mathsf{sys}} + m + s_{i-1}, s_i).$$

Therefore, the total cost of the overall algorithm satisfies

$$\mathcal{C} \leq \sum_{i \in [k]} \mathcal{C}_i \leq \sum_{i \in [k]} \mathcal{C}_{\mathsf{LLM}}(\mathcal{L}_{\mathsf{sys}} + m + s_{i-1}, s_i).$$

Further analysis can provide insights for choosing the hyperparameters that minimize the total cost, though we omit the details here to avoid repetition.

# B   Proof of Proposition 2

We prove the proposition by induction. First, consider the base case where $v$ is a leaf node of the DAG that has no input. Then it can be checked that Eq. (6) holds true:

$$\mathcal{E}(y(v)) \le \mathcal{E}_v = \sum_{w \in \mathcal{V}} \sum_{\text{path} \in \mathcal{P}(w \to v)} S^{|\text{path}|} \times \mathcal{E}_w.$$

The inequality follows the assumption, while the equality holds true because $\mathcal{P}(w \to v)$ for a leaf node $v$ is an empty set unless $w = v$, in which case $\mathcal{P}(w \to v)$ contains only a hypothetical path of length 0.

Next, consider a non-leaf node $v$, whose predecessor nodes are denoted by $u_1, u_2, \ldots, u_k$. Suppose that Eq. (6) holds true for each $u_i$, namely

$$\mathcal{E}(y(u_i)) \le \sum_{w \in \mathcal{V}} \sum_{\text{path} \in \mathcal{P}(w \to u_i)} S^{|\text{path}|} \times \mathcal{E}_w, \quad i \in [k].$$

To prove that Eq. (6) also holds true for node $v$, we start with the following:

$$\mathcal{E}(y(v)) \le \mathcal{E}_v + S \times \sum_{i \in [k]} \mathcal{E}(y(u_i))$$

$$\le \mathcal{E}_v + S \times \sum_{i \in [k]} \sum_{w \in \mathcal{V}} \sum_{\text{path} \in \mathcal{P}(w \to u_i)} S^{|\text{path}|} \times \mathcal{E}_w$$

$$= \mathcal{E}_v + S \times \sum_{i \in [k]} \sum_{w \in \mathcal{V} \setminus \{v\}} \sum_{\text{path} \in \mathcal{P}(w \to u_i)} S^{|\text{path}|} \times \mathcal{E}_w$$

$$= \mathcal{E}_v + S \times \sum_{w \in \mathcal{V} \setminus \{v\}} \sum_{i \in [k]} \sum_{\text{path} \in \mathcal{P}(w \to u_i)} S^{|\text{path}|} \times \mathcal{E}_w$$

$$= \mathcal{E}_v + \sum_{w \in \mathcal{V} \setminus \{v\}} \sum_{i \in [k]} \sum_{\text{path} \in \mathcal{P}(w \to u_i)} S^{|\text{path}|+1} \times \mathcal{E}_w.$$

Here in the third line, we replace the summation over $w \in \mathcal{V}$ with $w \in \mathcal{V} \setminus \{v\}$, since there is certainly no path from node $v$ to its predecessor node $u_i$. To move forward, notice that $\mathcal{P}(w \to v)$ is a union of $k$ disjoint sets, where the $i$-th set contains paths ending with the directed edge from $u_i$ to $v$:

$$\mathcal{P}(w \to v) = \bigcup_{i \in [k]} \Big\{ \text{path} + [u_i \to v] : \text{path} \in \mathcal{P}(w \to u_i) \Big\}.$$

With this, we can further simplify the previous upper bound:

$$\mathcal{E}(y(v)) \le \mathcal{E}_v + \sum_{w \in \mathcal{V} \setminus \{v\}} \sum_{i \in [k]} \sum_{\text{path} \in \mathcal{P}(w \to u_i)} S^{|\text{path}|+1} \times \mathcal{E}_w$$

$$= \mathcal{E}_v + \sum_{w \in \mathcal{V} \setminus \{v\}} \sum_{\text{path} \in \mathcal{P}(w \to v)} S^{|\text{path}|} \times \mathcal{E}_w$$

$$= \sum_{w \in \mathcal{V}} \sum_{\text{path} \in \mathcal{P}(w \to v)} S^{|\text{path}|} \times \mathcal{E}_w.$$

This concludes our proof of the proposition.

# C    Supplementary materials for Section 4.2

This appendix includes full details of our case study, presented in Section 4.2, on iterative retrieval and reasoning. For concreteness, we investigate a challenging version of the RAG task studied in Section A.5 that requires multi-hop reasoning (Yang et al., 2018; Li et al., 2024a). Recall that in the previous RAG example, given a question and multiple text chunks, the algorithm (with the pattern of parallel decomposition) will first retrieve useful sentences from each chunk, and then invoke one LLM call to answer the question based on the retrieved information. Such a method might not be feasible in challenging scenarios where the needles embedded in the haystack are logically related, and some of the needles are related to the targeted question only *indirectly* via connection to other needles. For example, suppose that the question is "what is the numeric value of A", while the needles are "A = B", "B = C", and "C = 100", located separately in different chunks. Retrieving needles from their corresponding chunks solely based on the targeted question will certainly fail in this case.

To tackle this challenge, a natural idea is to extend the RAG algorithm considered in Section A.5, allowing it to perform multiple rounds of iterative retrieval and reasoning. For each round, the algorithm performs retrieval based on the targeted question as well as additional information collected from previous rounds, followed by reasoning about the retrieved sentences and deciding whether the targeted question can be answered, or further retrieval and reasoning is needed. Similar approaches have been widely adopted in prior work, e.g., in the Selection-Inference framework (Creswell et al., 2023), in a RAG system with iterative follow-up questions for applications in medicine (Xiong et al., 2024), as a technique of extending the effective context window of a LLM-powered agent system (Qwen-Team, 2024), among others (Yue et al., 2024). The resulting algorithm for reasoning with iterative retrieval exhibits a hierarchical structure: the original task is decomposed into multiple sequential rounds, and each round is further decomposed into multiple steps of retrieval and reasoning.

In the rest of this section, we first elaborate the concrete setup and algorithms for this case study, and then demonstrate our analysis of accuracy and efficiency based on the proposed analytical framework, as well as insights derived from it. Finally, numerical experiments are conducted to validate the efficacy of the considered algorithm, as well as the derived analysis and insights.

## C.1    Concrete setup

For concreteness, let us consider the task of finding the numeric value of a targeted variable embedded within a grid, similar to the setting considered in (Ye et al., 2024). A visualization can be found in Figure 16. Configurations of the grid include its depth $d$, width $w$, and degree $g$, which control the difficulty of this task. More specifically, the grid consists of $d$ levels, each containing $w$ variables; each variable at a certain level is a function (e.g., addition, substraction, maximum or minimum) of $g$ variables at the next level, except for the variables at the final level, whose numeric values are given. Such information is provided in clues of the form "A2 = B1 + B3" or "C1 = 10", each of which corresponds to one variable. As a result, the total number of clues is $d \times w$, and each clue has length $O(g)$. We refer to the clues that are necessary and sufficient for calculating the targeted variable as the needles, which constitute a directed acyclic graph (DAG) embedded within the grid. It is assumed that the level of reasoning required in this case study is non-trivial yet also simple enough, in the sense that one single LLM call with step-by-step reasoning is sufficient for answering the targeted question correctly if all needles are given *a priori*.

## C.2    Algorithm

To solve this task, we consider an LLM-based algorithm that involves multiple rounds of retrieval and reasoning, which is visualized in Figure 5 and explained in the following:

1. Divide the input clues into $k$ chunks. In addition, initialize an empty list of references, i.e., clues retrieved by LLM calls, which will be maintained and updated throughout the algorithm.

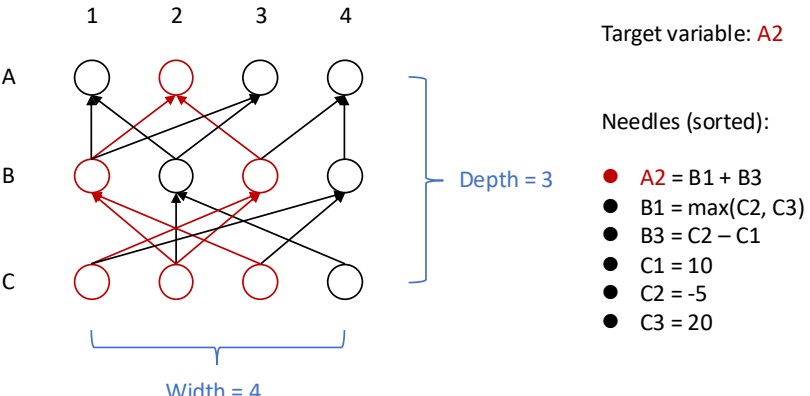

Figure 16: An example grid of variables with depth $d = 3$, width $w = 4$, and degree $g = 2$. The targeted variable is A2 at the top level, which is a function of B1 and B3 at the next level. The variables whose numeric values are necessary and sufficient for calculating A2 are highlighted in red, and their corresponding clues constitute a DAG of needles.

2. For each round of the algorithm, invoke $k$ sequential (Figure 5a) or parallel (Figure 5b) LLM calls for retrieval from $k$ chunks based on the targeted question and references, then invoke one LLM call to reason about the updated references and try to answer the targeted question.

   - If the LLM returns an answer, then the overall algorithm outputs that answer and terminate.
   - If the LLM cannot answer, and no additional clue has been retrieved in this round, then the algorithm terminates and fails[5].
   - Otherwise, move on to the next round of retrieval and reasoning, with the updated references.

**Options for reasoning and answering.**   We consider two options of prompting the LLM to reason about the references and answer the targeted question: answering directly, or thinking step by step before answering (Kojima et al., 2022). It is reasonable to expect that the latter will boost the accuracy while incurring higher costs, which will soon be investigated analytically and quantitatively in Section C.3.

**Options for retrieval.**   We consider two options, referred to as "cyclic" and "parallel", for retrieval from multiple chunks during each round of the algorithm.

   - With the "cyclic" option (Figure 5a), chunks are processed sequentially; more specifically, after retrieval for each chunk, the retrieved clues are added to the list of references immediately, before retrieval for the next chunk starts. Consequently, the chunks are processed in a cyclic manner throughout the overall algorithm, which gives rise to the name of this option.

   - With the "parallel" option (Figure 5b), chunks are processed independently and in parallel, after which the list of references is updated once. This exemplifies using parallel decomposition (cf. Section 3) as a building block for more complicated LLM-based algorithms.

With the "cyclic" option, the overall algorithm typically need fewer rounds to answer the targeted question correctly, since each LLM call for retrieval always leverages the most updated references. On the other hand, the "parallel" option can leverage parallelism of LLM calls for reducing the end-to-end latency of the overall algorithm. The comparison between these two options will soon be elaborated in Section C.3.

---

[5]This is due to the assumption that greedy decoding, which is deterministic, is adopted. In this case, running one more round of retrieval and reasoning will give the same result as that of the current round, which will be useless and wasteful. A potential improvement is to adaptively decrease (e.g., halve) the chunk size and continue the algorithm with more rounds; such improvements are not the focus of this work, and hence omitted here.

### C.3 Analysis

#### C.3.1 Error analysis

There exist two major failure modes for this algorithm:

1. *Mistakes in reasoning and answering.* For example, the LLM call responsible for reasoning and answering might mistakenly decide that it is not yet ready to answer the targeted question, or make mistakes when doing the arithmetic and thus return a wrong numeric value, even when all needles have been successfully retrieved. It is also possible that the LLM call mistakenly returns a numeric value (e.g., the value of a variable whose name is similar to that of the targeted variable), while the needles have not been fully retrieved yet. The probability of this failure mode is particularly related to whether the LLM call responsible for reasoning and answering is prompted to answer directly or think step by step before answering.

2. *Failure to retrieve all needles.* As long as the ground-truth DAG of needles has a limited size, the probability of this failure mode is largely determined by the choice of the chunk size. Since this has been thoroughly investigated in our previous retrieval (Section A.4) and RAG (Section A.5) examples, it will not be our focus in this section.

#### C.3.2 Cost analysis

**Notations.** Recall from Eq. (2) that $\mathcal{C}_{\mathsf{LLM}}(\mathcal{L}_{\mathsf{pre}}, \mathcal{L}_{\mathsf{dec}}) = \mathcal{C}_{\mathsf{pre}}(\mathcal{L}_{\mathsf{pre}}) + \mathcal{C}_{\mathsf{dec}}(\mathcal{L}_{\mathsf{pre}}, \mathcal{L}_{\mathsf{dec}})$ stands for the cost of one LLM call with a prompt of length $\mathcal{L}_{\mathsf{pre}}$ and generated text of length $\mathcal{L}_{\mathsf{dec}}$. We also adopt some notations from Section 3 in our previous study of parallel decomposition: $n$ represents the total length (in tokens or characters) of the input clues, $m$ represents an upper bound for the chunk size, $k \lesssim n/m$ denotes the number of chunks, and $\mathcal{L}_{\mathsf{sys}}$ denotes an upper bound for the length of the system prompt for each LLM call.

In addition, let $r_{\mathsf{cyc}}$ (resp. $r_{\mathsf{par}}$) denote the number of rounds determined adaptively by the "cyclic" (resp. "parallel") algorithm itself at runtime. Finally, let $\ell$ be the total length of the needles, i.e., clues that are necessary and sufficient for answering the targeted question correctly. The value of $\ell$ can depend on the depth $d$, width $w$ and degree $g$ of the grid in various ways:

- If $g = 1$, then the needles form a chain of length $d$, in the form of "A = B", "B = C", "C = D", and so on. In this case, we have $\ell \lesssim d$.

- If the width $w > d^{g-1}$ is sufficiently large, then the number of needles can be upper bounded by the size of a tree with $d$ levels and $g$ children per node, namely $1 + g + g^2 + \cdots + g^{d-1} = (g^d - 1)/(g - 1)$. Since each needle has length $O(g)$, we have $\ell \lesssim g \times (g^d - 1)/(g - 1)$.

- If the width $w \ll d^{g-1}$ is limited, then a tighter bound for the number of needles will be $d \times w$, and thus $\ell \lesssim g \times d \times w$.

**The "cyclic" version (Figure 5a).** Let us first study the cost of each LLM call.

- An LLM call responsible for retrieval takes one chunk and the references as input, which has length $O(m + \ell)$. In addition, it is supposed to output a list of clues, which has length $O(\ell)$, that is relevant to the targeted question. In other words, we have $\mathcal{L}_{\mathsf{pre}} \leq \mathcal{L}_{\mathsf{sys}} + O(m + \ell)$, $\mathcal{L}_{\mathsf{dec}} \lesssim \ell$, and thus

$$\mathcal{C}(\text{retrieval}) \leq \mathcal{C}_{\mathsf{LLM}}(\mathcal{L}_{\mathsf{sys}} + m + \ell, \ell)$$

- An LLM call responsible for reasoning and answering takes the references of length $O(\ell)$ as input, namely $\mathcal{L}_{\mathsf{pre}} \leq \mathcal{L}_{\mathsf{sys}} + O(\ell)$. As for the length of generated text $\mathcal{L}_{\mathsf{dec}}$, it is reasonable to assume that $\mathcal{L}_{\mathsf{dec}} \lesssim 1$ if the LLM is prompted to answer directly, and $\mathcal{L}_{\mathsf{dec}} \lesssim \ell$ if it is prompted to do the

calculation step by step before returning its final answer[6]. Consequently, one has

$$\mathcal{C}(\text{reasoning}) \leq \mathcal{C}_{\mathsf{LLM}}(\mathcal{L}_{\mathsf{sys}} + \ell, 1 \text{ or } \ell).$$

Now we are ready to find out the cost of the overall algorithm with $r_{\mathsf{cyc}}$ rounds of iterative retrieval (for $k$ chunks) and reasoning:

$$\mathcal{C} \leq r_{\mathsf{cyc}} \times \mathcal{C}(\text{one round}) \tag{7}$$

$$= r_{\mathsf{cyc}} \times \Big( k \times \mathcal{C}(\text{retrieval}) + \mathcal{C}(\text{reasoning}) \Big) \tag{8}$$

$$\leq r_{\mathsf{cyc}} \times \Big( k \times \mathcal{C}_{\mathsf{LLM}}(\mathcal{L}_{\mathsf{sys}} + m + \ell, \ell) + \mathcal{C}_{\mathsf{LLM}}(\mathcal{L}_{\mathsf{sys}} + \ell, 1 \text{ or } \ell) \Big). \tag{9}$$

In particular, this bound quantifies how the cost of LLM calls responsible for reasoning and answering, namely $r_{\mathsf{cyc}} \times \mathcal{C}_{\mathsf{LLM}}(\mathcal{L}_{\mathsf{sys}} + \ell, 1 \text{ or } \ell)$, only occupies a small fraction of the total cost of the overall algorithm.

**The "parallel" version (Figure 5b).** Analysis of cost metrics for the "parallel" version is largely the same as that for the "cyclic" version, with the following two major differences:

- For the same task instance, the "parallel" version requires the same or a larger number of rounds ($r_{\mathsf{par}} \geq r_{\mathsf{cyc}}$), since in the "cyclic" version, each retrieval step always leverages the most updated references. For example, consider the case with degree $g = 1$ and a chain of $d$ needles located separately in different chunks. Then, the number of rounds will be $r_{\mathsf{par}} = d$ for the "parallel" version, but can be as small as $r_{\mathsf{cyc}} = 1$ for the "cyclic" version, if the needles happen to appear in the right order in the original clues.

- If the cost metric of interest is the end-to-end latency with parallelism degree $p$, then the $k$ LLM calls for retrieval within each round can be parallelized, which implies

$$\mathcal{C} \leq r_{\mathsf{par}} \times \left( \lceil \frac{k}{p} \rceil \times \mathcal{C}(\text{retrieval}) + \mathcal{C}(\text{reasoning}) \right)$$

$$\leq r_{\mathsf{par}} \times \left( \lceil \frac{k}{p} \rceil \times \mathcal{C}_{\mathsf{LLM}}(\mathcal{L}_{\mathsf{sys}} + m + \ell, \ell) + \mathcal{C}_{\mathsf{LLM}}(\mathcal{L}_{\mathsf{sys}} + \ell, 1 \text{ or } \ell) \right). \tag{10}$$

As a result, the end-to-end latency of the "parallel" version with a sufficiently large parallelism degree $p$ can be potentially smaller than that of the "cyclic" version.

### C.3.3 Insights

In sum, we derive two major insights from the above analysis:

1. LLM nodes responsible for reasoning and answering only occupy a small fraction of the costs of the overall algorithm, while playing a critical role in the output of the algorithm and thus in the error metrics. Therefore, our general recommendation is to prompt these LLM calls to think step by step before answering (instead of answering directly), which will most likely boost the accuracy significantly with negligible loss in efficiency.

2. Regarding the "cyclic" and "parallel" options for retrieval, we conclude that each of them has its own pros and cons. With the "cyclic" option, fewer rounds of retrieval and reasoning are needed (thanks to the timely updates to the references), but the downside is that all retrieval steps have to be executed sequentially. In contrast, with the "parallel" option, the algorithm typically requires more rounds of retrieval and reasoning and thus larger costs in terms of most metrics, but can leverage parallelism of LLM calls for reducing the end-to-end latency. Therefore, whether the "cyclic" or "parallel" option is preferred largely depends on which cost metric is of more concern.

---

[6]There might exist more precise characterizations of $\mathcal{L}_{\mathsf{dec}}$ in this case, e.g., $\mathcal{L}_{\mathsf{dec}} \lesssim \ell \times g$ if the LLM tends to take $g$ steps for calculating the sum of $g$ numbers. For simplicity, we mostly assume $g = O(1)$ in this case study, and thus stick to the assumption that $\mathcal{L}_{\mathsf{dec}} \lesssim \ell$.

### C.4 Experiments

We validate our analysis via experiments with `Qwen-2-72B-Instruct` (Yang et al., 2024), supported by vLLM (Kwon et al., 2023) and running on a server with 4 Nvidia A100-80G GPUs. For each experiment, we vary the depth, width or degree of the grid of variables (which controls the difficulty of task instances), while validating the derived insights by comparing two options for prompting LLM calls that are responsible for reasoning and answering, or comparing the "cyclic" and "parallel" options for retrieval.

Error metrics of interest include (1) the exact-match error, which takes value 0 if the answer returned by the algorithm matches the ground-truth solution exactly, and value 1 otherwise; (2) the absolute error, i.e., the absolute value of the difference between the algorithm's answer and the ground-truth solution; and (3) the missed-coverage error, i.e., the ratio of needles that the algorithm fails to retrieve and hence are not included in the references maintained by the algorithm. Note that missed coverage might arise not just from failures of the LLM calls responsible for retrieval, but also from the possibility that the algorithm terminates too early with fewer rounds than necessary, due to hallucination by the LLM calls responsible for reasoning and answering. Cost metrics of interest are the same as those investigated in previous examples, plus the number of rounds, a metric specific to this case study.

*Remark* 4 (Mitigating hallucination in retrieval). During our early experiments, we observed that LLMs tend to make up clues not present in the input during the retrieval steps. To address this, we prompt the LLM to return exact copies of clues from the original text, and further use a symbolic program to check whether the retrieved clues are indeed exact copies (which, from the perspective of LLM-powered agent systems, might be regarded as tool use). Only those passing the test will be added to the list of references. It has been confirmed empirically that this simple method effectively mitigates the errors caused by hallucination of LLMs during retrieval, making the overall algorithm much more robust and accurate in our experiments.

**Comparing two prompting options.** For this experiment, we consider shallow and wide grids of variables, with depth $d = 2$ and width $w = 300$. The degree $g$ varies, controlling the difficulty in arithmetic. The algorithm uses a fixed chunk size of 50 clues, and the "parallel" version of retrieval. In this case, the number of rounds should be 2 if each retrieval step succeeds.

Empirical results for this experiment are illustrated in Figure 17, which confirm our first insight, i.e., prompting the LLM to think step by step for reasoning and answering significantly boosts the accuracy of the overall algorithm while incurring minor overhead in cost metrics, compared to answering directly. Note from the last row, though, that the relative difference in terms of end-to-end latencies increases with the parallelism degree $p$, which has been predicted by our analysis, in particular Eq. (10).

**Comparing the "cyclic" and "parallel" versions.** In the following experiments, we consider grids with fixed degree $g = 1$ and thus chains of needles. We either fix the depth $d = 5$ and vary the width $w$, or fix the width $w = 100$ and vary the depth $d$. The algorithm uses a fixed chunk size of 100 clues, and prompts the LLM calls responsible for reasoning and answering to think step by step.

Empirical results for these two experiments can be found in Figures 18 and 19, which confirm that the considered algorithm achieves high accuracy for a wide range of task configurations. The second insight from our analysis is also validated: compared with the "cyclic" version, the "parallel" version incurs a larger number of rounds and hence higher cost metrics, except for the end-to-end latencies with parallelism, for which the benefits of parallelism outweigh the downside of requiring more rounds.

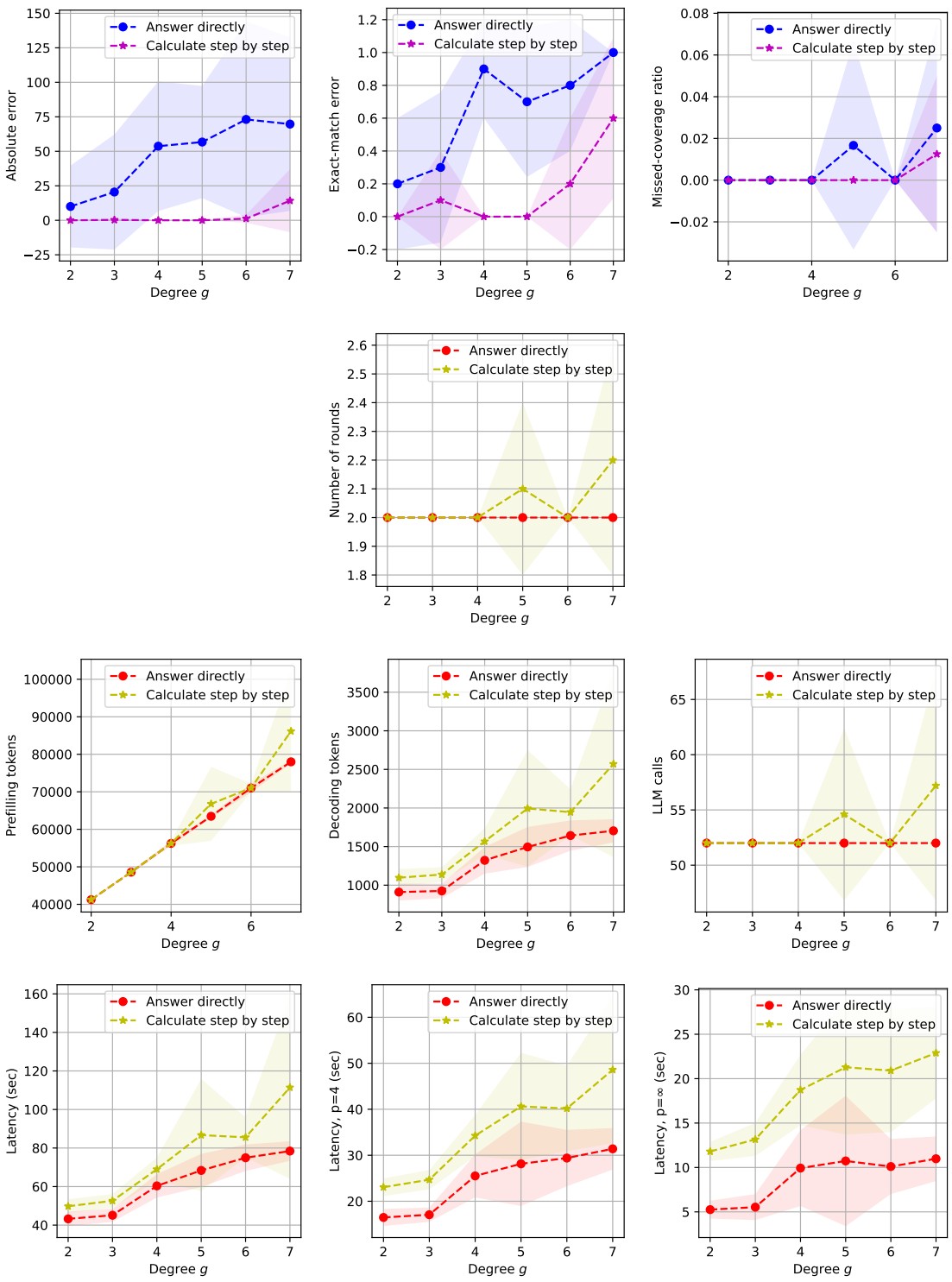

Figure 17: Empirical results (10 trials) for reasoning with iterative retrieval, where two options of prompting the LLM calls responsible for reasoning and answering are compared. The depth $d = 2$ and width $w = 300$ are fixed, while the degree $g$ varies. The algorithm uses a chunk size of 50 clues, and the "parallel" version of retrieval.

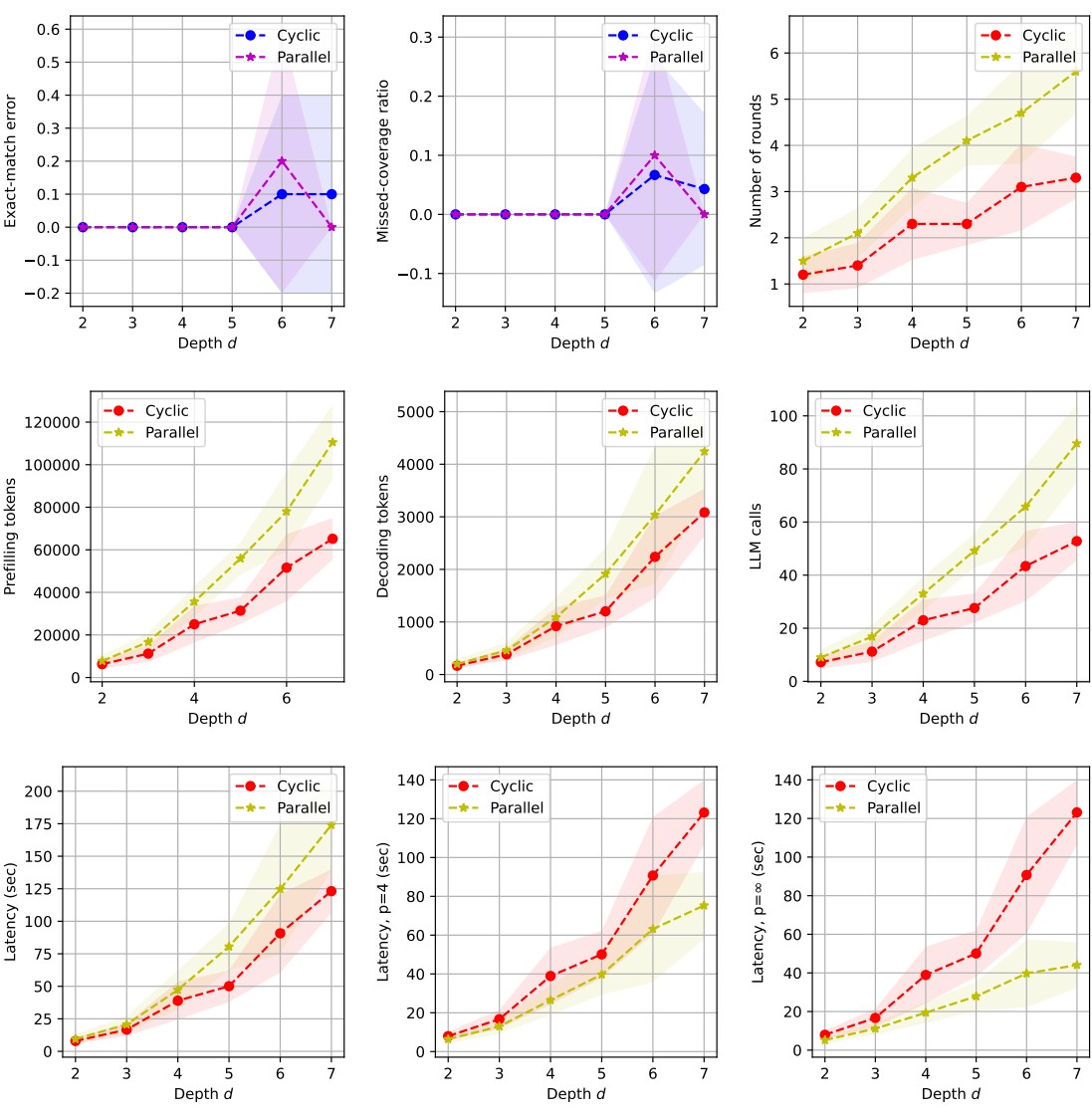

Figure 18: Empirical results (10 trials) for reasoning with iterative retrieval, where two options of retrieval are compared. The degree $g = 1$ and width $w = 100$ are fixed, while the depth $d$ varies. The algorithm uses a chunk size of 100 clues, and prompts the LLM calls responsible for reasoning and answering to think step by step.

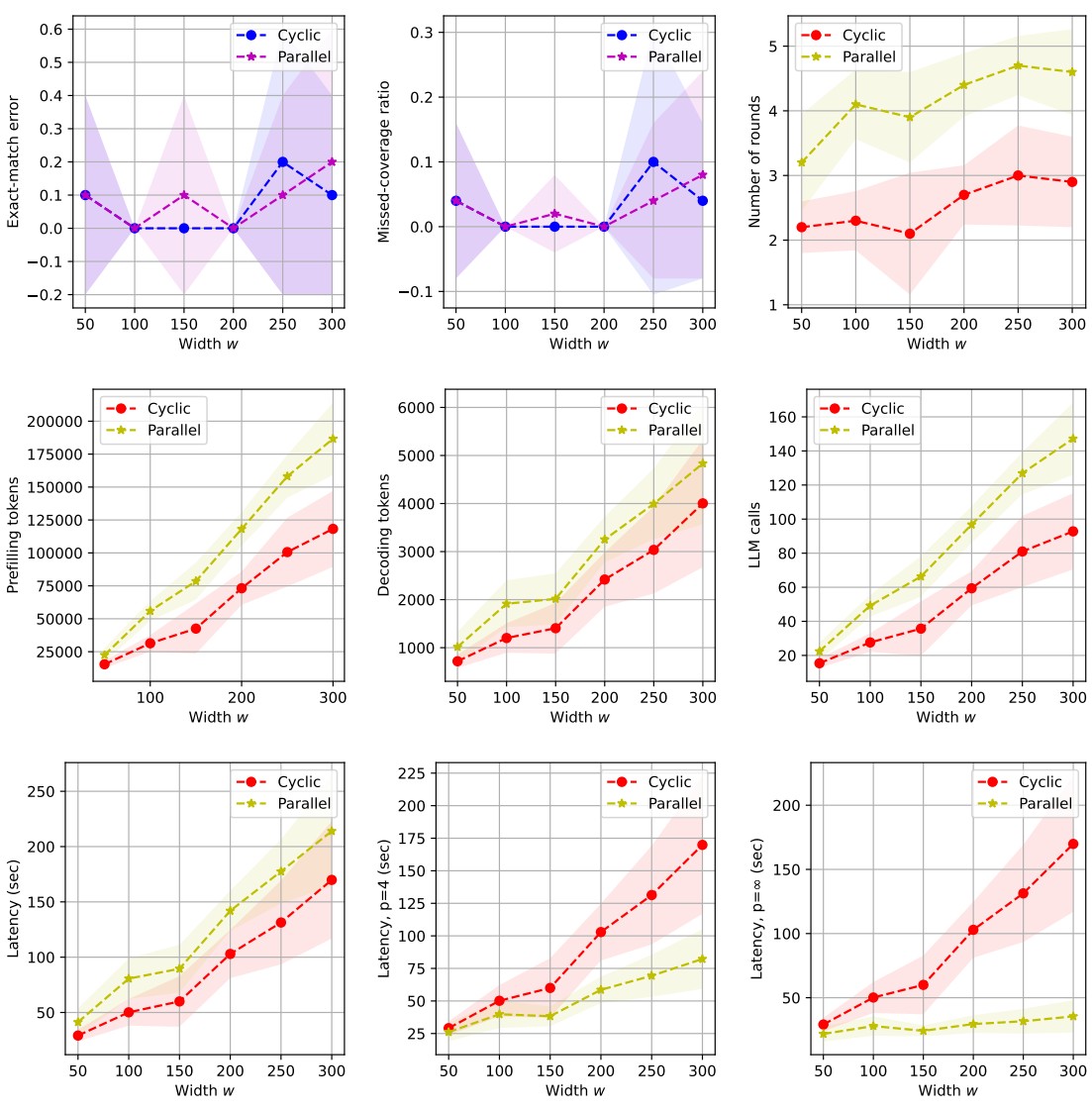

Figure 19: Empirical results (10 trials) for reasoning with iterative retrieval, where two options of retrieval, namely "cyclic" and "parallel", are compared. The degree $g = 1$ and depth $d = 5$ are fixed, while the width $w$ varies. The algorithm uses a chunk size of 100 clues, and prompts the LLM calls responsible for reasoning and answering to think step by step.

# D    Supplementary materials for Section 4.3

Recursive LLM-based algorithms have been widely applied in prior works, e.g., in LAMBADA (Kazemi et al., 2023), LLM programs (Schlag et al., 2023), ADaPT (Prasad et al., 2024), THREAD (Schroeder et al., 2024), Recursion of Thought (Lee & Kim, 2023), Decomposed Prompting (Khot et al., 2023), ReDel (Zhu et al., 2024), among many others. This appendix includes full details for our case study, presented in Section 4.3, of recursive task decomposition. In the following, we first introduce the concrete task under consideration, and then design a recursive LLM-based algorithm for solving it, highlighting the dynamic construction of its computational graph at runtime. We then provide analysis for its accuracy and efficiency using the proposed framework, which is further validated with numerical experiments.

## D.1    Concrete setup

For concreteness, we consider the same task introduced in Section C.1 and visualized in Figure 16, which is about calculating a targeted variable based on clues about a grid of many variables.

One major difference here is that we consider much larger depth $d$, width $w$ and/or degree $g$ for the grid of variables, which implies a larger DAG of needles, i.e., clues relevant to the targeted question. Even if all relevant clues are given *a priori*, the complex reasoning required to answer the targeted question correctly can be well beyond the capability of one single LLM call, which motivates decomposing the reasoning process into multiple LLM calls.

The other major difference is, we assume that the clues are not directly given in plain text, but rather need to be accessed via querying a database (or more generally, a source of information). For each query, the database takes a name as input, and returns a clue for the variable of the same name, e.g., "A2 = B1 + B3" for input "A2" or "C1 = 10" for input "C1". The LLM-based algorithm need to decide by itself what to query from the database. Such a setting is motivated by, and can be regarded as an abstraction of, practical scenarios where an autonomous agent in the wild need to actively retrieve relevant information by itself in order to accomplish a task (unlike in typical problem-solving benchmarks where such information is given), via querying a real database or knowledge graph, using a search engine, retrieving from documents, etc. In this case study, we assume that querying the database does not involve LLM calls, and neglect its costs in our analysis for simplicity.

## D.2    Algorithm

We consider the following recursive LLM-based algorithm for solving this task. The overall algorithm maintains a dictionary of variables that have been calculated throughout its execution, and eventually outputs the value returned by applying a function named `ProcessNode` to the targeted variable.

The function `ProcessNode`, which takes the name of a variable as input and returns its numeric value, is defined in a recursive manner:

1. If the input variable is found in the dictionary of calculated variables, then return its value directly.

2. Otherwise, query the database with the input variable, and invoke one LLM call with the returned clue, which is prompted to either answer with a numeric value or conduct further task decomposition, i.e., identifying variables whose values will be necessary and sufficient for calculating the input variable.

   (a) If the LLM chooses to answer with a numeric value, then return it.
   (b) Otherwise, invoke `ProcessNode` once for each variable identified by the LLM call, collect the answers returned by these children tasks, and finally invoke one LLM call to calculate and return the numeric value of the input variable.

We make a few comments about this recursive algorithm. (1) The computational graph of this algorithm, or a generic LLM-based algorithm following the pattern of recursive decomposition, is constructed dynamically

in a depth-first-search style at runtime. See Figure 7 for a visualization. The final graph is symmetric about the nodes corresponding to the leaf variables (D, E); on the left are LLM calls for identifying children variables for the non-leaf variables (A, B, C), while on the right are LLM calls for calculating their numeric values. (2) The process of complex reasoning is decomposed recursively, so that each LLM call within this algorithm only need to handle a small step of reasoning, whose complexity is irrelevant to that of the overall task. (3) One particular configuration of this algorithm, like in the previous case study, is about how the LLM calls (especially the ones responsible for calculation) are prompted, e.g., to answer directly or think step by step. The impact of this design choice on the accuracy and efficiency of the overall algorithm will be investigated analytically and empirically soon in this section.

### D.3 Error analysis

Recall from Eq. (1) that for a specific node $v$ with inputs $x_1, \ldots, x_k$, the error of its output $y$ is assumed to be upper bounded by $\mathcal{E}(y) \leq f_v(\mathcal{E}(x_1), \mathcal{E}(x_2), \ldots, \mathcal{E}(x_k))$ for some function $f_v$. For the concrete setting of the current case study, there are two major failure modes for the aforementioned algorithm: errors in recursive task decomposition, and errors in calculating the numeric values of specific variables. Empirical results during our early exploration suggested that LLMs make very few mistakes of the first kind, while the second failure mode is much more common, due to the limited arithmetic capabilities of the LLMs under consideration. Moreover, all mathematical operations considered in this concrete settings (including addition, substraction, maximum and minimum) are 1-Lipschitz continuous with respect to each input variable. Hence Proposition 2, which provides error bounds under the assumption of additive errors and bounded sensitivity, can be adopted for this case study, although we omit the details here for brevity.

### D.4 Cost analysis

Let us first consider the cost of each LLM call. For each leaf variable, one LLM call is needed for finding its numeric value based on its corresponding clue of length $O(1)$, which implies $\mathcal{L}_{\mathsf{pre}} \leq \mathcal{L}_{\mathsf{sys}} + O(1)$ and $\mathcal{L}_{\mathsf{dec}} \leq O(1)$, and thus $\mathcal{C}(\mathsf{leaf}) \leq \mathcal{C}_{\mathsf{LLM}}(\mathcal{L}_{\mathsf{sys}} + 1, 1)$. For each non-leaf variable, two LLM calls are needed. One of them is responsible for task decomposition, i.e., identifying its $g$ children variables, which implies $\mathcal{L}_{\mathsf{pre}} \leq \mathcal{L}_{\mathsf{sys}} + O(g)$ and $\mathcal{L}_{\mathsf{dec}} \leq O(g)$, and thus $\mathcal{C}(\mathsf{decomposition}) \leq \mathcal{C}_{\mathsf{LLM}}(\mathcal{L}_{\mathsf{sys}} + g, g)$. The other LLM call is responsible for calculating the numeric value of the current variable based on the values of its children variables, which has $\mathcal{L}_{\mathsf{pre}} \leq \mathcal{L}_{\mathsf{sys}} + O(g)$, and $\mathcal{L}_{\mathsf{dec}} \leq O(1)$ if the LLM is prompted to answer directly, or $\mathcal{L}_{\mathsf{dec}} \leq \mathcal{L}_{\mathsf{calc}}(g)$ for some function $\mathcal{L}_{\mathsf{calc}}$ if it is prompted to do the calculation step by step (e.g., $\mathcal{L}_{\mathsf{calc}}(g) \leq O(g)$ or $O(g^2)$, depending on how the LLM executes the calculation with $g$ variables). In sum, we have $\mathcal{C}(\mathsf{calculation}) \leq \mathcal{C}_{\mathsf{LLM}}(\mathcal{L}_{\mathsf{sys}} + g, 1 \text{ or } \mathcal{L}_{\mathsf{calc}}(g))$ for each LLM call responsible for calculation.

The analysis above has also revealed the total number of LLM calls. Let $n$ and $n_{\mathsf{leaf}}$ be the number of relevant variables and the number of relevant leaf variables, respectively. These parameters are determined by $d, w, g$ and other factors, as is the case for the $\ell$ parameter (total length of relevant clues) in Section C. Recall from above that each leaf variable requires one LLM call for finding its value from its corresponding clue, while each non-leaf variable requires two LLM calls, one for task decomposition and one for calculation. Assuming that task decomposition is done correctly by the LLM for each non-leaf variable, the total number of LLM calls will be $2 \times (n - n_{\mathsf{leaf}}) + n_{\mathsf{leaf}} = 2 \times n - n_{\mathsf{leaf}}$.

Putting things together, we finally arrive at the total cost of the overall recursive algorithm:

$$\mathcal{C} \leq (n - n_{\mathsf{leaf}}) \times \Big( \mathcal{C}_{\mathsf{LLM}}\big(\mathcal{L}_{\mathsf{sys}} + g, g\big) + \mathcal{C}_{\mathsf{LLM}}\big(\mathcal{L}_{\mathsf{sys}} + g, 1 \text{ or } \mathcal{L}_{\mathsf{calc}}(g)\big) \Big) + n_{\mathsf{leaf}} \times \mathcal{C}_{\mathsf{LLM}}(\mathcal{L}_{\mathsf{sys}}, 1). \quad (11)$$

In particular, prompting the LLM to do the calculation step by step, which is anticipated to significantly improve the accuracy of the overall algorithm, will increase most cost metrics (though not for the number of LLM calls or the total number of prefilling tokens) by a *multiplicative* factor, in contrast to the previous case study in Section C where step-by-step prompting incurs a minor *additive* cost overhead.

### D.5 Experiments

We validate our analysis via experiments with `Qwen-2-72B-Instruct` (Yang et al., 2024). For each experiment, we vary the task complexity while comparing two options of prompting the LLM calls responsible for calculation, namely "answer directly" and "calculate step by step". Empirical results are shown in Figures 20 and 21. For the former, we fix the width $w = 6$ and degree $g = 4$, while varying the depth $d$; for the latter, we fix $d = 3$ and $w = 100$, while varying $g$. The results confirm our previous analysis: compared to "answer directly", prompting the LLM calls to "calculate step by step" significantly boosts the accuracy of the overall algorithm and leads to satisfactory performance for a wide range of task configurations, while incurring higher costs in the latency and total number of decoding tokens by a multiplicative factor, and making no difference in the number of LLM calls or total number of prefilling tokens.

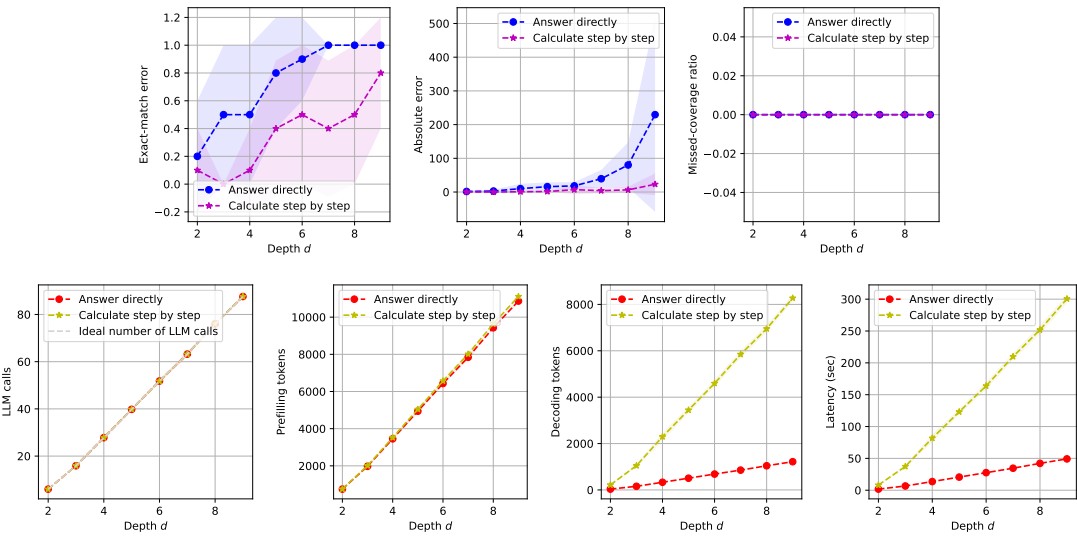

Figure 20: Empirical results (10 trials) for reasoning with recursive task decomposition, where two options of prompting are compared. The width $w = 6$ and degree $g = 4$ are fixed, while the depth $d$ varies.

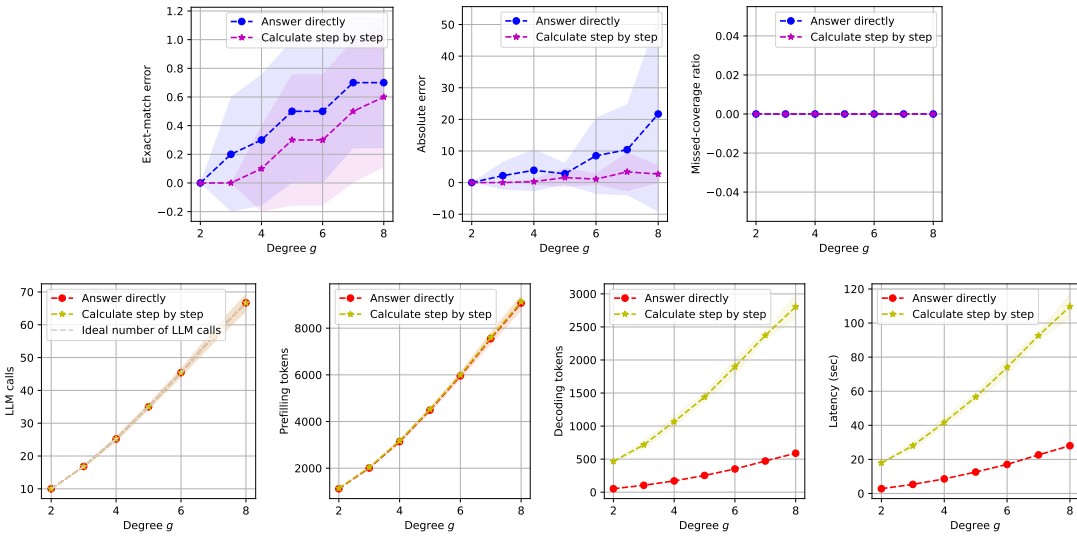

Figure 21: Empirical results (10 trials) for reasoning with recursive task decomposition, where two options of prompting are compared. The depth $d = 3$ and width $w = 100$ are fixed, while the degree $g$ varies.

