# OpenReview forum: "Designing Algorithms Empowered by Language Models: An Analytical Framework, Case Studies, and Insights"
_TMLR — Accepted by TMLR_

### Review · Reviewer_rZ8M · 2025-08-01

**Summary Of Contributions:**

The paper proposes a framework for analyzing "LLM-based algorithms," an algorithm that leverages LLM calls as a subroutine to conduct computation. The work formalizes task decomposition via LLMs into computational DAGs, bounds on errors given LLM failures, and provides theoretical and empirical evidence in support of the framework.

**Audience:**

Yes

**Claims And Evidence:**

No

**Requested Changes:**

* My main concern is whether the theoretical tools provided have added value given pre-deep learning era work on distributed computing; i.e., is there a reason why handing off subtasks to LLMs is not "merely" an instance of MapReduce where nodes could provide noisy answers? Why are analytical approaches in past analysis of parallel distributed algorithms (e.g., Byzantine agreement methods [1]) not applicable here?
    * At a minimum, I'd like to see further discussion of such approaches in the related work.
* Here's a couple of suggestions for how to improve the clarity of presentation:
    * A worked-example on a simple task (e.g., counting) near Eq. (4), maybe as a motivating example.
    * Comparison to standard time- and memory-complexity analysis of algorithms, showing how one would draw different conclusions. I get that the "units" of analysis are different in the proposed approach but showing this comparison would really emphasize the value added by the proposed approach.

[1] Pease, Marshall, Robert Shostak, and Leslie Lamport. "Reaching agreement in the presence of faults." Journal of the ACM (JACM) 27.2 (1980): 228-234.

**Strengths And Weaknesses:**

## Strengths
* The paper chooses a salient topic for analysis: the usage of LLMs as subroutines for solving larger tasks. While the paper largely focuses on toy settings and problems, this helps the paper build a solid foundation for this type of analysis.
* The paper provides empirical validation for many of its theoretical claims.

## Weaknesses
* The novelty of the approach is unclear. The analysis of parallel decompositions feels very similar to that of parallel computation algorithms such as MapReduce [1] or Byzantine fault tolerance — except the compute nodes ("map" stage) are LLMs and we've replaced some deterministic algorithm with arbitrary time/memory complexity with an LLM (i.e., something non-deterministic w/ some non-certain success probability). In Table 1, Step 2 is essentially "map" and Step 3 is essentially "reduce."
    * At the very least, the paper would be significantly strengthened by discussion of the proposed framework in the context the literature in distributed computing. Textbooks on distributed computing [2] could be a good starting point for further literature review to discuss (1) connections to previously-known results, and (2) how the LLM-based setting introduces novel challenges.
    * I think there's a subtlety that "not all tokens cost the same" (e.g., prefill vs. decoding phases) but it's not clear to me how this changes the analysis.
* The clarity of the examples could be improved — counting and sorting are straightforward tasks, and it would help to see worked examples of these tasks to build further intuition.
* Section 4.2 feels out of place — it's not clear to me why comparing CoT to the default prompting strategy used is an instance where the proposed analytical framework is useful.
* Minor point: The nomenclature of Type I/Type II LLMs in 3.2.3 is a little confusing given Type I/II errors in hypothesis testing (e.g., I see that Type II hallucinates false positives = rejecting the null incorrectly = Type I?) — maybe there's a descriptive name that could be used instead?

[1] Karloff, Howard, Siddharth Suri, and Sergei Vassilvitskii. "A model of computation for mapreduce." Proceedings of the twenty-first annual ACM-SIAM symposium on Discrete Algorithms. Society for Industrial and Applied Mathematics, 2010.
[2] Lynch, Nancy A. Distributed algorithms. Elsevier, 1996.

---

> ### Author Response · Authors · 2025-08-15
> **Response to Reviewer rZ8M (1/3)**
>
> Thank you for your detailed and constructive review of our work!
> We address your concerns and questions in the following.
>
>
> ---
>
> **Question: comparisons with the literature of distributed computing and standard time/memory complexity analysis.**
>
> Thank you for pointing us to these directions of literature, and suggesting that discussing our work in such context could strengthen our paper and better clarify our actual contributions.
>
> We agree that some elements in our work (e.g., the parallel decomposition pattern) closely resemble those from other literature (e.g., MapReduce), and we certainly do not intend to claim novelty about them.
> From the authors' perspective, the novelty and values of our work is mainly about proposing an analytical framework that *allows formal and rigorous analysis of LLM-based algorithms*, and presenting case studies that demonstrate how one can *derive useful insights with this framework* that provide practical guidance for LLM applications.
>
>
> To further explain this, let us consider a generic process of problem solving:
> 1. Identify the practical problem of interest;
> 2. Turn the problem into a mathematical model, via appropriate abstractions and assumptions;
> 3. Decide what theoretical tools to use, and develop new ones if needed;
> 4. Apply the theoretical tools to derive analytical results that can provide valuable insights for the original practical problem.
>
> This process indeed resembles the overview of our proposed framework presented in Figure 1.
> In our case, the practical problem of interest in Step 1 is to "provide formal analysis for LLM-based algorithms".
>
> Our main novelty and contributions are about *Steps 2 and 4* of this process:
>
> * Step 2 corresponds to Section 2 where we introduce the analytical framework.
> While LLM-based algorithms are capable of solving problems that traditional algorithms cannot handle (thanks to the general-purpose problem solving capabilities of LLMs),
> practitioners usually have to rely on trial-and-errors to understand and tune their performance,
> as the black-box nature of LLMs makes it seem challenging (if not impossible) to conduct any formal analysis.
> Our framework, on the other hand, shows that it is indeed possible to *bridge LLM-based algorithms with symbolic programs and mathematical models* that facilitate rigorous analysis,
> via identifying the task decomposition principle and computational graph representation,
> and appropriate abstractions of LLM's capabilities and cost models.
> In particular, the *diverse and unique characterization of LLMs* (e.g., relation between LLM's output error and input size, the impact of chain-of-thought prompting on LLM's behavior, using token as a unit of complexity measure since LLMs are next-token-prediction machines) need to be taken care of, in order to eventually arrive at theoretical results with practical relevance.
> * Step 4 corresponds to our case studies in Sections 3 and 4 (as well as the appendix),
> where we show the framework in action and derive non-trivial (and sometimes counter-intuitive) insights for LLM-based algorithms, with extensive empirical validation.
> * Step 3, on the other hand, is not the main focus of our contributions.
> The theoretical tools used in this work are relatively basic, compared to the papers mentioned in your review, e.g., computation model of MapReduce, or Byzantine agreement methods.
> The focus and purpose of analysis are also different here; the relevance of theoretical tools in the mentioned prior works to our case does not seem obvious.
> We have developed some small theoretical results as needed along the way though, e.g., Proposition 1 for the error bound of merging multiple approximately sorted sequences.
> Please let us know if any of these results has appeared in prior literature; we will certainly cite the related work and avoid making misclaims about novelty.
>
>
> We hope that our response above has better clarify the novelty and values of our work when compared to the literature of distributed computing and standard complexity analysis.
> We plan to add a new paragraph in the "related works" section, to include the above discussions and add citations accordingly.
> We are committed to ensuring that our work meets the "claims and evidence" criterion of TMLR, and in particular, that we do not make misclaims about novelty in any part of our results.

---

> ### Author Response · Authors · 2025-08-15
> **Response to Reviewer rZ8M (2/3)**
>
> **Suggestion: a worked-example on a simple task (e.g., counting)**
>
> Thank you for the suggestion. We will add a worked example in our revision.
>
> Below is a concrete example for the counting task,
> showing the prompt and response of an LLM call for solving a sub-task,
> i.e., counting the number of digits in a segment of the original input string.
> The segment "dhd73695051jcbcg9jbeh903bi64ic736icfag96" is inserted into the prompt template,
> and the answer extracted from the LLM's response is 19.
>
>     User: Count the number of digits in the following string:
>     dhd73695051jcbcg9jbeh903bi64ic736icfag96
>     Answer in the following format: 'There are {answer} digits in the given string'. Do NOT output anything else.
>
>     Assistant: There are 19 digits in the given string
>
>
> ---
>
> **Weakness: Section 4.2 feels out of place — it's not clear to me why comparing CoT to the default prompting strategy used is an instance where the proposed analytical framework is useful.**
>
> For the case of Section 4.2, the proposed framework is useful for *analytically quantifying* how much cost overhead will be incurred by CoT prompting compared to the default prompting strategy.
>
> It is quite intuitive that CoT prompting for the reasoning nodes can improve accuracy while incurring a higher cost of the overall iterative-retrieval-and-reasoning algorithm.
> However, without formal analysis, it remains unclear how much cost overhead exactly will be incurred, and whether it is worthwhile for the accuracy gain.
> Will the overall cost increase by 10% or 300%?
> How does it change with the input size $n$, chunk size $m$, or parallel degree $p$?
> Our framework provides precise answers, i.e., Eq. (9) and (10) in Appendix C (as mentioned in Insight 4), to such questions,
> confirming that CoT prompting will incur only *small additive cost overhead* for this iterative-retrieval-and-reasoning algorithm.
>
> Similarly, for the recursive algorithm from the case study in Section 4.3,
> we prove with the proposed framework that CoT prompting will incur a *multiplicative cost overhead*,
> which is precisely characterized by Eq. (11) in Appendix D (as mentioned in Insight 5).
>
> These two case studies demonstrate that the impact of CoT prompting on costs can vary significantly across tasks and algorithms.
> With the proposed framework, one may gain a much better understanding of how much cost overhead will be incurred in a specific scenario of interest.
> Answering such questions is indeed one of the motivations behind our proposed framework,
> as explained in the "goal and motivations" paragraph (3rd bullet point) of Section 1,
> and the "LLM test-time scaling" paragraph of Section 5.

---

> ### Author Response · Authors · 2025-08-15
> **Response to Reviewer rZ8M (3/3)**
>
> **Minor point: The nomenclature of Type I/Type II LLMs in 3.2.3 is a little confusing given Type I/II errors in hypothesis testing**
>
> Thank you for pointing this out.
> We plan to replace "Type 1/2" with "Type A/B" in our revision at the moment, until we can think of a better and more descriptive name.
>
>
> ---
>
> **Question: I think there's a subtlety that "not all tokens cost the same" (e.g., prefill vs. decoding phases) but it's not clear to me how this changes the analysis.**
>
> Thank you for pointing this out.
> Most of our current cost analysis are indeed not impacted much by the distinction between prefilling and decoding tokens.
> We will simplify the notations in our revision by replacing $C_{pre}(L_{pre}) + C_{dec}(L_{pre}, L_{dec})$ with $C_{LLM}(L_{pre}, L_{dec})$ whenever feasible, e.g., in Section 3.1.
>
> Our emphasis on distinguishing the costs of prefilling / decoding phases was initially motivated by real-world LLM applications.
> The computational process of LLM inference exhibits notable differences between the prefilling and decoding phases.
> Similarly, for API calls of proprietary LLMs, monetary cost per decoding token is typically much higher than cost per prefilling token.
> Future work might improve upon our current cost analysis by taking this distinction into account for finer-grained analysis.
>
> ---
>
> Thank you again for your thoughtful review, which helps us a lot in improving this work!
> Our plan for the upcoming week is to revise the manuscript accordingly.
> In the meantime, we are always open to discussions if you have any further concern or suggestion :)

---

> > ### Comment · Reviewer_rZ8M · 2025-08-18
> >
> > Thanks! This is all really helpful. I think, then, my main concern lies in the responses "Step 2 of the problem-solving process" and whether the proposed framework is sufficiently novel, or an extension of the previous frameworks I've mentioned, merely replacing "compute node" with "LLM."
> >
> > While the approaches I mentioned might not explicitly use the verbiage of computation graphs, they still propose some step-by-step model of computation. My worry is that the type of rigorous analysis proposed ends up being an extension of previous approaches. To make this more clear, perhaps we can use the "count # of letters" example here and try to pattern match the MapReduce example in Sec. 1 of Karloff et al. — I'll use the "Step" indices from their paper:
> >
> > * Mapping — I chunk the string into tuples representing subtasks:
> >     * (0, dhd7),
> >     * (1, 3695),
> >     * (2, 051j),
> >     * (3, cbcg),
> >       ...
> >     * (9, ag96)
> > * In their Step 1, tuples are grouped by key. Since our "items" (substrings) can only appear once, this is just a no-op.
> > * In Step 2, I count the letters in each subtask, yielding tuples of the form {(dhd7, 1), (3695, 4), ... (ag96, 2)}.
> > * Steps 3-4 follow identically — map all tuples to the same key {($, 1), ... ($, 2)} and sum -> ($, 19).
> >
> > Ultimately, the prior work's proposed framework + definition for a "mapper" and a "reducer" subsumes a lot of these functions. Now, I *think* the main difference/potential source of novelty in this manuscript is the potential for these mapping steps, which may be delegated to an LLM, to return an incorrect response. I see this a little bit in the error analysis section (3.2), but it's focused on analysis of specific tasks such as counting, sorting, and retrieval. In retrospect this is much more related to MapReduce than Byzantine fault tolerance — the connection for the latter is merely "coordination across multiple processes."
> >
> > Could the authors further comment on the distinction here? My ask isn't for the same theoretical/mathematical tools to be used here as in the prior work, but for an explanation for why the two frameworks — MapReduce based computation + the proposed LLM-based computation — differ.

---

> > > ### Author Response · Authors · 2025-08-19
> > >
> > > Thanks for your reply, and in particular, for elaborating on your concern about the novelty of this work.
> > >
> > > Let us answer your question, i.e., **"why the two frameworks — MapReduce based computation + the proposed LLM-based computation — differ"**, in the following:
> > >
> > > * Parallel decomposition, or the MapReduce pattern, is only a special case for this work.
> > > Our proposed framework is targeted at generic LLM-based algorithms, and several case studies beyond parallel decomposition have been investigated in Section 4 and the corresponding appendices.
> > > Perhaps looking at the *didactic* counting/sorting example alone could give the impression that we're just using LLMs to solve standard tasks that classical MapReduce (or other symbolic programs) can solve, but this is obviously not true, as can be seen from other case studies in this work.
> > > * We do not claim novelty about task decomposition, computational graphs, or the MapReduce pattern.
> > > After all, these are all foundational and basic elements in computer science.
> > > * The main point of our proposed framework is really about (1) identifying the relevance of these elements to LLM-based algorithms, and (2) integrating LLM-based algorithms with these elements and other math/theoretical tools (e.g., those originally developed for symbolic programs and "classical"/non-LLM algorithms), via appropriate abstractions and characterization of black-box LLMs, so that formal analysis for their accuracy and efficiency is now possible.
> > > The unique features of LLMs, and their implications, distinguish our framework from MapReduce computation model or other related frameworks in the literature.
> > >
> > >
> > > Regarding the last bullet point,
> > > we refrain from making a claim about how novel or significant it should be, since it would be merely our own subjective opinion.
> > > Our goal in this work is just to provide something useful and interesting for the community, e.g., the SOP illustrated in Figure 1: given an LLM-based algorithm of interest, formulate it as a computational graph -> characterize the LLM's error and cost functions for solving a sub-task -> analyzing the error and cost of the overall algorithm (by leveraging the graph topology) -> come up with concise insights for the algorithm.
> > > LLM practitioners might use this work as a handbook and follow the SOP for formal analysis of their own LLM applications, which can potentially lead to useful insights and reduce the trial-and-errors needed.
> > >
> > >
> > > Ultimately, if we agree that
> > >
> > > 1. the proposed framework has led to non-trivial and empirically validated insights for important questions about LLM-based algorithms (e.g., those in the "goal and motivations" paragraph of introduction), and
> > > 2. such questions cannot be answered by previous frameworks (after all, how can the MapReduce model say anything about the impacts of chain-of-thought prompting, or the distinction between Type-A vs Type-B LLM and its implications in a retrieval task?),
> > >
> > > then we may also agree that there are indeed some unique values in this work.
> > >
> > > ---
> > >
> > > Thank you again for the thought-provoking question;
> > > we hope that our response has addressed your main concern.
> > > Please let us know if you have further comments!

---

> ### Comment · Reviewer_rZ8M · 2025-08-28
>
> Great, I think this clears up some of my misunderstandings. I think as-is, especially since the paper leads with these figures, it's easy to read this work as focusing on the graph decomposition itself and conclude that this is just some tweak to parallel decomposition paradigms in distributed computing. Reframing the work to lead with unique characteristics of LLMs, before motivating "look, XYZ is similar to ABC 'old' techniques from algo analysis/design" might help mitigate such issues, and I think I would've come around sooner.
>
> On the claimed contributions in the response "(1) identifying the relevance of these elements to LLM-based algorithms, and (2) integrating LLM-based algorithms with these elements and other math/theoretical tools (e.g., those originally developed for symbolic programs and "classical"/non-LLM algorithms)" — sure, I think I can agree that the paper demonstrates this.
>
> It's not clear to me whether this jump — showing previous frameworks can be applied to LLM-based algorithms — represents a significant theoretical extension to the body of work that already exists in algorithmic analysis. However, since this is a subjective criterion, and I think there's independent value in showing that existing frameworks can be applied to "modern/current" problem settings, I don't consider this to be a reason to reject. Ultimately I agree that there are potentially interesting findings for the community here.

---

> > ### Author Response · Authors · 2025-09-01
> >
> > Thank you for the update!
> > We are glad to know that some initial misunderstandings have been cleared up,
> > and appreciate your acknowledgement of the contributions and findings in this work.

---

### Review · Reviewer_Bz3T · 2025-08-05

**Summary Of Contributions:**

This paper introduces a formal analytical framework for designing and analyzing algorithms that embed one or more large language model (LLM) calls as core subroutines, named  LLM-based algorithms. By modeling algorithms as computational graphs, this work quantifies task decomposition and captures key abstractions like LLM characteristics and inference costs. Through this framework, the authors derive theoretical relationships between decomposition granularity, algorithmic accuracy, and computational efficiency. Through detailed case studies, the paper provides generalizable insights across a range of design patterns, including parallel, hierarchical, and recursive decomposition, and addresses trade-offs involving chain-of-thought reasoning and iterative retrieval.

**Audience:**

Yes

**Claims And Evidence:**

Yes

**Requested Changes:**

1. Key assumptions in the theoretical framework require stronger empirical grounding or citations to establish their relevance. For example:
   - Section 3.1: The claim that the Transformer model with full attention suffers from quadratic complexity in long-sequence processing.
   - Section 4.1: The use of simplified quadratic or linear cost models for token generation, and assumptions regarding bounded error sensitivity.
2. Expand the case study, or at least, discuss how the framework could be extended to support more complex agentic workflows.

**Strengths And Weaknesses:**

**Strengths**
1. This paper addresses a fundamentally under-explored problem with a formal framework. Rather than purely empirical benchmarking or heuristic-driven system design, it presents an analytical approach for LLM-based algorithms.
2. The proposed use of computational graphs to model LLM-based algorithms enables the natural integration of both symbolic and LLM-invoked subroutines within a unified representation.
3. The paper's insights are validated through both theoretical analysis and empirical case study.

**Weaknesses**

1. The case studies focus on highly synthetic tasks that abstract away many of the complexities present in real-world agentic workflows. For example, the retrieval task (Section 3.2.3) simplifies multi-document question answering into chunked retrieval followed by majority voting. It remains unclear how well the proposed framework scales to dynamic agent environments or tool-augmented workflows commonly used in real-world LLM systems.
2. The insights are either trivial or closely aligned with intuition. Some key findings—such as "finer-grained task decomposition leads to better accuracy but higher costs"—are expected and may not offer significant new understanding.
3. The analysis builds upon several assumptions, including simplified quadratic or linear cost models for token generation and bounded error sensitivity (Assumption 1 in Section 4.1), without sufficient justification or supporting citations.

---

> ### Author Response · Authors · 2025-08-15
> **Response to Reviewer Bz3T (1/3)**
>
> Thank you for your detailed and constructive review of our work!
> We address your concerns and questions in the following.
>
> ---
>
> **Question: key assumptions in the theoretical framework require stronger empirical grounding or citations to establish their relevance.**
>
> Thank you for your suggestion.
> Let us further establish the relevance of the mentioned assumptions in the following,
> which will be incorporated into our revision.
>
> Regarding quadratic / linear cost models:
>
> * The claim "Transformer with full attention suffers from quadratic complexity in long-sequence processing" is backed by a FLOPS calculation for its inference process, and extensive empirical validation in the literature. We will cite (Yuan et al., 2024; Zhou et al., 2024) in Section 3.1 where we made this claim.
> * The linear cost model is not necessarily a simplification, but rather can be a realistic setting. For example, API calls to proprietary LLMs are typically charged by the number of prefilling and decoding tokens, in which case the monetary cost of an LLM call processing a size-$m$ input is indeed linear in $m$ (as long as $L_{dec} = O(m)$).
>
> Regarding "bounded error sensitivity" in Assumption 1:
>
> * To clarify, the main message that we try to convey in Section 4.1 and Insight 3 is that "we have the error bound in Proposition 2 *for scenarios where Assumption 1 holds true*".
> We certainly do not expect or claim that Assumption 1 should be universally valid for all LLM-based algorithms.
> These clarifications will be incorporated into Section 4.1 in our revision,
> to prevent confusion or misunderstanding regarding the intended scope of Assumption 1 and Insight 3.
> * This "bounded error sensitivity" property can be rigorously proved for a non-LLM node in some cases.
> For example, considering the aggregation node in the counting algorithm,
> it has been shown in Section 3.2.1 that its sensitivity parameter is $S = 1$ for absolute counting error,
> or $S = 1 / k$ for normalized counting error.
> As another example, considering the aggregation node in the sorting algorithm,
> Proposition 1 indicates that its output $\ell\_{\infty}$ error is bounded by $\mathcal{E} \le \max\\{\mathcal{E}\_1, \dots, \mathcal{E}\_k\\} \le \mathcal{E}\_1 + \dots + \mathcal{E}\_k$, which means its sensitivity parameter is $S = 1$.
> * For an LLM node, "bounded error sensitivity" is an assumption that we impose on the *stability* of LLM's behavior when solving a *sufficiently simple* (with respect to its capability) sub-task.
> One motivating example for this assumption is the recursive algorithm from our case study in Section 4.3.
> This algorithm involves many LLM calls that are asked to calculate a 1-Lipschitz function (such as sum or max) of several input variables.
> We have observed in our empirical study that the LLM, with CoT prompting, performs pretty well for such sub-tasks,
> behaving with sufficient stability and making small errors occasionally.
> It is this "bounded error sensitivity" property that makes the final error of the overall algorithm stays low and grows slowly (rather than blows up rapidly) with the task complexity and size of computational graph,
> as can be observed in Figures 20 and 21 in Appendix D.5.

---

> ### Author Response · Authors · 2025-08-15
> **Response to Reviewer Bz3T (2/3)**
>
> **Question: gap between this work and real-world LLM applications.**
>
> To address your concern on this aspect,
> we would like to clarify that the scope of this work is to make a first step towards formal and rigorous analysis of LLM-based algorithms.
> We do not intend to overclaim the scope or contributions of the current work,
> or the implications of our obtained results for the most complex real-world agentic workflows.
>
> For the purpose of building a solid foundation for rigorously understanding the accuracy and efficiency of LLM-based algorithms,
> we have chosen to stick with synthetic tasks, which can be regarded as abstract versions of practical LLM usage.
> As discussed in Appendix A.1, this brings various benefits, such as avoiding data contamination issues, facilitating unambiguous and precise measurement of accuracy, and allowing full transparency and control over task configurations.
>
> Applying the proposed framework to a more realistic task or complex agentic workflow is beyond the scope of our current focus in this work.
> Nonetheless, we believe that our framework has shown the potential of supporting more complex scenarios:
> * In addition to the examples presented in Section 3,
> more complex examples of parallel decomposition that we have investigated include retrieval-augmented generation (Appendix A.5) and long-text summarization with chunking (Appendix A.6).
> We have also made substantial efforts to extend our analysis to more complex and realistic settings,
> such as iterative retrieval and multi-hop reasoning over long documents (Section 4.2),
> and recursive algorithm for active information seeking and aggregation (Section 4.3);
> these settings and algorithms are much more dynamic than parallel decomposition,
> and resemble the settings in some prior empirical works mentioned at the beginning of Sections 4.2 and 4.3.
> * Investigation for simple algorithm patterns and basic tasks could be beneficial for studying more complex real-world LLM applications in future work.
> For example, [Anthropic's "building effective agents" blog](https://www.anthropic.com/engineering/building-effective-agents)
> highlights that the most successful LLM agents in industry are often built with *simple, composable patterns*.
> We anticipate that our analysis for simple patterns (like parallel decomposition) could be used as building blocks for analysis of more complex agentic workflows in the future.
> * The insights that we have derived in simple settings can be potentially useful in broader settings.
> For example, Insights 1 and 2 can, at the minimum, warn practitioners against the common misconception that "finer-grained task decomposition implies higher accuracy / higher total cost of the overall LLM-based algorithm", beyond the parallel decomposition setting where these insights were originally derived.
>
> It would be intriguing future work to expand the proposed framework and validate its usefulness,
> by taking into account some practical elements (e.g., the costs of tool calls in tool-augmented workflows) that we might have abstracted away in the current work,
> or by considering more complex algorithm patterns and realistic tasks.

---

> ### Author Response · Authors · 2025-08-15
> **Response to Reviewer Bz3T (3/3)**
>
> **Weakness: The insights are either trivial or closely aligned with intuition. Some key findings -- such as "finer-grained task decomposition leads to better accuracy but higher costs" -- are expected and may not offer significant new understanding.**
>
> We respectfully disagree with the claim that "the insights are either trivial or closely aligned with intuition".
>
> To clarify, the statement mentioned in the review -- "finer-grained task decomposition leads to better accuracy but higher costs" -- is NOT a key finding of our work.
> In contrast, this is exactly the kind of common misconception that our Insights 1 and 2 intend to refute:
>
> * Insight 1 (along with the analysis in Section 3.1) indicates that finer-grained task decomposition does not necessarily lead to higher cost of the overall LLM-based algorithm. Depending on factors like the quadratic complexity of Transformer, the choice of cost metric, or the parallelism degree, the overall cost can be monotonely increasing / monotonely decreasing / non-monotone in the granularity of task decomposition.
> * Insight 2 (along with the analysis in Section 3.2) refutes the claim that "finer-grained task decomposition leads to better accuracy". This property requires certain conditions; for example, it is valid if each sub-task error is monotonely increasing in the sub-task size, and the overall error can be bounded by a smooth aggregation of sub-task errors. Otherwise, there exist situations (e.g., the retrieval algorithm with a Type-2 LLM, as elaborated in Section 3.2.3) where this property is invalid.
>
> We hope that this clarification helps explain why the insights and analysis derived with the proposed framework are indeed non-trivial and can be counter-intuitive.
>
> ---
>
> Thank you again for your thoughtful review, which helps us a lot in improving this work!
> Our plan for the upcoming week is to revise the manuscript accordingly.
> In the meantime, we are always open to discussions if you have any further concern or suggestion :)

---

### Review · Reviewer_cp4X · 2025-08-06

**Summary Of Contributions:**

This paper presents an analytical framework for designing and analyzing LLM-based algorithms using computational graphs. The key idea is to model LLM algorithms as graphs where nodes are either LLM calls or non-LLM operations, then analyze how design choices (task decomposition granularity, prompting strategies) affect accuracy and efficiency.
The main contributions are: a formal computational graph representation, cost and error analysis methodology, five practical insights from case studies on parallel/hierarchical/recursive decomposition patterns, and empirical validation on synthetic tasks. The framework aims to replace trial-and-error approaches with principled analysis of accuracy-efficiency tradeoffs.

**Audience:**

Yes

**Claims And Evidence:**

Yes

**Requested Changes:**

I need evidence this works for realistic tasks. Add at least one case study on a real application, such as code generation, mathematical reasoning, document analysis, etc. Show me the framework provides actionable insights beyond synthetic examples.
I also need validation of the additive error assumption. Provide empirical evidence it holds for complex reasoning, or discuss when it breaks down and what that means.
Address scalability concerns: how does this work when optimal decomposition isn't obvious? When you lack ground truth? When LLM behavior is prompt-sensitive?
Additional improvements: test on diverse model families, provide concrete implementation guidance, compare predictions with established benchmarks.

**Strengths And Weaknesses:**

I appreciate that this work tackles an important problem i.e., most LLM system design currently relies on ad-hoc experimentation. The computational graph abstraction is elegant and the mathematical treatment is rigorous. The derived insights about optimal granularity and cost-accuracy tradeoffs feel actionable.
However, my main concern is the complete lack of realistic validation. All experiments are on toy problems: counting digits, sorting numbers, needle-in-haystack tasks. I have no evidence this framework would provide useful insights for actual applications like software engineering agents or complex reasoning systems. The gap between "optimal chunk size for counting digits" and real-world LLM applications feels enormous.
The assumptions also worry me. The additive error model assumes errors propagate linearly, but in my experience, reasoning errors cascade unpredictably. The framework treats LLM behavior as stable functions, ignoring context-sensitivity and prompt-dependence that characterize real usage.
Finally, it's unclear how practitioners would apply this. The framework requires characterizing LLM error functions and estimating parameters, but there's little guidance on doing this for novel tasks without ground truth.

---

> ### Author Response · Authors · 2025-08-15
> **Response to Reviewer cp4X (1/3)**
>
> Thank you for your detailed and constructive review of our work!
> We address your concerns and questions in the following.
>
> ---
>
> **Question: gap between this work and real-world LLM applications.**
>
> To address your concern on this aspect,
> we would like to clarify that the scope of this work is to make a first step towards formal and rigorous analysis of LLM-based algorithms.
> We do not intend to overclaim the scope or contributions of the current work,
> or the implications of our obtained results for the most complex real-world LLM applications.
>
> For the purpose of building a solid foundation for rigorously understanding the accuracy and efficiency of LLM-based algorithms,
> we have chosen to stick with synthetic tasks, which can be regarded as abstract versions of practical LLM usage.
> As discussed in Appendix A.1, this brings various benefits, such as avoiding data contamination issues, facilitating unambiguous and precise measurement of accuracy, and allowing full transparency and control over task configurations.
>
> Applying the proposed framework to a more realistic task or complex agentic workflow is beyond the scope of our current focus in this work.
> Nonetheless, we believe that our work has shown the potential of providing useful insights for real-world applications:
> * To clarify, our case studies are certainly not limited to those simple tasks -- "counting digits, sorting numbers, needle-in-haystack tasks" -- mentioned in the review.
> More complex examples of parallel decomposition that we have investigated include retrieval-augmented generation (Appendix A.5) and long-text summarization with chunking (Appendix A.6).
> We have also made substantial efforts to extend our analysis to more complex and realistic settings,
> such as iterative retrieval and multi-hop reasoning over long documents (Section 4.2),
> and recursive algorithm for active information seeking and aggregation (Section 4.3);
> these settings and algorithms are much more dynamic than parallel decomposition,
> and resemble the settings in some prior empirical works mentioned at the beginning of Sections 4.2 and 4.3.
> A practical element mentioned in your review, namely prompt-dependence, has been taken into account in these case studies,
> where we investigate the impacts of CoT prompting formally.
> * Investigation for simple algorithm patterns and basic tasks could be beneficial for studying more complex real-world LLM applications in future work.
> For example, [Anthropic's "building effective agents" blog](https://www.anthropic.com/engineering/building-effective-agents)
> highlights that the most successful LLM agents in industry are often built with *simple, composable patterns*.
> We anticipate that our analysis for simple patterns (like parallel decomposition) could be used as building blocks for analysis of more complex algorithms in the future.
> * The insights that we have derived in simple settings can be potentially useful in broader settings.
> For example, Insights 1 and 2 can, at the minimum, warn practitioners against the common misconceptions that "finer-grained task decomposition implies higher accuracy / higher total cost of the overall LLM-based algorithm", beyond the parallel decomposition setting where these insights were originally derived.
>
>
> It would be intriguing future work to expand the proposed framework and validate its usefulness,
> by taking into account some practical elements that we might have abstracted away in the current work,
> or by considering more complex algorithm patterns and realistic tasks.
>
>
> We hope that our response above has assured you that
> the current work makes meaningful contributions to the LLM community and shows sufficient practical relevance.
> We plan to incorporate the above response into the revision,
> better clarifying the scope of the current work and acknowledging its limitations in terms of more realistic validation.

---

> ### Author Response · Authors · 2025-08-15
> **Response to Reviewer cp4X (2/3)**
>
> **Question: validation of the additive error assumption**
>
> Before we provide validation for Assumption 1 ("additive error and bounded sensitivity"),
> we would like to first make some clarifications.
> The main message that we try to convey in Section 4.1 and Insight 3 is that "we have the error bound in Proposition 2 *for scenarios where Assumption 1 holds true*".
> We certainly do not expect or claim that Assumption 1 should be universally valid for all LLM-based algorithms.
> These clarifications will be incorporated into Section 4.1 in our revision,
> to prevent confusion or misunderstanding regarding the intended scope of Assumption 1 and Insight 3.
>
> Assumption 1 can be rigorously proved for a non-LLM node in some cases.
> For example, considering the aggregation node in the counting algorithm,
> our analysis in Section 3.2.1 implies that Assumption 1 holds true with node-specific additive error $\mathcal{E}\_{v} = 0$, and sensitivity parameter $S = 1$ for absolute counting error,
> or $S = 1 / k$ for normalized counting error.
> As another example, considering the aggregation node in the sorting algorithm,
> Proposition 1 indicates that its output $\ell\_{\infty}$ error is bounded by $\mathcal{E} \le \max \\{\mathcal{E}\_1, \dots, \mathcal{E}\_k \\} \le \mathcal{E}\_1 + \dots + \mathcal{E}\_k$,
> which means Assumption 1 holds with $\mathcal{E}\_{v} = 0$ and $S = 1$.
>
> For an LLM node, "additive error and bounded sensitivity" is an assumption that we impose on the *stability* of LLM's behavior when solving a *sufficiently simple* (with respect to its capability) sub-task.
> One motivating example is the recursive algorithm from our case study in Section 4.3.
> This algorithm involves many LLM calls that are asked to calculate a 1-Lipschitz function (such as sum or max) of several input variables.
> We have observed in our empirical study that the LLM, with CoT prompting, performs pretty well for such sub-tasks,
> behaving with sufficient stability and making small errors occasionally.
> It is this property that makes the final error of the overall algorithm stays low and grows slowly (rather than blows up rapidly) with the task complexity and size of computational graph,
> as can be observed in Figures 20 and 21 in Appendix D.5.
>
>
> ---
>
> **Weakness: the framework requires characterizing LLM error functions and estimating parameters**
>
> We agree that characterizing LLM error functions could be a challenge when applying the proposed framework in a novel task.
> In general, we would require at least a handful of "training samples" for the task,
> or for sub-tasks derived from it,
> so that we can evaluate the LLM's capabilities and limitations on the task / sub-task via profiling.
> This evaluation can be quantitative or qualitative, precise or coarse, depending on the purpose of analysis;
> some related discussions can be found in Remark 1.
>
>
> ---
>
>
> **Additional improvement: test on diverse model families**
>
> We have conducted experiments in this work with a variety of model families, including Llama, Qwen and GPT,
> supported by ollama on a Macbook Pro, vLLM on A100-80G GPUs, or API calls.
> We believe that this diversity of model families would be sufficient for the purpose of our empirical study in this work.
>
>
> ---
>
>
> **Additional improvement: provide concrete implementation guidance**
>
> We have deferred implementation details to Appendix A, C and D, in order to keep the main text concise.
> Please let us know if you feel that it would be better to present certain implementation details in the main text.
> To further enhance the reproducibility of our empirical results,
> we have also uploaded the source code to OpenReview during the discussion period.

---

> ### Author Response · Authors · 2025-08-15
> **Response to Reviewer cp4X (3/3)**
>
> Thank you again for your thoughtful review, which helps us a lot in improving this work!
> Our plan for the upcoming week is to revise the manuscript accordingly.
> In the meantime, we are always open to discussions if you have any further concern or suggestion :)

---

### Author Response · Authors · 2025-08-15
**General response to reviewers**

Dear reviewers,

We sincerely appreciate your detailed and thoughtful reviews, which acknowledge the strengths of this work while pointing out directions for further improvements.

We have just posted our response to each reviewer.
We plan to revise the paper accordingly in the upcoming week, and submit a revised version before the end of the reviewer-author discussion period.
In the meantime, we are always open to discussions if you have further concerns or suggestions about our work :)


Best,

Authors

---

> ### Author Response · Authors · 2025-08-20
> **Follow-up: revised version uploaded**
>
> Dear reviewers,
>
> &nbsp;
>
> We have uploaded the revised version of our manuscript, with the following modifications:
>
> - In Section 4.1, we clarify the intended scope of Assumption 1 and provide justifications for it.
> - In Section 3.1, we explain why we have chosen to stick with synthetic settings for our empirical studies.
> In Section 6, we elaborate on the limitation of this work in terms of lacking experiments for more realistic settings,
> as well as potential directions for future work.
> - In Section 5, we compare this work with the literature of distributed computing (e.g., MapReduce) and classical / non-LLM algorithm design and analysis.
> - Other modifications corresponding to the remaining comments by the reviewers.
>
> We hope that this revision has addressed your concerns about the current work.
> Feel free to let us know if you have any comment!
>
> &nbsp;
>
> Best,
>
> Authors

---

### Decision · Action_Editor_P5NB · 2025-09-07

**Recommendation:** Accept with minor revision

**Additional Comments:**

In general, this is a good paper, which basically reaches the standard for publications in TMLR. According to the feedback from the reviewers, I would gently suggest that the authors make the following updates in the final version of the paper:

- Reinforce the discussion about the connection between the proposed framework and existing retrieval-based algorithms.

- Revise the paper to explicitly discuss the connection and intergration about each building block anaylsized by the paper.

**Audience:**

Yes

**Audience Explanation:**

The paper discusses an interesting problem that would be interesting for the machine learning community.

**Claims And Evidence:**

Yes

**Claims Explanation:**

The study presented in this paper is relatively solid, and the claims and statements are supported by corresponding evidence.

---

> ### Author Response · Authors · 2025-09-30
> **Final version uploaded**
>
> Dear Action Editor,
>
> &nbsp;
>
> We sincerely appreciate your feedback and decision regarding our work.
> We have uploaded the final version of our paper, which incorporates all requested revisions.
> Please let us know if any further adjustments are needed.
> Thank you!
>
> &nbsp;
>
> Best,
>
> Authors